# CAMO: Convergence-Aware Multi-Fidelity Bayesian Optimization*

**Wei W. Xing**
University of Sheffield
w.xing@sheffield.ac.uk

**Zhenjie Lu**[†]
SUSTeh & Shenzhen University
12432976@mail.sustech.edu.cn

**Akeel A. Shah** [‡]
Chongqing University
akeelshah@cqu.edu.cn

## Abstract

Existing Multi-fidelity Bayesian Optimization (MFBO) methods ignore the convergence behavior of the multi-fidelity surrogate as the fidelity increases, leading to inefficient exploration and suboptimal performance. We introduce CAMO (Convergence-Aware Multi-fidelity Optimization), a principled framework based on Linear Fidelity Differential Equations (LFiDEs) that explicitly encodes convergence of fidelity-indexed outputs and employs a closed-form nonstationary kernel. We rigorously prove the existence and pointwise/uniform convergence to the high fidelity surrogate under mild restrictions and provide new convergence results for general FiDEs using smooth, non-smooth and even non-convex Lyapunov functions, establishing a bridge between MFBO and the theory of subgradient flows in non-smooth optimization theory. Combined with a fidelity-aware acquisition function, CAMO outperforms state-of-the-art MFBO methods on a majority of synthetic and real-world benchmarks, with up to a four-fold improvement in optimization performance and a dramatic speed-up in convergence. CAMO offers a tractable and theoretically grounded approach to convergence-aware MFBO.

## 1 Introduction

Bayesian optimization (BO) is an efficient method for optimising costly black-box functions, finding applications in various domains of engineering and science [1]. BO leverages a surrogate model, typically a Gaussian Process (GP), to approximate the objective function and guide the search towards promising regions of parameter space using an acquisition function. This approach is particularly effective when the black-box function evaluations are costly or time-consuming. In many applications, black-box solutions can be obtained at varying levels of fidelity. Lower-fidelity solutions are associated with lower accuracy and lower computational costs, the latter of which typically increase dramatically as fidelity increases. The fidelity can be controlled in a number of ways, by varying the modelling choices or the solver settings. For example, electronic-structure methods involve increasing levels of theory, with exponentially increasing costs [2]. Different fidelities can also be defined *via* the numerical formulation, e.g., use different mesh sizes, time steps or convergence thresholds. For the purposes of prediction, and especially for optimization, it is of course desirable and sometimes essential to predict at the highest fidelity. In practice, however, the number of high-fidelity results obtainable within a typical computational budget or time frame is often constrained.

Multi-Fidelity Bayesian optimization (MFBO) methods aim to reduce the cost of optimising high-fidelity black-box functions by leveraging lower-fidelity approximations [3, 4]. In typical scenarios, the multi-fidelity objective evolves towards the highest fidelity function as the fidelity parameter increases. Existing MFBO approaches ignore this evolution and treat the fidelity as an unstructured input, which can lead to inefficient exploration in high-cost regions and, ultimately, suboptimal

---

*Code is available at https://github.com/IceLab-X/CAMO. [†]The majority of this work was conducted while Zhenjie Lu was at Shenzhen University. [‡]Corresponding author

performance. In this paper we propose CAMO, a principled framework for MFBO that uses a convergence-aware surrogate model derived from linear fidelity differential equations (LiFiDEs). The "convergence" here refers to a systematic progression towards a high-fidelity function. In contrast to conventional methods, CAMO structurally enforces convergence as fidelity increases, ensuring that predictions at lower fidelities provably approach the high-fidelity objective under mild regularity assumptions. Our key contributions are: (1) a convergence-aware surrogate that encodes the structural tendency of simulations to converge as fidelity increases; (2) a closed form non-stationary LiFiDE kernel that captures fidelity-wise correlations, eliminating the need to specify a relationship between fidelities *a-priori* (CAMO adaptively learns convergence behaviour from the data); (3) a general theory for fidelity-indexed systems using Lyapunov functions, establishing uniform and pointwise convergence under smooth, non-smooth and even non-convex scenarios, and revealing a formal link to subgradient-based dynamics in modern non-smooth optimization theory; (4) when combined with BOCA (Bayesian optimization with Continuous Approximations) [3] to leverage both fidelity-aware exploration and convergence-aware modelling we show that CAMO consistently outperforms MFBO baselines on synthetic and real-world benchmarks.

## 2 Related Work

Bayesian optimization (BO) [5] uses a surrogate model, typically a Gaussian Process (GP) [6] (see Appendix A), to approximate the objective function, together with an acquisition function to guide the search towards promising regions. Acquisition functions balance exploration and exploitation to efficiently search for a global optimum. Popular choices include the Expected Improvement (EI) [7], the Upper Confidence Bound (UCB) [8], and the Probability of Improvement (PI) [9]. MFBO extends standard BO by utilising solutions at different fidelity levels, aiming to reduce the overall optimization cost and improve convergence speed. Most of the work in MFBO has focused on discrete fidelity sets. Huang et al. [10] proposed Sequential Kriging optimization (SKO), which employs a hierarchical GP approach to capture the correlations between different fidelity levels. Kandasamy et al. [11] developed the Multi-Fidelity (MF) Gaussian Process Upper Confidence Bound (MF-GP-UCB) method based on a generic GP formulation. Le Gratiet and Garnier [12] proposed Recursive Co-Kriging, which builds a hierarchy of GP models of the residual between successive fidelity levels. Perdikaris et al. [13] used low-fidelity information as an input to a high-fidelity GP, leading to a deep GP structure, while Cutajar et al. [14] directly employed a deep GP to learn nonlinear mappings between fidelities.

In many real-world applications, the fidelity level can be treated as a continuous variable (e.g., a mesh size or time step). This has led to the development of continuous-fidelity MFBO methods such as BOCA [3], which extends MF-GP-UCB by using a two-step procedure to select the next query point. Poloczek et al. [15] introduced the Multi-Information Source optimization (MISO) framework, which treats the objective function as a linear combination of GPs, each corresponding to a different fidelity level. Klein et al. [16] proposed FABOLAS, combining a GP with a linear fidelity kernel and an Entropy Search acquisition function. Wu and Frazier [17] developed the continuous fidelity knowledge gradient (cfKG) method based on a GP with a knowledge gradient acquisition function. In contrast, Li et al. [18] took a deep learning approach with Deep Neural Network MF Bayesian optimization (DNN-MFBO), which uses a fidelity-wise Gauss-Hermite quadrature and a moment-matching mutual information acquisition function. These studies illustrated the benefits of continuous-fidelity MFBO over discrete formulations. Most practical GP-based methods, however, lack insights into the convergence behaviour of the objective function across fidelity levels, leading to the over-exploration of costly regions and, therefore, suboptimal performance. Although DNNs can in theory capture the complex relationships between fidelity levels, observations at high fidelity are typically sparse, leading to overfitting and high model variance. Recent advances in MF modelling include fidelity differential equations (FiDE) [19], modelled by NeuralODE [20] or continuous autoregression (CAR) [19] to capture the convergence behaviour of the objective function. The training times for NeuralODE are up to $10^4$-fold higher than those for CAR. In this work we therefore build upon the FiDE concept to propose a tractable and scalable MFBO method.

Hyperparameter optimization (HPO) and neural architecture search (NAS) are not considered core applications of CAMO, which is focused on expensive blackbox solvers in science and engineering. In such applications, continuous fidelity indices are unambiguously defined. They arise naturally as controllable, continuous parameters intrinsic to the physical model or numerical approximation. Their effect on the objective function is deterministic, with typically a monotonic convergence toward the highest fidelity. In contrast, HPO/NAS operate in a setting in which fidelity is stochastic and algorithm-dependent, e.g., epoch number and budget are algorithmic artefacts with no deterministic

convergence law or trajectory. The cross-fidelity structure is non-smooth and data-dependent, which requires empirical (ideally discrete-fidelity) MFBO methods such as BOHB [21], DyHPO [22] and FastBO [23]. Adapting CAMO to HPO/NAS would potentially require reformulating the underlying convergence model as a stochastic process, which is orthogonal to the contributions of this paper.

# 3 Background

## 3.1 Problem Formulation

We consider the problem of optimising a function $y : \mathcal{X} \times [0, T) \to \mathbb{R}$, $(\mathbf{x}, t) \mapsto y(\mathbf{x}, t)$, in which $\mathcal{X} \subseteq \mathbb{R}^d$ is a design variable space, and $[0, T) \subset \mathbb{R}$ is a range of fidelity levels. $T = \infty$ is allowed and is of particular interest. We assume that both the accuracy and computational costs of evaluating $y(x, t)$ increases with $t$. The goal is to find the design variable $x^* = \arg\max_{\mathbf{x} \in \mathcal{X}} y(\mathbf{x}, T)$ that maximises the high-fidelity objective while minimising the total cost of evaluations. The observations $y_i$ at different $x$ and $t$ incorporate additive Gaussian noise (see Appendix A). $L^p(\mathcal{X})$ ($1 \leq p < \infty$) denotes the space of functions $f : \mathcal{X} \to \mathbb{R}$ such that $\|f\|_{L^p(\mathcal{X})}^p = \int_{\mathcal{X}} |f(\mathbf{x})|^p \, d\mathbf{x} < \infty$, where $d\mathbf{x}$ denotes integration w.r.t. Lebesgue measure. $C(\mathcal{X})$ denotes the space of continuous functions $f : \mathcal{X} \to \mathbb{R}$. $C^{a,b}(\mathcal{X} \times [0, \infty))$ denotes the space of functions $f : \mathcal{X} \times [0, \infty) \to \mathbb{R}$ with continuous derivatives up or orders $a$ and $b$ in $\mathbf{x}$ and $t$, with similar notation for functions of more than two variables (the superscript is omitted if $a = b = 0$). $\|f\|_{L^\infty(A)} = \text{ess sup}_{\mathbf{x} \in \mathcal{X}} f(\mathbf{x})$ denotes the $L^\infty$ norm. $AC([0, \infty))$ denotes the space of absolutely continuous functions and $L^1_{\text{loc}}([0, \infty))$ is the space of locally integrable functions on $[0, \infty)$.

# 4 CAMO and Theoretical Results

## 4.1 Multi-Fidelity via FiDE: Well-Posedness and Link to Subgradient Flows

MFBO requires a surrogate to model $y(\mathbf{x}, t)$, often in the form of a GP: $y(\mathbf{x}, t) \sim \mathcal{GP}(0, k(\mathbf{x}, t, \mathbf{x}', t'))$ with some kernel $k$. Li et al. [20] and Xing et al. [19] proposed that $y(\mathbf{x}, t)$ can be modelled as the solution to a Fidelity Differential Equation (FiDE), namely

$$\partial_t y(\mathbf{x}, t) = \phi(\mathbf{x}, t, y(\mathbf{x}, t)), \quad (\mathbf{x}, t) \in \mathcal{X} \times [0, T), \tag{1}$$

subject to some initial condition $y(\mathbf{x}, 0) = y_0(\mathbf{x})$. Here, $\phi(\mathbf{x}, t, y)$ describes the system 'dynamics', with the fidelity parameter $t$ playing the role of physical time. Without loss of generality, we set $T = \infty$ for the theoretical results. Treating the MF system as a dynamical system is entirely natural since the index $t$ is strictly ordered. To conduct MFBO with such a model we need guarantees on the well-posedness of the FiDE problem (1), i.e, that unique solutions exist and that $y(\mathbf{x}, t)$ converges to a unique equilibrium solution $y_\infty(\mathbf{x}) = \lim_{t \to \infty} y(\mathbf{x}, t)$. The convergence can be interpreted in a pathwise sense, e.g., for each fixed input $\mathbf{x}$, the solution trajectory converges deterministically to the high-fidelity target. In this work we do not consider stochastic FiDEs, i.e., convergence in stochastic process topologies. The dynamics are entirely deterministic once the training data are fixed.

Convergence can be studied using a Lyapunov analysis. A Lyapunov function $V : (\mathbf{x}, t, y) \mapsto V(\mathbf{x}, t, y)$ (dependent on $\phi$) serves as a energy-like functional that measures the deviation of $y(\mathbf{x}, t)$ from an equilibrium. Our first result establishes minimal conditions for the existence and uniqueness of classical solutions pointwise in $\mathbf{x}$ (see Appendix B.1 for the proof)

**Lemma 1.** *[Existence and Regularity for General FiDEs] Let $\mathcal{X} \subseteq \mathbb{R}^d$ be a non-empty set. Consider the system (1) and assume that: (1) for each fixed $\mathbf{x} \in \mathcal{X}$, the function $(t, y) \mapsto \phi(\mathbf{x}, t, y)$ is continuous on $[0, \infty) \times \mathbb{R}$; (2) for each fixed $\mathbf{x} \in \mathcal{X}$, $\phi(\mathbf{x}, t, y)$ is locally Lipschitz in $y$, uniformly for $t$ in compact subsets of $[0, \infty)$; (3) the initial condition $y_0(\mathbf{x}) \in \mathbb{R}$ is finite for each $\mathbf{x}$. Then for each fixed $\mathbf{x} \in \mathcal{X}$, there exists a unique maximal solution $y(\mathbf{x}, t) \in C^1([0, t_*(\mathbf{x})))$ defined on some maximal interval $[0, t_*(\mathbf{x}))$, with $0 < t_*(\mathbf{x}) \leq \infty$. Moreover: (a) if $y(\mathbf{x}, t)$ remains bounded for all $t \in [0, \infty)$, then $t_*(\mathbf{x}) = \infty$ and the solution exists globally on $[0, \infty)$; (b) if $\phi$ is globally Lipschitz in $y$, uniformly in $t$, then the solution exists globally on $[0, \infty)$.*

The following Lemmas (see Appendix B.2 for the proofs) establish minimal regularity requirements on $V$, $y_0$ and $\mathcal{X}$ for pointwise and uniform (in $\mathbf{x}$) convergence to hold, respectively.

**Lemma 2** (Pointwise Convergence for FiDE). *Let $y(\mathbf{x}, \cdot) \in C^1([0, \infty))$, $\mathbf{x} \in \mathcal{X}$ be a solution to (1) with $y_0(\mathbf{x}) < \infty$ for each $\mathbf{x} \in \mathcal{X}$. Suppose there exists a Lyapunov function $V : \mathcal{X} \times [0, \infty) \times \mathbb{R} \to \mathbb{R}$, $(\mathbf{x}, t, y) \mapsto V(\mathbf{x}, t, y)$, and that the following conditions are satisfied: (1) $V \in C^{0,1,1}(\mathcal{X} \times [0, \infty) \times \mathbb{R})$; (2) $V(\mathbf{x}, t, y) \geq 0$, $\forall (\mathbf{x}, t, y) \in \mathcal{X} \times [0, \infty) \times \mathbb{R}$ and $V(\mathbf{x}, t, y) = 0$ iff $y = y_\infty(\mathbf{x})$; (3) for all*

$(\mathbf{x}, t) \in \mathcal{X} \times [0, \infty)$

$$\dot{V}(\mathbf{x}, t, y) = \partial_y V(\mathbf{x}, t, y)\phi(\mathbf{x}, t, y) + \partial_t V(\mathbf{x}, t, y) < 0 \tag{2}$$

Then $y(\mathbf{x}, t) \to y_\infty(\mathbf{x})$ pointwise on $\mathcal{X}$, i.e., $\lim_{t \to \infty} y(\mathbf{x}, t) = y_\infty(\mathbf{x})$ for each $\mathbf{x} \in \mathcal{X}$.

**Lemma 3** (Uniform Convergence for FiDE). *Let $y(\mathbf{x}, \cdot) \in C^1([0, \infty))$, $\mathbf{x} \in \mathcal{X}$ is a solution to (1). Suppose there exists a Lyapunov function $V : \mathcal{X} \times [0, \infty) \times \mathbb{R} \to \mathbb{R}$, $(\mathbf{x}, t, y) \mapsto V(\mathbf{x}, t, y)$ and the following conditions are satisfied: (1) $y_0(\mathbf{x}) \in C(\mathcal{X})$; (2) $V \in C^{0,1,2}(\mathcal{X} \times [0, \infty) \times \mathbb{R})$; (3) $V(\mathbf{x}, t, y) \geq 0, \forall (\mathbf{x}, t, y) \in \mathcal{X} \times [0, \infty) \times \mathbb{R}$ and $V(\mathbf{x}, t, y) = 0$ iff $y = y_\infty(\mathbf{x})$; (4) $\exists c_{\min} > 0$ such that $c(\mathbf{x}, t) \geq c_{\min} > 0, \forall (\mathbf{x}, t) \in \mathcal{X} \times [0, \infty)$, in which $c(\mathbf{x}, t) = \frac{1}{2}\partial_{yy}^2 V(\mathbf{x}, t, y_\infty(\mathbf{x}))$; (5) for all $\mathbf{x}, t \in \mathcal{X} \times [0, \infty), \exists \alpha > 0$ such that*

$$\dot{V}(\mathbf{x}, t, y) = \partial_y V(\mathbf{x}, t, y)\phi(\mathbf{x}, t, y) + \partial_t V(\mathbf{x}, t, y) \leq -\alpha V(\mathbf{x}, t, y); \tag{3}$$

*(6) $\exists V_{\max}$ such that $\|V(\mathbf{x}, 0, y(\mathbf{x}, 0))\|_{L^\infty(\mathcal{X})} \leq V_{\max}$. Then $\lim_{t \to \infty} y(\mathbf{x}, t) = y_\infty(\mathbf{x})$ uniformly on $\mathcal{X}$, with*

$$\|\mathbf{y}(\mathbf{x}, t) - y_\infty(\mathbf{x})\|_{L^\infty(\mathcal{X})} \leq c e^{-\alpha t}, \quad c = \sqrt{V_{\max}}/c_{\min}. \tag{4}$$

If $y(\mathbf{x}, t) \in C(\mathcal{X} \times [0, \infty))$ then $y_\infty \in C(\mathcal{X})$ since the uniform limit of continuous functions is continuous. The uniform-convergence conditions are readily satisfied for functions $V$ that have at most a mild dependence on $t$, such as exponential decay or bounded variations, notably quadratic functions $V(\mathbf{x}, t, y) = a(\mathbf{x}, t)(y - y_\infty(\mathbf{x}))^2$, $a(\mathbf{x}, t) \geq a_0 > 0$ uniformly. Non-smooth Lyapunov functions such as $V_{|\cdot|} = |y(x, t) - y_\infty(x)|$ permit a convergence analysis even if $y(\mathbf{x}, t)$ exhibits non-smooth behaviour due to noise or sparse evaluations at low fidelities (in which case $\partial_t y$ has to be interpreted in a distributional sense). Non-smooth $V$ allow for the certification of asymptotic behaviour and exponential convergence in such non-smooth settings. Lemma B1 in Appendix B.3 establishes uniform convergence for $V_{|\cdot|}(\mathbf{x}, t, y)$, which is readily extended to Elastic-Net-type and Group-Sparsity-type $V$. In fact, the result extends to non-convex Lyapunov functions via the Clarke subdifferential [24] (see Appendix B.3 for a proof)

**Lemma 4.** *Let $\mathcal{X}$ be a nonempty set, and let $y : \mathcal{X} \times [0, \infty) \to \mathbb{R}$ be such that for each $\mathbf{x} \in \mathcal{X}$, $y(\mathbf{x}, \cdot) \in C^1[0, \infty)$. Let $V : \mathcal{X} \times \mathbb{R} \to \mathbb{R}$, $(\mathbf{x}, y) \to V(\mathbf{x}, y(\mathbf{x}, t))$ satisfy the following conditions. (1) For every fixed $(\mathbf{x}, t)$, the map $y \mapsto V(\mathbf{x}, y, t)$ is locally Lipschitz. (2) For each fixed $(\mathbf{x}, t) \in \mathcal{X} \times [0, \infty)$, define $u[\mathbf{x}](\cdot) : \mathbb{R} \to \mathbb{R}$, $y \mapsto V(\mathbf{x}, y)$, and for a fixed $\mathbf{x} \in \mathcal{X}$, define $v[\mathbf{x}](\cdot) : [0, \infty) \to \mathbb{R}$, $t \mapsto V(\mathbf{x}, y(\mathbf{x}, t))$; assume that for all $(\mathbf{x}, t) \in \mathcal{X} \times [0, \infty)$ and all $\xi(\mathbf{x}, t) \in \partial_C u[\mathbf{x}](y)$ the inequality $\xi(\mathbf{x}, t)\phi(\mathbf{x}, t, y(\mathbf{x}, t)) \leq -\alpha v[\mathbf{x}](t)$ holds for some constant $\alpha > 0$, independent of $\mathbf{x}$ and $t$. (3) The initial deviation is bounded, i.e., $\|y(\cdot, 0) - y_\infty(\cdot)\|_{L^\infty(\mathcal{X})} < \infty$. Then, for all $t \in [0, \infty)$, we have*

$$\|y(\cdot, t) - y_\infty(\cdot)\|_{L^\infty(\mathcal{X})} \leq \|y(\cdot, 0) - y_\infty(\cdot)\|_{L^\infty(\mathcal{X})} e^{-\alpha t}. \tag{5}$$

These lemmas reveal a close analogy between the convergence of FiDEs and subgradient-based dynamics in non-smooth convex optimization. For a non-smooth Lyapunov function $V$, the evolution of $y(\mathbf{x}, t)$ can be analysed using differential inclusions involving the subdifferential $\partial_y V$. This structure mirrors subgradient flows and proximal-map methods commonly encountered in sparse learning and variational optimization. Under mild conditions, such dynamics guarantee exponential convergence toward the high-fidelity target $y_\infty(\mathbf{x})$, with $y(\mathbf{x}, t)$ following a descent trajectory that reduces the fidelity error encoded by $V$. Quantitatively, subgradient dynamics induced by non-smooth $V$ can be interpreted as the limit of smooth gradient flows applied to a Moreau-Yosida regularisation [25]. For instance, define the smoothed functional

$$V_\lambda(\mathbf{x}, y) = \inf_{z \in \mathbb{R}} \left\{ |z - y_\infty(\mathbf{x})| + \frac{1}{2\lambda}(y - z)^2 \right\} \tag{6}$$

as a regularised approximation of $V(\mathbf{x}, y) = |y - y_\infty(\mathbf{x})|$. The associated gradient flow $\partial_t y_\lambda(\mathbf{x}, t) = -\nabla_y V_\lambda(\mathbf{x}, y_\lambda(\mathbf{x}, t))$ converges (in the sense of graphical or Mosco convergence) to the subgradient inclusion $\partial_t y(\mathbf{x}, t) \in -\partial |y(\mathbf{x}, t) - y_\infty(\mathbf{x})|$ as $\lambda \to 0$. Although FiDEs do not explicitly follow this gradient flow, the analogy provides a useful interpretation: the FiDE-induced surrogate evolution resembles a continuous-$t$ analogue of proximal or subgradient descent, where Moreau smoothing offers stability while preserving convergence guarantees. We refer to Appendix B.4 for further discussion.

## 4.2 Linear FiDEs and Convergence Guarantees

While Lemmas 2–4 provide a robust and general framework for ensuring convergence, finding a suitable Lyapunov function for a specific $\phi$ is complicated by the fact that the latter is generally unknown and at the very least is difficult to model. Li et al. [20] resort to learning $\phi$ and $y_0(\mathbf{x})$ using neural networks. In contrast, Xing et al. [19] considered a more tractable case in which $\phi$ admits a linear form in $y$. In this paper we adopt the same approach, first establishing the existence of a unique equilibrium and the validity of a constructive variation-of-constants approach. If $\phi$ admits a linear form in $y$, we obtain the following linear FiDE for each fixed $\mathbf{x} \in \mathcal{X}$

$$\partial_t y(\mathbf{x}, t) = \beta(\mathbf{x}, t)\, y(\mathbf{x}, t) + u(\mathbf{x}, t), \quad t \in [0, \infty), \tag{7}$$

subject to some $y_0(\mathbf{x}) = y(\mathbf{x}, 0) \in \mathbb{R}$ (i.e., finite for each $\mathbf{x}$). This is a first-order linear, non-autonomous ODE in $t$. We first establish the global existence of unique solutions and validity of the variation-of-constants formula pointwise in $\mathbf{x} \in \mathcal{X}$ (see Appendix C.1 for a proof).

**Lemma 5.** *(Existence and Uniqueness for Linear FiDE) Let $\mathcal{X} \subset \mathbb{R}^d$ be given and suppose that: (A1) $\beta(\mathbf{x}, \cdot), u(\mathbf{x}, \cdot) \in C([0, \infty))$; and (A2) $y_0(\mathbf{x}) \in \mathbb{R}$. Then there exists a maximal time $t_*(\mathbf{x})$ such that a unique local solution $y(\mathbf{x}, \cdot) \in C^1([0, t_*(\mathbf{x})))$ satisfying (7) exists and is given by*

$$y(\mathbf{x}, t) = e^{\int_0^t \beta(\mathbf{x}, s)ds} y_0(\mathbf{x}) + \int_0^t e^{\int_s^t \beta(\mathbf{x}, z)dz} u(\mathbf{x}, s)ds, \quad t \in [0, t_*(\mathbf{x})). \tag{8}$$

*Moreover, either $t_*(\mathbf{x}) = \infty$ or $\limsup_{t \to t_*(\mathbf{x})^-} |y(\mathbf{x}, t)| = \infty$ (finite-time blowup). If in addition: (A3) $\|\beta(\mathbf{x}, t)\|_{L^\infty(\mathcal{X} \times [0, \infty))} < \infty$ and $\|u(\mathbf{x}, t)\|_{L^\infty(\mathcal{X} \times [0, \infty))} < \infty$, then $t_*(\mathbf{x}) = \infty$. Now let $\mathcal{X} \subset \mathbb{R}^d$ be compact and suppose instead that: (B1) $\beta(\mathbf{x}, t), u(\mathbf{x}, t) \in C(\mathcal{X} \times [0, \infty))$; (B2) $y_0(\mathbf{x}) \in C(\mathcal{X})$; and (A3) above. Then there exists a unique solution $y(\mathbf{x}, t) \in C(\mathcal{X} \times [0, \infty))$, $y(\mathbf{x}, \cdot) \in C^1([0, \infty))$ of (7) given by (8).*

Having established the conditions for existence, the following two lemmas guarantee pointwise or uniform convergence of $y(\mathbf{x}, t)$ on $\mathcal{X}$ towards an equilibrium under some general conditions on $\mathcal{X}$, $y_0(\mathbf{x})$, $\beta(\mathbf{x}, t)$ and $u(\mathbf{x}, t)$ (see Appendices C.2 and C.3 for proofs).

**Theorem 1** (Linear FiDE Pointwise convergence). *Let $y(\mathbf{x}, t)$ satisfy the LiFiDE (7) for each fixed $\mathbf{x} \in \mathcal{X}$ and assume: (1) for each $\mathbf{x} \in \mathcal{X}$, $\beta(\mathbf{x}, \cdot) \in C([0, \infty)) \cap L^1_{\text{loc}}[0, \infty)$, $\beta(\mathbf{x}, t) < 0$, $\forall (\mathbf{x}, t) \in \mathcal{X} \times [0, \infty)$ and $\exists \beta^*(\mathbf{x}) \in \mathbb{R}$ such that $\lim_{t \to \infty} \beta(\mathbf{x}, t) = \beta^*(\mathbf{x}) < 0$; (2) for each $\mathbf{x} \in \mathcal{X}$, $u(\mathbf{x}, \cdot) \in C([0, \infty))$ and $\lim_{t \to \infty} u(\mathbf{x}, t) = u^*(\mathbf{x}) \in \mathbb{R}$; (3) $\|u(\mathbf{x}, t)\|_{L^\infty(\mathcal{X} \times [0, \infty))} = M < \infty$ and $\|\beta(\mathbf{x}, t)\|_{L^\infty(\mathcal{X} \times [0, \infty))} = \lambda < \infty$. Then*

$$\lim_{t \to \infty} y(\mathbf{x}, t) = -u^*(\mathbf{x})/\beta^*(\mathbf{x}), \quad \forall \mathbf{x} \in \mathcal{X}. \tag{9}$$

The 'long-time' behavior of the solution $y(\mathbf{x}, t)$ is governed by the limiting values of the coefficients $\beta(\mathbf{x}, t)$ and $u(\mathbf{x}, t)$. As $t \to \infty$, the original non-autonomous equation (7) effectively behaves like the constant-coefficient equation

$$\dot{y} = \beta^*(\mathbf{x})\, y + u^*(\mathbf{x}), \tag{10}$$

which has the unique equilibrium point $y_\infty(\mathbf{x}) = -u^*(\mathbf{x})/\beta^*(\mathbf{x})$. Theorem 1 shows that the solution $y(\mathbf{x}, t)$ does indeed tend to this equilibrium as $t \to \infty$ for each fixed $\mathbf{x} \in \mathcal{X}$.

**Theorem 2** (Linear FiDE Uniform Convergence). *Let $y(\mathbf{x}, t)$ satisfy (7). Assume: (1) $y_0(\mathbf{x}) \in C(\mathcal{X})$; (2) $\beta(\mathbf{x}, t) \in C(\mathcal{X} \times [0, \infty))$, $\beta(\mathbf{x}, t) < 0$, $\forall (\mathbf{x}, t) \in \mathcal{X} \times [0, \infty)$, $\lim_{t \to \infty} \|\beta(\mathbf{x}, t) - \beta^*(\mathbf{x})\|_{L^\infty(\mathcal{X})} = 0$, $\beta^*(\mathbf{x}) \in C(\mathcal{X})$; (3) $u(\mathbf{x}, t) \in C(\mathcal{X} \times [0, \infty))$, $\lim_{t \to \infty} \|u(\mathbf{x}, t) - u^*(\mathbf{x})\|_{L^\infty(\mathcal{X})} = 0$, $u^*(\mathbf{x}) \in C(\mathcal{X})$; (4) $\exists \lambda' > 0$ such that $\beta^*(\mathbf{x}) \leq -\lambda'$, $\forall \mathbf{x} \in \mathcal{X}$. Then*

$$\lim_{t \to \infty} \|y(\mathbf{x}, t) + u^*(\mathbf{x})/\beta^*(\mathbf{x})\|_{L^\infty(\mathcal{X})} = 0. \tag{11}$$

**Remark 1** (Consistency of assumptions). *The conditions for convergence in Theorems 1 and 2 build naturally upon the conditions established in Lemma 5. In particular, the continuity assumptions $\beta(\mathbf{x}, \cdot), u(\mathbf{x}, \cdot) \in C([0, \infty))$ ensure the existence of a unique classical solution $y(\mathbf{x}, t) \in C^1$. The additional assumptions required for convergence (pointwise or uniform convergence of $\beta(\mathbf{x}, t) \to \beta^*(\mathbf{x})$ and $u(\mathbf{x}, t) \to u^*(\mathbf{x})$) serve to control the long-time asymptotics of the system. That is, while continuity ensures a solution exists, convergence of the coefficients guarantees that the solution stabilises to an equilibrium. Moreover, the boundedness conditions in Theorem 1 and uniformity in Theorem 2 (along with compactness of $\mathcal{X}$) allow us to lift the pointwise result to uniform convergence.*

Examples of coefficients that satisfy the assumptions in Theorems 1 and 2 are as follows.

*1. Exponential decay:* if $\beta(\mathbf{x}, t) = \beta^*(\mathbf{x}) + \delta_\beta(\mathbf{x})e^{-\mu t}$, $u(\mathbf{x}, t) = u^*(\mathbf{x}) + \delta_u(\mathbf{x})e^{-\mu t}$ with $\mu > 0$ and $\delta_\beta, \delta_u \in C(\mathcal{X})$, then uniform convergence holds.

*2. Polynomial decay:* slower convergence rates, e.g., with $\beta(\mathbf{x}, t) = \beta^*(\mathbf{x}) + \delta_\beta(\mathbf{x})t^{-\gamma}$, $\gamma > 1$, will suffice for pointwise convergence, since integrability over $[0, \infty)$ is preserved.

*3. Time-invariant case:* when $\beta(\mathbf{x}, t) \equiv \beta^*(\mathbf{x}) < 0$, $u(\mathbf{x}, t) \equiv u^*(\mathbf{x})$, the solution converges exponentially to the equilibrium $y_\infty(\mathbf{x}) = -u^*(\mathbf{x})/\beta^*(\mathbf{x})$.

*4. Bounded perturbations:* $\beta(\mathbf{x}, t) = \beta^*(\mathbf{x}) + \epsilon(\mathbf{x}, t)$ with bounded $\epsilon(\mathbf{x}, t)$, oscillatory or decaying. If $\epsilon(\mathbf{x}, t) \to 0$, the function converges to $\beta^*(\mathbf{x})$. If it does so uniformly, then Theorem 2 applies. If only pointwise convergence holds and $\epsilon(\mathbf{x}, \cdot) \in L^1([0, \infty))$, then Theorem 1 applies.

### 4.3 Tractable Autoregressive Gaussian Process Multi-Fidelity Surrogate

Based on the linear FiDE surrogate, we now introduce a tractable MF model that guarantees convergence at least pointwise while maintaining the probabilistic framework of GPs. Let $(\Omega, \mathcal{F}, \mathbb{P})$ be a probability space supporting two independent stochastic processes $y_0 : \mathcal{X} \times \Omega \to \mathbb{R}$ and $u : \mathcal{X} \times [0, \infty) \times \Omega \to \mathbb{R}$. We place Gaussian process priors over both

$$y_0(\mathbf{x}) \sim \mathcal{GP}(0, k_0(\mathbf{x}, \mathbf{x}')), \qquad u(\mathbf{x}, t) \sim \mathcal{GP}(0, k_u(\mathbf{x}, t, \mathbf{x}', t')). \tag{12}$$

The joint law of $(y_0, u)$ is the product measure induced by $\mathbb{P}$ on $\mathcal{Y}_0 \times \mathcal{U}$, reflecting their independence. Under these assumptions we have the following result (see Appendix D).

**Proposition 1.** *The linear model solution via the variation-of-constants formula*

$$y(\mathbf{x}, t) = e^{\int_0^t \beta(\mathbf{x}, \xi)\, d\xi} y_0(\mathbf{x}) + \int_0^t e^{\int_s^t \beta(\mathbf{x}, \xi)\, d\xi} u(\mathbf{x}, s)\, ds \tag{13}$$

*is a zero-mean Gaussian process with covariance function* $k(\mathbf{x}, t, \mathbf{x}', t') = \mathbb{E}_{\omega \sim \mathbb{P}}[y(\mathbf{x}, t; \omega)\, y(\mathbf{x}', t'; \omega)]$, *given explicitly by*

$$e^{\int_0^t \beta(\mathbf{x}, \xi)\, d\xi} e^{\int_0^{t'} \beta(\mathbf{x}', \xi)\, d\xi} k_0(\mathbf{x}, \mathbf{x}') + \int_0^t \int_0^{t'} e^{\int_s^t \beta(\mathbf{x}, \xi)\, d\xi} e^{\int_{s'}^{t'} \beta(\mathbf{x}', \xi)\, d\xi} k_u(\mathbf{x}, s, \mathbf{x}', s')\, ds\, ds'. \tag{14}$$

The integrals in the second term in Eq. (14) would in general require numerical quadrature. However, they can be evaluated analytically for various forms of $\beta(\mathbf{x}, t)$, e.g., a constant $\beta(\mathbf{x}, t) = -\beta$ with stationary kernels such as Matérn and periodic for $k_u(\mathbf{x}, t, \mathbf{x}, t')$ (see Appendix E). We also assume that $k_u(\mathbf{x}, t, \mathbf{x}, t') = k_\mathbf{x}(\mathbf{x}, \mathbf{x}')\, k_t(t, t')$, in which $k_t(t, t') = \exp\left(-(t - t')^2/2\ell^2\right)$, while $k_\mathbf{x}$ and $k_0(\mathbf{x}, \mathbf{x}')$ are kept arbitrary. This leads to the 'LiFiDE kernel'

$$k(\mathbf{x}, t, \mathbf{x}, t') = e^{-\beta t} e^{-\beta t'} k_0(\mathbf{x}, \mathbf{x}') + k_\mathbf{x}(\mathbf{x}, \mathbf{x}')\mathcal{I}(t, t'), \tag{15}$$

in which $\mathcal{I}(t, t') = (\sqrt{\pi}\ell/2)(h(t', t) + h(t, t'))$, where $h(t', t)$ is given by [26]

$$e^{\alpha^2} \left\{ e^{-\beta\ell\tau} \left( \mathrm{erf}\,(\tau - \alpha) + \mathrm{erf}\,(t/\ell + \alpha) \right) - e^{-\beta\ell\hat{\tau}} \left( \mathrm{erf}\,(t'/\ell - \alpha) + \mathrm{erf}\,(\alpha) \right) \right\} / (2\beta), \tag{16}$$

with $\mathrm{erf}(\cdot)$ denoting the error function, $\alpha = \beta\ell/\sqrt{2}$, $\tau = (t - t')/\ell$, and $\hat{\tau} = (t + t')/\ell$. The kernel (15) is non-stationary. Its structure allows us to reduce the training time complexity of the MF surrogate from $\mathcal{O}((\sum_f N_f)^3)$ to $\mathcal{O}(\sum_f N_f^3)$, where $N_f$ is the number of fidelity $f$ data points [19].

### 4.4 optimization Strategies Using Continuous Fidelity Acquisition Functions

There are several strategies for guiding the selection of the input $\mathbf{x}$ and fidelity $t$ at each iteration. MF-GP-UCB [11] uses fidelity-specific upper confidence bounds and selects query inputs by maximising the minimum of a UCB acquisition across discrete fidelities. BOCA [3] extends this method to continuous $t$, with the fidelity selected from a filtered set. cfKG [4] extends the KG acquisition function to continuous fidelities by selecting the point $(\mathbf{x}, t)$ that maximises the expected reduction in the high-fidelity posterior minimum. FABOLAS [16] is based on a cost-aware EI acquisition function, in which the fidelity is typically a proxy. It models the validation loss and cost jointly as GPs and selects $(\mathbf{x}, t)$ values that maximise improvement per cost. MF-DNN [18] approximates mutual information between the function optimum and observations using a variational approximation, enabling efficient acquisition in large neural network tuning problems.

While some acquisition functions can be re-used across MF models, many (e.g., BOCA and cfKG) are tightly coupled to specific fidelity models. In our experiments, BOCA consistently delivered the best empirical performance among the methods above. It also provides theoretical guarantees alongside practical performance, and an improved rate of convergence to the optimum compared to UCB. The original formulation considers the fidelity set $\mathcal{T} = [0, 1]$ and input space $\mathcal{X} = [0, 1]^d$, with $y(\mathbf{x}, t) \sim \mathcal{GP}(0, \kappa)$, $\kappa = \kappa_t(t, t')\kappa_{\mathbf{x}}(\mathbf{x}, \mathbf{x}')$. The key idea in BOCA is to exploit cheaper, approximate fidelities $t < 1$ when they are sufficiently informative about the target function $f(\mathbf{x}) = y(\mathbf{x}, 1)$. BOCA filters the fidelity set using a cost constraint and a minimum uncertainty condition. By concentrating expensive evaluations in a small, polynomially-dilated variant $\mathcal{X}_{\rho,n}$ of a 'high-information' region $\mathcal{X}_\rho$, BOCA achieves tighter regret bounds than GP-UCB and faster convergence. Additionally, BOCA spends the majority of its evaluations on low-cost but informative fidelities, leading to improved capital ($\Lambda$) efficiency. The region $\mathcal{X}_\rho$ depends on a 'fidelity gap' $\xi(t) = \sqrt{1 - \kappa_t(t, 1)^2}$, which controls the tightness of the bound. The following is an informal version of the simple regret $r$ result in terms of the mutual information $\gamma_n$ (see Appendix F for full details).

**Theorem 3** (Kandasamy et al. [3]). *Let* $\mathcal{X} = [0, 1]^d$, $\mathcal{T} = [0, 1]$ *and* $y(\mathbf{x}, t) \sim \mathcal{GP}(0, \kappa)$, $\kappa = \kappa_t(t, t')\kappa_{\mathbf{x}}(\mathbf{x}, \mathbf{x}')$. *Choose* $\delta \in (0, 1)$ *and execute BOCA with* $\beta_n = \mathcal{O}(d \log(n/\delta))$. *Then, for any* $\alpha \in (0, 1)$, *there exist* $\rho(\alpha) > 0$ *such that for* $\Lambda$ *large enough, with probability at least* $1 - \delta$

$$r(\Lambda) \lesssim \sqrt{\gamma_{n_\Lambda}(\mathcal{X}_\rho) n_\Lambda^{-1}} + \sqrt{\gamma_{n_\Lambda^\alpha}(\mathcal{X}) n_\Lambda^{\alpha-2}}, \quad n_\Lambda = \lfloor \Lambda/\lambda(1) \rfloor. \quad (17)$$

For a fixed $\mathbf{x}, \mathbf{x}'$, the standard SE kernel, $k_{\mathrm{SE}}$ enforces high correlation near the diagonal $t \approx t'$, with $\mathcal{O}(|t - t'|^2)$ decay that is independent of $t$. This implies all fidelities are equally smooth and informative. In practical scenarios, however, low-fidelity evaluations are often noisy or unstable. In contrast, the LiFiDE kernel variance $k(t, t)$ increases with $t$, and $k(t, t')$ has $\mathcal{O}(|t - t'|)$ decay along the diagonal $t \approx t'$, decreasing with $t$ (see Appendix G). This models a convergent fidelity process: evaluations become more stable and informative as fidelity increases, reflecting a more realistic structure. For the SE kernel, $\xi(t) = \mathcal{O}(|t - T|)$ for $|t - T| \ll 1$ for a maximum fidelity $T$. In contrast, for the LiFiDE kernel $\xi(t) = \mathcal{O}(\sqrt{|t - T|})$ (see Remark G1). Although this leads to looser regret guarantees via $\mathcal{X}_\rho$, the LiFiDE kernel offers stronger empirical performance due to its convergence-aware structure and because it does not concentrate as sharply at $t \to T$, discarding potentially useful results at low fidelity. Indeed, the experimental results will show that CAMO conducts most of its exploration in low-fidelity regions, which reduces the query cost and leads to a fast convergence rate in the early stages of the optimization process, without sacrificing accuracy.

## 5 Experimental Results

We assess CAMO on synthetic benchmarks, including continuous and discrete MFBO tasks, as well as real-world engineering design tasks. We compare the results to those of: (1) BOCA with a standard GP [3], (2) Fabolas [16], and (3) SMAC3 [27]. On discrete fidelity tasks, we compare the results to: (1) AR [28], (2) ResGP [29], and (3) a GP [6], and discrete MFBO baselines (1) MF-UCB [30], (2) MF-EI [10], and (3) cfKG [17].

**Settings.** We assess CAMO with the LiFiDE kernel Eq. (15). Except for DNN-MFBO, Fabolas, and SMAC3 (original implementations and default settings), all methods were implemented in Pytorch. All GP models used the SE Kernel for a fair comparison. Each model is updated for 200 steps using an Adam optimizer with a learning rate of $0.01$ to ensure model convergence. In each case, 10 low-fidelity and 4 high-fidelity designs were randomly selected to form the initial training set. We repeated the experiments 20 times with random seeds and report the mean values. Figures showing the actual optimization progress are provided in Appendix H. The optimization performance was measured by the simple regret ($\gamma$), defined as the difference between the global optimum and the best-queried design so far: $\gamma_i = \max f(\mathbf{x}, T) - \max_{j<i} f(\mathbf{x}_j, T)$. All experiments were performed on a workstation with an AMD 7800x CPU, Nvidia RTX4080 GPU, and 32GB RAM.

### 5.1 Synthetic Benchmark Evaluation

We consider: (1) three canonical continuous-fidelity tasks [31], the Park, Currin and Branin functions; and (2) three further synthetic continuous-fidelity tasks [32], the nonlinear sin, Forrester, and Bohachevsky functions. In the latter 3 we set $f(\mathbf{x}, t) = (1 - w(t))f_{\mathrm{low}}(\mathbf{x}) + w(t)f_{\mathrm{high}}(\mathbf{x})$ with $w(t) = \ln(9t + 1)$. All functions are defined in Appendix I. The query costs were set to $c(t) = 10^t$.

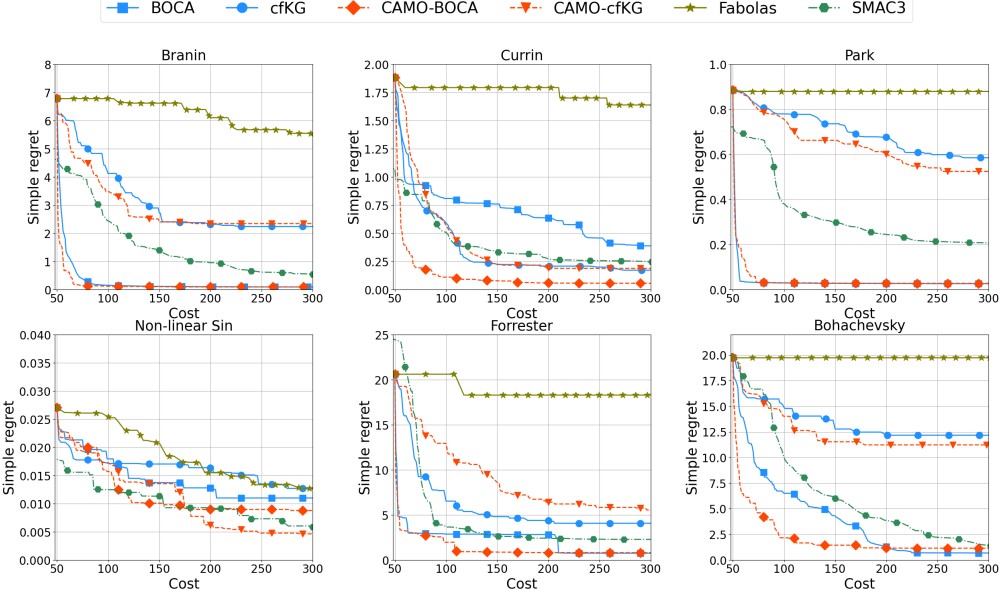

Figure 1: Simple regret vs cost for all continuous fidelity functions functions.

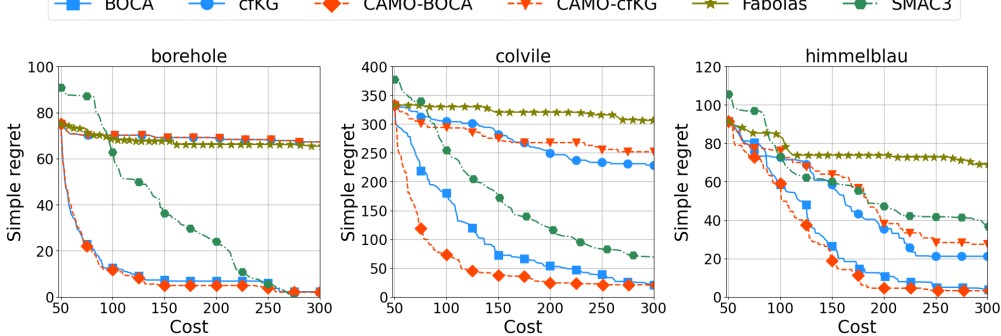

Figure 2: Simple regret for the Borehole, Colville and Himmelblau functions.

We show $\gamma$ on the six tasks under increasing query cost in Fig. 1, which clearly demonstrates the superiority of CAMO across all tasks, with the exception of the non-linear sin at low cost. BOCA also performs well on the Branin and Park functions. The results for different random seeds are shown in Fig. H1–Fig. H9 in Appendix H to demonstrate the consistency of CAMO. These figures also show that methods equipped with cfKG (including CAMO) are generally less competitive in terms of $\gamma$ and convergence speed compared to those with BOCA. To investigate the scalability of the results in terms of $d$, we further evaluated CAMO on the Borehole, Colville [33], and Himmelblau [34] functions (6, 4, and 2 design variables, respectively). The results are shown in Fig. 2. In these cases the superiority of CAMO is just as pronounced, suggesting robustness as $d$ increases.

One of the key factors in MFBO is the cost $c(t)$ of querying the high-fidelity function. We conducted experiments with different costs: $c(t) = 10^t$, $c(t) = 5t$, and $c(t) = \log_2(2 + t)$ for the Currin and Bohachevsky functions. The results are shown in Fig. 3, and for the Forrester function in Fig. H10. Clearly, CAMO outperforms all other methods under all cost settings. An exponential cost setting is more challenging for all methods. The advantage of CAMO is increasingly significant from $c(t) = \log_2(2 + t) \to 5t \to 10^t$, which highlights the advantage of being convergence-aware in MFBO by maximising the benefit/cost ratio. For a logarithmic cost setting, CAMO performs similarly to BOCA because being "convergence-aware" is less "rewarded" when the cost also increases logarithmically. In contrast, if the cost increases beyond linear, the advantage of CAMO becomes significant. Such a setting is common in real-world applications, e.g., FEM (cubic complexity with mesh resolution [35]) and Monte Carlo estimation (quadratic sample requirements [36]). Importantly, CAMO always exhibits a fast convergence rate in the early stages of the optimization since it does not discard useful low-fidelity information. This is consistent with the theoretical analysis.

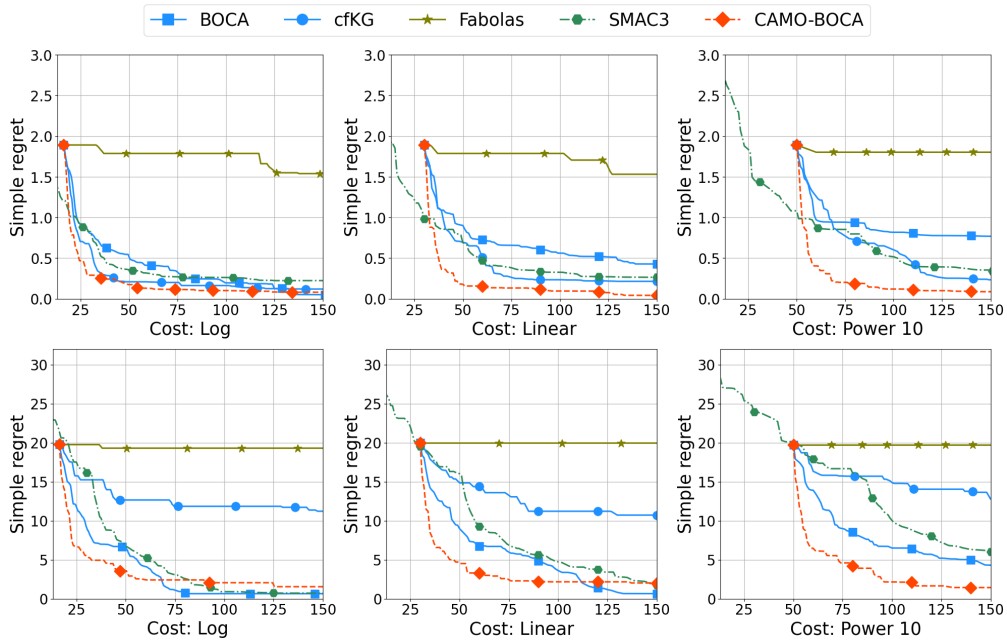

Figure 3: Continuous MFBO for the Currin (top row) and Bohachevsky (bottom row) functions using logarithmic ($\log_2(2 + t)$), linear ($5t$), and exponential ($10^t$) cost $c(t)$.

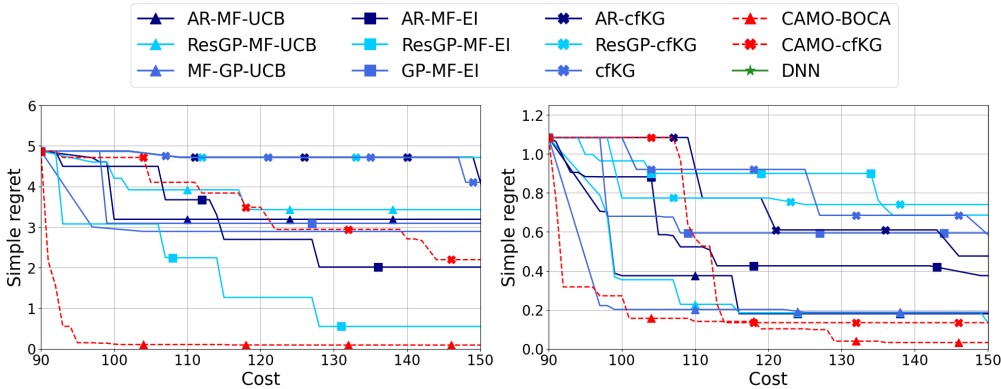

Figure 4: Discrete MFBO on the Branin (left) and Currin functions (right).

**Discrete MFBO Assessment.** To examine performance on discrete MFBO tasks, we discretise the Branin and Currin into ten discrete fidelities. We also compare with the DNN-MFBO method [18]. The results under increasing query costs are shown in Fig. 4. We can see that CAMO again outperforms all other methods by a wide margin on the Branin function, while the advantages are clear but less dramatic on the Currin function. DNN-MFBO essentially failed.

For MFBO, the time for model update and acquisition-function based optimization is also crucial. The average times (over all the benchmarks, iterations, and seeds) of the optimization queries and model training are shown in Fig. 5 for different query and MF approaches, respectively. Despite the favourable performance reported in Li et al. [18], MutualInfo and FABOLAS are impractically slow in terms of query time. SMAC3 has the shortest simulation time, but it is not competitive in terms of accuracy and convergence rate. CAMO is intermediate between these methods for both query time using BOCA and training, achieving a good trade-off between performance and computational cost.

## 5.2 Real-World Applications in Engineering Design

We now consider real-world continuous tasks (see Appendix J). **(1) Mechanical Plate Vibration.** We optimize the natural vibration frequency of a 3-D, supported, square elastic plate ($10 \times 10 \times 1$ in m) over the Young's modulus ($\in [100, 500]$[GPa]), Poisson ratio ($\in [0.2, 0.6]$) and mass density ($\in [6 \times 10^3, 9 \times 10^3]$[kgm$^{-3}$]). This is a parametric FE modal analysis. The maximum element size

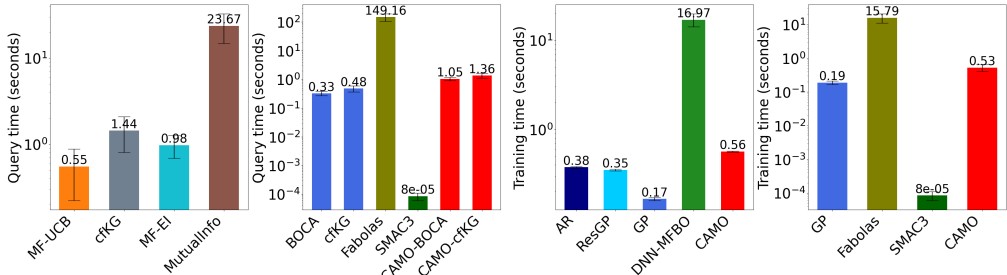

Figure 5: From left to right: average query time on discretised tasks, average query time on continuous tasks, average training on discretised tasks, and average training time on continuous tasks.

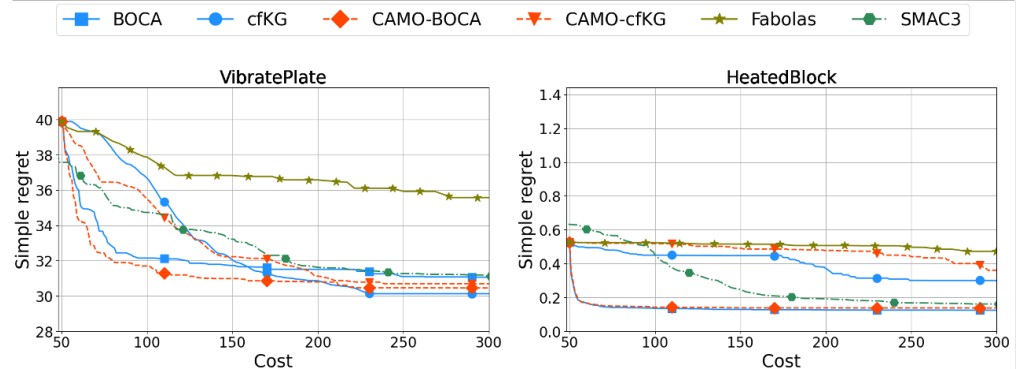

Figure 6: MFBO on the Mechanical Plate Vibration (left) and Thermal Conductor Design (right) problems with different query cost.

$h_{max} \in [0.2, 1.2][m]$ is the fidelity. **(2) Thermal Conductor.** We optimize the shape of an elliptical central hole in which a conductor is placed in order to maximise a heat conduction rate. The hole shape is parameterised by the semi-major and semi-minor axes and orientation angle. We used the time to reach 70 degrees as the objective function value. $h_{max} \in [0.1, 2][m]$ is the fidelity.

The $\gamma$ versus simulation cost is shown in Fig. 6. The cost scales according to the inverse cubic of the maximal element size, which is the standard computational complexity for FEM problems. CAMO and BOCA outperform the other methods in both tasks by a significant margin. In particular, the fast convergence of CAMO carries over to these real-world problems, with BOCA yielding comparable performance on the Thermal Conductor and a slightly worse performance than CAMO on the Mechanical Plate Vibration. In terms of wall clock time to a given $\gamma$, CAMO and BOCA are very similar, with FABOLAS again impractically slow.

## 6 Conclusions

We propose CAMO, a convergence-aware MFBO framework based on LiFiDEs for continuous fidelity problems. CAMO captures the fidelity-wise evolution of the objective function and provides a theoretically-grounded surrogate model. For general FiDE we used a Lyapunov-based analysis to establish convergence guarantees even for non-smooth objectives. Combined with BOCA, CAMO delivers strong empirical performance, consistently outperforming state-of-the-art MFBO methods on synthetic and real-world tasks. Crucially, CAMO adapts to and exploits informative low-fidelity queries rather than discarding them, enabling more efficient use of the evaluation budget and fast convergence.

CAMO has several limitations that are worth noting. Computationally, the kernel optimization adds roughly 20% overhead compared to standard GP methods, and the approach requires sufficient low-fidelity data to properly model convergence behaviour. Methodologically, the LiFiDE assumption may not capture all (especially non-monotonic) convergence patterns, and performance depends on the existence of a good cost model. CAMO is suited to scenarios in which the computational budget is the primary constraint and early progress is valuable. For unlimited budget scenarios, more aggressive high-fidelity exploration might be preferable.

## Funding Disclosure

The authors declare no external funding.

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

# Appendices

## A  Gaussian process

Consider observations $y_i = f(\mathbf{x}_i) + \varepsilon$, $i = 1, \ldots, N$, in which $\varepsilon \sim \mathcal{N}(0, \sigma^2)$ is additive noise. In a GP model, a prior distribution is placed over $f(\mathbf{x})$

$$\mathbf{f}(\mathbf{x})|\boldsymbol{\theta} \sim \mathcal{GP}\left(m_0(\mathbf{x}), k(\mathbf{x}, \mathbf{x}'|\boldsymbol{\theta})\right), \tag{A1}$$

in which $m_0(\mathbf{x}) = \mathbb{E}[f(\mathbf{x})]$ is the mean function and $k(\mathbf{x}, \mathbf{x}'|\boldsymbol{\theta}) = \mathbb{E}[(f(\mathbf{x}) - m_0(\mathbf{x}))(f(\mathbf{x}') - m_0(\mathbf{x}'))]$ is the covariance function. A set of hyperparameters $\boldsymbol{\theta}$ fully characterises the kernel function and in most cases the data is centred (the empirical mean is subtracted) to justify setting $m_0(\mathbf{x}) \equiv 0$ as a simplification. Various forms can be adopted for the covariance function, with the following exponential ARD kernel being the most widely favoured

$$k(\mathbf{x}, \mathbf{x}'|\boldsymbol{\theta}) = \theta_0 \exp\left(-(\mathbf{x} - \mathbf{x}')^\top \mathrm{diag}(\theta_1^{-2}, \ldots, \theta_l^{-2})(\mathbf{x} - \mathbf{x}')\right). \tag{A2}$$

By the key property of GPs, the joint distribution of $f(x_i)$, $i = 1, \ldots, N$, is a multivariate Gaussian. This leads to the following conditional predictive posterior conditioned on $\mathbf{y} = (y_1, \ldots, y_N)^\top$ and $\mathbf{x}$

$$\hat{f}(\mathbf{x})|\mathbf{y} \sim \mathcal{N}\left(\mu(\mathbf{x}), v(\mathbf{x}, \mathbf{x}')\right),$$
$$\mu(\mathbf{x}) = \mathbf{k}(\mathbf{x})^\top \left(\mathbf{K} + \sigma^2\mathbf{I}\right)^{-1}\mathbf{y}, \tag{A3}$$
$$v(\mathbf{x}) = \sigma^2 + k(\mathbf{x}, \mathbf{x}) - \mathbf{k}^\top(\mathbf{x})\left(\mathbf{K} + \sigma^2\mathbf{I}\right)^{-1}\mathbf{k}(\mathbf{x}).$$

in which $\mathbf{K} = [K_{ij}]$, $K_{ij} = k(\mathbf{x}_i, \mathbf{x}_j)$, $i, j = 1, \ldots, N$, is the covariance matrix, and $\mathbf{k} = (k(\mathbf{x}_i, \mathbf{x}, \ldots, k(\mathbf{x}_N, \mathbf{x}))$. The hyperparameters are typically inferred from the likelihood, given by

$$\mathcal{L} = -\frac{1}{2}\mathbf{y}^\top(\mathbf{K} + \sigma^2\mathbf{I})^{-1}\mathbf{y} - \frac{1}{2}\ln|\mathbf{K} + \sigma^2\mathbf{I}| - \frac{N}{2}\log(2\pi). \tag{A4}$$

## B  Well-posedness of general FiDEs

### B.1  Existence and uniqueness

Here we consider the well-posedness of solutions to the general Fidelity Differential Equation (FiDE)

$$\partial_t y(\mathbf{x}, t) = \phi(\mathbf{x}, t, y(\mathbf{x}, t)), \quad (\mathbf{x}, t) \in \mathcal{X} \times [0, \infty), \tag{B1}$$

in which $\mathcal{X}$ is any nonempty set, subject to some initial condition $y(\mathbf{x}, 0) = y_0(\mathbf{x})$.

**Lemma 1.** *[Existence and Regularity for General FiDEs] Let $\mathcal{X} \subseteq \mathbb{R}^d$ be a non-empty set. Consider the system (B1) and assume that:*

1. *for each fixed $\mathbf{x} \in \mathcal{X}$, the function $(t, y) \mapsto \phi(\mathbf{x}, t, y)$ is continuous on $[0, \infty) \times \mathbb{R}$*

2. *for each fixed $\mathbf{x} \in \mathcal{X}$, $\phi(\mathbf{x}, t, y)$ is locally Lipschitz continuous in $y$, uniformly for $t$ in compact subsets of $[0, \infty)$*

3. *the initial condition $y_0(\mathbf{x}) \in \mathbb{R}$ is finite for each $\mathbf{x}$.*

*Then for each fixed $\mathbf{x} \in \mathcal{X}$, there exists a unique maximal solution $y(\mathbf{x}, t) \in C^1([0, t_*(\mathbf{x})))$ defined on some maximal interval $[0, t_*(\mathbf{x}))$, with $0 < t_*(\mathbf{x}) \leq \infty$. Moreover:*

1. *if $y(\mathbf{x}, t)$ remains bounded for all $t \in [0, \infty)$, then $t_*(\mathbf{x}) = \infty$ and the solution exists globally on $[0, \infty)$*

2. *if $\phi$ is globally Lipschitz in $y$, uniformly in $t$, then the solution exists globally on $[0, \infty)$.*

*Proof.* Fix $\mathbf{x} \in \mathcal{X}$, then the equation becomes an ordinary differential equation (ODE) in the variable $t$

$$\dot{y} = \phi(\mathbf{x}, t, y), \quad y(0) = y_0(\mathbf{x}). \tag{B2}$$

Since by assumption (1), the function $(t, y) \mapsto \phi(\mathbf{x}, t, y)$ is continuous, and by assumption (2), $\phi(\mathbf{x}, t, y)$ is locally Lipschitz in $y$, it follows from the Picard–Lindelöf Theorem that there exists a local time $t_*(\mathbf{x}) > 0$ and a unique solution $t \mapsto y(\mathbf{x}, t) \in C^1([0, t_*(\mathbf{x})))$ that solves the ODE for $t \in [0, t_*(\mathbf{x}))$. Uniqueness of the solution $y(\mathbf{x}, t)$ follows directly from the local Lipschitz property of $\phi$ in $y$, which prevents branching of solutions and ensures that the solution is unique once $y_0(\mathbf{x})$ is fixed.

By the general theory of ODEs (continuation theorems) [37], the maximal existence time $t_*(\mathbf{x})$ satisfies

$$t_*(\mathbf{x}) = \infty \quad \text{or} \quad \limsup_{t \to t_*(\mathbf{x})^-} |y(\mathbf{x}, t)| = \infty, \tag{B3}$$

the latter representing blow-up of the solution in a finite time $t_*$. Thus, if we can guarantee that $y(\mathbf{x}, t)$ remains bounded for all $t \in [0, \infty)$, then it follows that $t_*(\mathbf{x}) = \infty$, and the solution extends globally. If $\phi$ is globally Lipschitz in $y$ uniformly in $t$, then no blow-up can occur, and the solution is guaranteed to exist globally. Since $\phi$ is continuous in $(t, y)$ by assumption (1), and $y(\mathbf{x}, t)$ solves the ODE, standard regularity theory tells us that $y(\mathbf{x}, \cdot) \in C^1([0, t_*(\mathbf{x})))$. Specifically, $\partial_t y(\mathbf{x}, t) = \phi(\mathbf{x}, t, y(\mathbf{x}, t))$ is continuous, as the composition of continuous functions. $\qquad\square$

## B.2 Pointwise and Uniform Convergence for FiDE

We now consider the convergence of solutions to (B1). The following results provide a practical set of conditions under which $y(\mathbf{x}, t) \to y_\infty(\mathbf{x})$ pointwise or uniform in $\mathbf{x}$.

**Lemma 2** (Pointwise Convergence for FiDE). *Let $y(\mathbf{x}, \cdot) \in C^1([0, \infty))$, $\mathbf{x} \in \mathcal{X}$ be a solution to (1) with $y_0(\mathbf{x}) < \infty$ for each $\mathbf{x} \in \mathcal{X}$. Suppose there exists a Lyapunov function $V : \mathcal{X} \times [0, \infty) \times \mathbb{R} \to \mathbb{R}$, $(\mathbf{x}, t, y) \mapsto V(\mathbf{x}, t, y)$, and that the following conditions are satisfied:*

1. *$V \in C^{0,1,1}(\mathcal{X} \times [0, \infty) \times \mathbb{R})$*

2. *$V(\mathbf{x}, t, y) \geq 0$, $\forall (\mathbf{x}, t, y) \in \mathcal{X} \times [0, \infty) \times \mathbb{R}$ and $V(\mathbf{x}, t, y) = 0$ iff $y = y_\infty(\mathbf{x})$*

3. *for all $(\mathbf{x}, t) \in \mathcal{X} \times [0, \infty)$*

$$\dot{V}(\mathbf{x}, t, y) = \partial_y V(\mathbf{x}, t, y) \phi(\mathbf{x}, t, y) + \partial_t V(\mathbf{x}, t, y) < 0 \tag{B4}$$

*Then $y(\mathbf{x}, t) \to y_\infty(\mathbf{x})$ pointwise on $\mathcal{X}$, i.e., $\lim_{t\to\infty} y(\mathbf{x}, t) = y_\infty(\mathbf{x})$ for each $\mathbf{x} \in \mathcal{X}$.*

*Proof.* For each $t$, define the function $y_t : \mathcal{X} \to \mathbb{R}$, $\mathbf{x} \mapsto y(\mathbf{x}, t)$. Under the assumptions, we will show that the family of functions $(y_t)_{t \geq 0}$ converges pointwise to $y_\infty$ on $\mathcal{X}$, that is

$$\lim_{t\to\infty} y(\mathbf{x}, t) = y_\infty(\mathbf{x}), \quad \forall \mathbf{x} \in \mathcal{X}. \tag{B5}$$

Fix $\mathbf{x} \in \mathcal{X}$ and consider the function $t \mapsto V(\mathbf{x}, t, y(\mathbf{x}, t))$. By the chain rule, and using the assumptions, we have

$$\begin{aligned}
\dot{V}(\mathbf{x}, t, y) &= \partial_y V(\mathbf{x}, t, y) \, \partial_t y(\mathbf{x}, t) + \partial_t V(\mathbf{x}, t, y) \\
&= \partial_y V(\mathbf{x}, t, y) \, \phi(\mathbf{x}, t, y) + \partial_t V(\mathbf{x}, t, y),
\end{aligned} \tag{B6}$$

using $\partial_t y(\mathbf{x}, t) = \phi(\mathbf{x}, t, y)$. By assumption, $\frac{d}{dt} V(\mathbf{x}, t, y)$ is strictly negative, so that $V(\mathbf{x}, t, y)$ is strictly decreasing along the trajectory for each fixed $\mathbf{x}$. Furthermore, $V(\mathbf{x}, t, y) > 0$ for all $t \geq 0$ by assumption, and $V(\mathbf{x}, t, y)$ is bounded below by 0. Now $t \mapsto V(\mathbf{x}, y(\mathbf{x}, t), t)$ is a continuous-time function and so for a fixed $\mathbf{x}$, $(V(\mathbf{x}, y(\mathbf{x}, t), t))_{t \in [0, \infty)} \subset \mathbb{R}$ defines a bounded, monotone net on the directed set $([0, \infty), \leq)$, where $\leq$ is the usual order relation. Therefore, this net converges to its infimum

$$\exists \lim_{t\to\infty} V((\mathbf{x}, t, y(\mathbf{x}, t))) = \inf_{t \in [0, \infty)} V((\mathbf{x}, t, y(\mathbf{x}, t))) = L(\mathbf{x}) \geq 0. \tag{B7}$$

Since $V$ is strictly decreasing in $t$ and positive, its limit must be 0 (otherwise it would eventually fall below $L(\mathbf{x})$, a contradiction) and thus $L(\mathbf{x}) = 0$, i.e.

$$\lim_{t\to\infty} V(\mathbf{x}, t, y) = 0. \tag{B8}$$

Finally, $V(\mathbf{x}, t, y) = 0$ if and only if $y = y_\infty(\mathbf{x})$, and thus

$$\lim_{t\to\infty} y(\mathbf{x}, t) = y_\infty(\mathbf{x}). \tag{B9}$$

Since $\mathbf{x} \in \mathcal{X}$ was arbitrary, we obtain pointwise convergence on $\mathcal{X}$. $\qquad\square$

**Lemma 3** (Uniform Convergence for FiDE). *Let $y(\mathbf{x}, \cdot) \in C^1([0, \infty))$, $\mathbf{x} \in \mathcal{X}$ is a solution to (1). Suppose there exists a Lyapunov function $V : \mathcal{X} \times [0, \infty) \times \mathbb{R} \to \mathbb{R}$, $(\mathbf{x}, t, y) \mapsto V(\mathbf{x}, t, y)$ and the following conditions are satisfied:*

1. $y_0(\mathbf{x}) \in C(\mathcal{X})$

2. $V \in C^{0,1,2}(\mathcal{X} \times [0, \infty) \times \mathbb{R})$

3. $V(\mathbf{x}, t, y) \geq 0, \forall (\mathbf{x}, t, y) \in \mathcal{X} \times [0, \infty) \times \mathbb{R}$ and $V(\mathbf{x}, t, y) = 0$ iff $y = y_\infty(\mathbf{x})$

4. $\exists c_{\min} > 0$ such that $c(\mathbf{x}, t) \geq c_{\min} > 0, \forall (\mathbf{x}, t) \in \mathcal{X} \times [0, \infty)$, in which $c(\mathbf{x}, t) = \frac{1}{2}\partial_{yy}^2 V(\mathbf{x}, t, y_\infty(\mathbf{x}))$

5. for all $\mathbf{x}, t \in \mathcal{X} \times [0, \infty)$, $\exists \alpha > 0$ such that

$$\dot{V}(\mathbf{x}, t, y) = \partial_y V(\mathbf{x}, t, y)\phi(\mathbf{x}, t, y) + \partial_t V(\mathbf{x}, t, y) \leq -\alpha V(\mathbf{x}, t, y) \tag{B10}$$

6. $\exists V_{\max}$ such that $\|V(\mathbf{x}, 0, y(\mathbf{x}, 0))\|_{L^\infty(\mathcal{X})} \leq V_{\max}$

Then $\lim_{t \to \infty} y(\mathbf{x}, t) = y_\infty(\mathbf{x})$ uniformly on $\mathcal{X}$, with

$$\|\mathbf{y}(\mathbf{x}, t) - y_\infty(\mathbf{x})\|_{L^\infty(\mathcal{X})} \leq ce^{-\alpha t}, \quad c = \frac{\sqrt{V_{\max}}}{c_{\min}}. \tag{B11}$$

*Proof.* From assumption 5 and Grönwall's inequality [37], we have for all $(\mathbf{x}, t) \in \mathcal{X} \times [0, \infty)$

$$V(\mathbf{x}, y(\mathbf{x}, t), t) \leq V(\mathbf{x}, y_0(\mathbf{x}), 0)e^{-\alpha t}. \tag{B12}$$

By the smoothness assumption $V \in C^{0,1,2}(\mathcal{X} \times [0, \infty) \times \mathbb{R})$, a Taylor expansion around $y = y_\infty(\mathbf{x})$ yields, for each $(\mathbf{x}, t)$

$$V(\mathbf{x}, y(\mathbf{x}, t), t) \geq c(\mathbf{x}, t)|y(\mathbf{x}, t) - y_\infty(\mathbf{x})|^2, \tag{B13}$$

in which $c(\mathbf{x}, t) = \frac{1}{2}\partial_{yy}^2 V(\mathbf{x}, y_\infty(\mathbf{x}), t)$, and by assumption 4, $c(\mathbf{x}, t) \geq c_{\min} > 0$ uniformly.

Thus

$$|y(\mathbf{x}, t) - y_\infty(\mathbf{x})| \leq \frac{1}{c(\mathbf{x}, t)}\sqrt{V(\mathbf{x}, y(\mathbf{x}, t), t)} \leq \frac{1}{c_{\min}}\sqrt{V(\mathbf{x}, y_0(\mathbf{x}), 0)}e^{-\alpha t/2}. \tag{B14}$$

Taking the supremum over $\mathcal{X}$ yields

$$\|y(\mathbf{x}, t) - y_\infty(\mathbf{x})\|_{L^\infty(\mathcal{X})} \leq \frac{\sqrt{V_{\max}}}{c_{\min}}e^{-\alpha t/2}, \tag{B15}$$

which proves uniform convergence.

$\square$

## B.3 Non-smooth Lyapunov functions

**Lemma B1.** *[Convergence for non-smooth convex Lyapunov function] Let $\mathcal{X}$ be a nonempty set and let $(\mathbf{x}, t)$ satisfy (B1). Define $V(\mathbf{x}, y, t) = |y - y_\infty(\mathbf{x})|$ for $(\mathbf{x}, y, t) \in \mathcal{X} \times \mathbb{R} \times [0, \infty)$. Suppose that for each $\mathbf{x} \in \mathcal{X}$:*

1. *The map $t \mapsto y(\mathbf{x}, t)$ is $C^1([0, \infty))$*

2. *Let $\partial_y V(\mathbf{x}, y, t)$ be the subdifferential mapping of $y \mapsto V(\mathbf{x}, y, t)$. For every $(\mathbf{x}, t) \in \mathcal{X} \times [0, \infty)$ and every $\xi(\mathbf{x}, t) \in \partial_y V(\mathbf{x}, y(\mathbf{x}, t), t)$, we have*

$$\xi(\mathbf{x}, t)\,\phi(\mathbf{x}, t, y(\mathbf{x}, t)) \leq -\alpha V(\mathbf{x}, y(\mathbf{x}, t), t) \tag{B16}$$

*for some constant $\alpha > 0$ independent of $\mathbf{x}$ and $t$.*

*Then for each $\mathbf{x} \in \mathcal{X}$ and every $t \in [0, \infty)$,*

$$|y(\mathbf{x}, t) - y_\infty(\mathbf{x})| \leq |y(\mathbf{x}, 0) - y_\infty(\mathbf{x})|e^{-\alpha t}. \tag{B17}$$

*If, moreover*

3 $\|y(\mathbf{x}, 0) - y_\infty(\mathbf{x})\|_{L^\infty(\mathcal{X})} = \ell < \infty$

*the convergence is uniform in $\mathbf{x} \in \mathcal{X}$*

$$\|y(\cdot, t) - y_\infty(\cdot)\|_{L^\infty(\mathcal{X})} \leq \ell e^{-\alpha t}, \quad \forall t \in [0, \infty). \tag{B18}$$

*Proof.* For each fixed $(\mathbf{x}, t) \in \mathcal{X} \times [0, \infty)$, the map

$$u[\mathbf{x}, t](\cdot) : \mathbb{R} \to \mathbb{R}, \quad y \mapsto V(\mathbf{x}, y, t) \tag{B19}$$

is convex and 1-Lipschitz continuous in $y$. For a fixed $\mathbf{x} \in \mathcal{X}$, we also define the function

$$v[\mathbf{x}](\cdot) : [0, \infty) \to \mathbb{R}, \quad t \mapsto V(\mathbf{x}, y(\mathbf{x}, t), t) \tag{B20}$$

$y(\mathbf{x}, \cdot) \in AC([0, \infty))$ (absolutely continuous) since it is in $C^1([0, \infty))$, and thus the composite map $v[\mathbf{x}](\cdot)$ is absolutely continuous on $[0, \infty)$. By the properties of absolutely continuous functions, $v[\mathbf{x}](\cdot)$ is differentiable almost everywhere, and its upper right Dini derivative

$$D^+ v[\mathbf{x}](t) = \limsup_{h \to 0^+} \frac{v[\mathbf{x}](t+h) - v[\mathbf{x}](t)}{h} \tag{B21}$$

exists for every $t \in [0, \infty)$.

The subdifferential at a point $y_*$ of the convex function $u[\mathbf{x}, t](\cdot)$ is defined as the set of subgradients $\partial u[\mathbf{x}, t](y_*) = \{g \in \mathbb{R} : u[\mathbf{x}, t](y) \geq u[\mathbf{x}, t](y_*) + g(y - y_*), \ \forall y \in \mathbb{R}\}$ [24, 38]. In the present case, the set-valued subdifferential mapping $y \mapsto \partial u[\mathbf{x}, t](y), \mathbb{R} \to 2^{\mathbb{R}}$ is given by

$$\partial u[\mathbf{x}, t](y) = \partial |y(\mathbf{x}, t) - y_\infty(\mathbf{x})| = \begin{cases} \{+1\}, & y(\mathbf{x}, t) > y_\infty(\mathbf{x}), \\ \{-1\}, & y(\mathbf{x}, t) < y_\infty(\mathbf{x}), \\ [-1, 1], & y(\mathbf{x}, t) = y_\infty(\mathbf{x}). \end{cases} \tag{B22}$$

Applying the chain rule to $u[\mathbf{x}, t](\cdot)$ and using the definition of the subdifferential mapping of a convex function we obtain

$$\dot{v}[\mathbf{x}](t) \in \partial_y u[\mathbf{x}, t](y) \partial_t y(\mathbf{x}, t) = \partial_y u[\mathbf{x}, t](y) \, \phi(\mathbf{x}, t, y(\mathbf{x}, t)), \quad \text{a.e. } t \in [0, \infty) \tag{B23}$$

by using the definition of $y$. Given assumption (2), for any subgradient $\xi(\mathbf{x}, t) \in \partial u[\mathbf{x}, t](y) \in \mathbb{R}$, we have

$$\xi(\mathbf{x}, t) \, \phi(\mathbf{x}, t, y(\mathbf{x}, t)) \leq -\alpha v[\mathbf{x}](t), \tag{B24}$$

which holds for $t \in [0, \infty)$ a.e and pointwise in $\mathbf{x} \in \mathcal{X}$, with $y = y(\mathbf{x}, t)$ fixed. Thus, for $t \in [0, \infty)$ a.e.

$$\dot{v}[\mathbf{x}](t) \leq -\alpha v[\mathbf{x}](t). \tag{B25}$$

Since $v[\mathbf{x}](t) \in AC([0, \infty))$, by the standard properties of Dini derivatives, this inequality extends to the upper Dini derivative, yielding

$$D^+ v[\mathbf{x}](t) \leq -\alpha v[\mathbf{x}](t), \quad \forall t \in [0, \infty). \tag{B26}$$

Applying the generalised Grönwall inequality for upper Dini derivatives (which states that if $D^+ u(t) \leq -\alpha u(t)$, then $u(t) \leq u(0)e^{-\alpha t}$), we obtain

$$V(\mathbf{x}, y(\mathbf{x}, t), t) \leq V(\mathbf{x}, y(\mathbf{x}, 0), 0)e^{-\alpha t}. \tag{B27}$$

Recalling that $V(\mathbf{x}, y, t) = |y - y_\infty(\mathbf{x})|$, this yields

$$|y(\mathbf{x}, t) - y_\infty(\mathbf{x})| \leq |y(\mathbf{x}, 0) - y_\infty(\mathbf{x})|e^{-\alpha t}, \quad \forall t \in [0, \infty), \tag{B28}$$

pointwise $\mathbf{x} \in \mathcal{X}$. By assumption (3) this holds in the $L^\infty(\mathcal{X})$ norm by taking the sup on both sides, i.e.

$$\|y(\cdot, t) - y_\infty(\cdot)\|_{L^\infty(\mathcal{X})} \leq \ell e^{-\alpha t}, \quad \forall t \in [0, \infty). \tag{B29}$$

$\square$

This result can be extended to non-convex Lyapunov functions by using the Clarke subdifferential [24]. We first introduce some definitions. Let $X$ be a Banach space with dual $X^*$. The weak $^*$ topology on $X^*$ is the coarsest topology such that all evaluation maps $f \mapsto f(x)$ for $x \in X$ are continuous. That is, $f_\alpha \to f$ in the weak$^*$ topology iff $f_\alpha(x) \to f(x)$ for all $x \in X$. Now let $f : \mathbb{R}^n \to \mathbb{R}$ be locally Lipschitz. The Clarke subdifferential of $f$ at $x$ is defined as

$$\partial_C f(x) = \mathrm{co} \left\{ \lim_{k \to \infty} \nabla f(x_k) : x_k \to x, \ f \text{ differentiable at } x_k \right\}, \tag{B30}$$

in which co denotes the convex hull of a set. This set is always nonempty, convex, and closed. If $f = g \circ h$, where $g : \mathbb{R} \to \mathbb{R}$ and $h : \mathbb{R} \to \mathbb{R}^n$ are locally Lipschitz, then for any $t \in \mathbb{R}$ ([24], Chain Rule Theorem 2.3.9)

$$\partial_C f(t) \subset \overline{\mathrm{co}}^{w^*} \{\alpha \zeta : \alpha \in \partial_C g(h(t)), \ \zeta \in \partial_C h(t)\}, \tag{B31}$$

in which $\overline{\mathrm{co}}^{w^*} \{\cdot\}$ denotes the closure of the convex hull in the weak$^*$ topology on $\mathbb{R}^* \cong \mathbb{R}$. In finite dimensions, notably in our case $\mathbb{R}$, this is simply the norm-closed convex hull and the standard-norm, weak, and weak$^*$ topologies are equivalent.

If we set $h(t) = y(\mathbf{x}, t)$ and $g(y) = u[\mathbf{x}](y)$ (we assume $V$ has no explicit dependence on $t$), then $f(t) = v[\mathbf{x}](t)$ as previously defined. Since $y(\mathbf{x}, \cdot) \in C^1([0, \infty))$, $\partial_C y(t) = \{\partial_t y(\mathbf{x}, t)\}$, so that

$$\partial_C v[\mathbf{x}](t) \subset \overline{\mathrm{co}} \left\{ \alpha \partial_t y(\mathbf{x}, t) : \alpha \in \partial_C u[\mathbf{x}](y) \right\}. \tag{B32}$$

Now, the image of a convex set under a linear map is convex, and therefore (B32) reduces to

$$\partial_C v[\mathbf{x}](t) \subset \partial_t y(\mathbf{x}, t) \partial_C u[\mathbf{x}](y), \tag{B33}$$

since $\overline{\mathrm{co}}^{w^*} \equiv \overline{\mathrm{co}}$ and furthermore the Clarke subdifferential is already closed and convex.

We know that the composition $v[\mathbf{x}](t) = V(\mathbf{x}, y(\mathbf{x}, t), t) \in AC([0, \infty))$ because $V$ is locally Lipschitz in $y$ and $v$ is differentiable. Then by Clarke's Chain Rule Theorem above

$$\dot{v}[\mathbf{x}](t) \in \partial_t y(\mathbf{x}, t) \partial_C u[\mathbf{x}](y) \quad \text{a.e. } t \in [0, \infty). \tag{B34}$$

If we assume that

$$\xi(\mathbf{x}, t)\phi(\mathbf{x}, t, y(\mathbf{x}, t)) \leq -\alpha v[\mathbf{x}](t), \quad \forall \xi(\mathbf{x}, t) \in \partial_C u[\mathbf{x}](y), \tag{B35}$$

then

$$D^+ v[\mathbf{x}](t) \leq -\alpha v[\mathbf{x}](t) \quad \forall t \in [0, \infty). \tag{B36}$$

We can then use Grönwall's inequality and take the supremum over $\mathbf{x} \in \mathcal{X}$ to obtain a uniform bound. This proves the following lemma.

**Lemma 4.** *Let $\mathcal{X}$ be a nonempty set, and let $y : \mathcal{X} \times [0, \infty) \to \mathbb{R}$ be such that for each $\mathbf{x} \in \mathcal{X}$, $y(\mathbf{x}, \cdot) \in C^1[0, \infty)$. Let $V : \mathcal{X} \times \mathbb{R} \to \mathbb{R}$, $(\mathbf{x}, y) \to V(\mathbf{x}, y(\mathbf{x}, t))$ satisfy the following conditions*

1. *For every fixed $(\mathbf{x}, t)$, the map $y \mapsto V(\mathbf{x}, y, t)$ is locally Lipschitz.*

2. *For each fixed $(\mathbf{x}, t) \in \mathcal{X} \times [0, \infty)$, define $u[\mathbf{x}](\cdot) : \mathbb{R} \to \mathbb{R}$, $y \mapsto V(\mathbf{x}, y)$, and for a fixed $\mathbf{x} \in \mathcal{X}$, define $v[\mathbf{x}](\cdot) : [0, \infty) \to \mathbb{R}$, $t \mapsto V(\mathbf{x}, y(\mathbf{x}, t))$. Assume that for all $(\mathbf{x}, t) \in \mathcal{X} \times [0, \infty)$ and all $\xi(\mathbf{x}, t) \in \partial_C u[\mathbf{x}](y)$ the inequality*

$$\xi(\mathbf{x}, t)\, \phi(\mathbf{x}, t, y(\mathbf{x}, t)) \leq -\alpha v[\mathbf{x}](t) \tag{B37}$$

*holds for some constant $\alpha > 0$, independent of $\mathbf{x}$ and $t$.*

3. *The initial deviation is bounded: $\|y(\cdot, 0) - y_\infty(\cdot)\|_{L^\infty(\mathcal{X})} < \infty$.*

*Then, for all $t \in [0, \infty)$, we have*

$$\|y(\cdot, t) - y_\infty(\cdot)\|_{L^\infty(\mathcal{X})} \leq \|y(\cdot, 0) - y_\infty(\cdot)\|_{L^\infty(\mathcal{X})} e^{-\alpha t}. \tag{B38}$$

## B.4 Connection to Subgradient Flows and Proximal Methods

We have analysed the convergence of fidelity-indexed systems $y : \mathcal{X} \times [0, \infty) \to \mathbb{R}$, which serve as multi-fidelity surrogate models for a high-fidelity target function $y_\infty : \mathcal{X} \to \mathbb{R}$. These surrogates evolve over fidelity index $t$ according to the FiDE dynamics (1). When a non-smooth Lyapunov function such as $V(\mathbf{x}, y, t) = |y - y_\infty(\mathbf{x})|$ is used to study convergence, the time evolution must be interpreted using subdifferential calculus. In particular, the upper Dini derivative satisfies the inclusion

$$D^+ V(\mathbf{x}, y(\mathbf{x}, t), t) \in \xi(\mathbf{x}, t)\, \phi(\mathbf{x}, t, y(\mathbf{x}, t)) + \frac{\partial V}{\partial t}(\mathbf{x}, y(\mathbf{x}, t), t), \tag{B39}$$

in which $\xi(\mathbf{x}, t) \in \partial_y V(\mathbf{x}, y(\mathbf{x}, t), t)$ is a subgradient. For example, when $V(\mathbf{x}, y, t) = |y - y_\infty(\mathbf{x})|$, $\partial_y V(\mathbf{x}, y, t)$ is given by (B22). This structure defines a differential inclusion, and is directly analogous to subgradient flows commonly used in convex optimization, particularly for non-smooth regularisation problems such as LASSO. In these cases, the evolution of a state $x(t)$ under a non-smooth objective $g$ is given by

$$\dot{x}(t) \in -\partial g(x(t)). \tag{B40}$$

For the case in Lemma B1, the surrogate $y(\mathbf{x}, t)$ evolves by descent on the 'fidelity error' $g(y) = |y - y_\infty(\mathbf{x})|$, driving it asymptotically towards the equilibrium $y_\infty(\mathbf{x})$. Under mild monotonicity assumptions on $\phi$, this descent yields exponential convergence, analogous to the behavior of proximal-point iterations in optimization.

This framework extends naturally to more general non-smooth Lyapunov functions, such as the elastic-net form

$$V(\mathbf{x}, y, t) = \lambda_1 |y - y_\infty(\mathbf{x})| + \lambda_2 (y - y_\infty(\mathbf{x}))^2, \tag{B41}$$

or group sparsity objectives for vector-valued surrogates $y : \mathcal{X} \to \mathbb{R}^d$

$$V(\mathbf{x}, y, t) = \sum_{g \in \mathcal{G}} \lambda_g \|y_g(\mathbf{x}, t) - y_{\infty, g}(\mathbf{x})\|_2, \tag{B42}$$

in which $\mathcal{G}$ is a partitioning of the coordinates. In all such cases, the surrogate $y(\mathbf{x}, t)$ evolves via a subdifferential inclusion that mirrors proximal-map dynamics in optimization, linking multi-fidelity modelling with tools from modern sparse learning and non-smooth convex analysis.

The subgradient dynamics discussed above can also be viewed as a limiting case of smooth gradient flows applied to a smoothed surrogate objective. In convex analysis, the Moreau envelope provides a classical smoothing of a non-smooth function. For a convex, lower semi-continuous function $f : \mathbb{R} \to \mathbb{R}$, the envelope is defined as [25]

$$f^\lambda(x) = \inf_{z \in \mathbb{R}} \left\{ f(z) + \frac{1}{2\lambda} |x - z|^2 \right\}, \tag{B43}$$

with $\nabla f^\lambda(x) = \frac{1}{\lambda}(x - \mathrm{prox}_{\lambda f}(x))$. This provides a differentiable approximation of $f$, and as $\lambda \to 0$, we have $f^\lambda(y) \to f(y)$ pointwise, and $\nabla f^\lambda(y) \to \partial f(y)$ in the sense of graphical (or Mosco) convergence.

While the actual FiDE dynamics are not given by a gradient flow, the Lyapunov convergence analysis admits a natural interpretation via these smoothed flows. For instance, using the fidelity error function $f(y) = |y - y_\infty(\mathbf{x})|$, we define the Moreau envelope

$$V_\lambda(\mathbf{x}, y) = \inf_{z \in \mathbb{R}} \left\{ |z - y_\infty(\mathbf{x})| + \frac{1}{2\lambda}(y - z)^2 \right\}. \tag{B44}$$

This yields the auxiliary gradient flow

$$\partial_t y_\lambda(\mathbf{x}, t) = -\nabla_y V_\lambda(\mathbf{x}, y_\lambda(\mathbf{x}, t)), \tag{B45}$$

which approximates the subgradient dynamics

$$\partial_t y(\mathbf{x}, t) \in -\partial |y(\mathbf{x}, t) - y_\infty(\mathbf{x})| \tag{B46}$$

as $\lambda \to 0$. While this flow is not the governing equation for $y(\mathbf{x}, t)$ in the FiDE, it serves as a useful analytical proxy, capturing the key descent structure and convergence behaviour of the surrogate. From this point of view, the Lyapunov function represents a fidelity error whose decay over $t$ encodes asymptotic convergence to the high-fidelity limit.

# C  Well-Posedness and Convergence of Linear FiDE: Proofs

We consider the linear FiDE

$$\partial_t y(\mathbf{x}, t) = \beta(\mathbf{x}, t) y(\mathbf{x}, t) + u(\mathbf{x}, t), \quad (\mathbf{x}, t) \in \mathcal{X} \times [0, \infty), \tag{C1}$$

subject to an initial condition $y_0(\mathbf{x}) = y(\mathbf{x}, 0)$.

## C.1  Proof of Lemma 5

*Proof.* Let $\mathcal{X} \subset \mathbb{R}^d$ be non-empty and fix $\mathbf{x} \in \mathcal{X}$. Suppose that:

(A1) $\beta(\mathbf{x}, \cdot), u(\mathbf{x}, \cdot) \in C([0, \infty))$

(A2) $y_0(\mathbf{x}) \in \mathbb{R}$

Define $\phi(\mathbf{x}, t, y) = \beta(\mathbf{x}, t)\, y + u(\mathbf{x}, t)$. For fixed $\mathbf{x}$, the map $\phi(\mathbf{x}, \cdot, \cdot)$ is continuous in $(t, y)$ since both $\beta(\mathbf{x}, t)$ and $u(\mathbf{x}, t)$ are continuous in $t$. Moreover, $\phi$ is globally Lipschitz in $y$ since

$$|\phi(\mathbf{x}, t, y_1) - \phi(\mathbf{x}, t, y_2)| = |\beta(\mathbf{x}, t)|\, |y_1 - y_2|. \tag{C2}$$

Thus, $|\beta(\mathbf{x}, t)|$ acts as a Lipschitz constant (depending only on $t$). By the Picard–Lindelöf Theorem, there exists a unique local solution $y(\mathbf{x}, t) \in C^1$ defined on some interval $[0, t_*(\mathbf{x}))$.

Now suppose additionally that: (A3) $\|\beta(\mathbf{x}, t)\|_{L^\infty([0,\infty))} < \infty$ and $\|u(\mathbf{x}, t)\|_{L^\infty([0,\infty))} < \infty$. Then the right-hand side of the ODE

$$\phi(\mathbf{x}, t, y(\mathbf{x}, t)) = \beta(\mathbf{x}, t) y(\mathbf{x}, t) + u(\mathbf{x}, t) \tag{C3}$$

remains bounded on compact intervals of $[0, \infty)$. Therefore, the solution $y(\mathbf{x}, t)$ cannot blow up in finite time, and the local solution can be extended globally to $[0, \infty)$, i.e., $t_*(\mathbf{x}) = \infty$, and $y(\mathbf{x}, t) \in C^1([0, \infty))$.

Define the integrating factor $\mu(\mathbf{x}, t) = e^{\int_0^t \beta(\mathbf{x}, s)\, ds}$. Then

$$\frac{d}{dt}(\mu(\mathbf{x}, t) y(\mathbf{x}, t)) = u(\mathbf{x}, t)\, \mu(\mathbf{x}, t), \tag{C4}$$

and integrating both sides yields the variation of constants formula

$$y(\mathbf{x}, t) = e^{\int_0^t \beta(\mathbf{x},s)\,ds} y_0(\mathbf{x}) + \int_0^t e^{\int_s^t \beta(\mathbf{x},z)\,dz} u(\mathbf{x}, s)\,ds. \tag{C5}$$

Now Let $\mathcal{X} \subset \mathbb{R}^d$ be compact, and suppose that:

(B1) $\beta(\mathbf{x}, t), u(\mathbf{x}, t) \in C(\mathcal{X} \times [0, \infty))$

(B2) $y_0(\mathbf{x}) \in C(\mathcal{X})$

in addition to (A3). Then for each $(\mathbf{x}, t) \in \mathcal{X} \times [0, \infty)$, the solution is given by the variation-of-constants formula

$$y(\mathbf{x}, t) = e^{\int_0^t \beta(\mathbf{x},s)\,ds} y_0(\mathbf{x}) + \int_0^t e^{\int_s^t \beta(\mathbf{x},z)\,dz} u(\mathbf{x}, s)\,ds. \tag{C6}$$

We now show that $y(\mathbf{x}, t) \in C(\mathcal{X} \times [0, \infty))$. Define

$$F_1(\mathbf{x}, t) = e^{\int_0^t \beta(\mathbf{x},s)\,ds} y_0(\mathbf{x}). \tag{C7}$$

Since $\beta(\mathbf{x}, s) \in C(\mathcal{X} \times [0, \infty))$, the map

$$(\mathbf{x}, t) \mapsto \int_0^t \beta(\mathbf{x}, s)\,ds \tag{C8}$$

is continuous. The exponential function is continuous, and $y_0(\mathbf{x}) \in C(\mathcal{X})$, so the product

$$(\mathbf{x}, t) \mapsto e^{\int_0^t \beta(\mathbf{x},s)\,ds} y_0(\mathbf{x}) \tag{C9}$$

is continuous on $\mathcal{X} \times [0, \infty)$.

Now define

$$F_2(\mathbf{x}, t) = \int_0^t e^{\int_s^t \beta(\mathbf{x},z)\,dz} u(\mathbf{x}, s)\,ds. \tag{C10}$$

and denote the integrand by

$$f(\mathbf{x}, t, s) = e^{\int_s^t \beta(\mathbf{x},z)\,dz} u(\mathbf{x}, s). \tag{C11}$$

Since $\beta(\mathbf{x}, z) \in C(\mathcal{X} \times [0, \infty))$, the map $(\mathbf{x}, t, s) \mapsto \int_s^t \beta(\mathbf{x}, z)\,dz$ is continuous. Since $u(\mathbf{x}, s) \in C(\mathcal{X} \times [0, \infty))$, it follows that $f(\mathbf{x}, t, s) \in C(\mathcal{X} \times [0, \infty)^2)$. Due to the compactness of $\mathcal{X}$ and $\beta, u \in L^\infty$, the integrand is uniformly bounded. Therefore, the parameter-dependent integral

$$F_2(\mathbf{x}, t) = \int_0^t f(\mathbf{x}, t, s)\,ds \tag{C12}$$

is continuous in $(\mathbf{x}, t)$. Since both terms $F_1(\mathbf{x}, t)$ and $F_2(\mathbf{x}, t)$ are continuous, it follows that

$$y(\mathbf{x}, t) = F_1(\mathbf{x}, t) + F_2(\mathbf{x}, t) \in C(\mathcal{X} \times [0, \infty)). \tag{C13}$$

$\square$

## C.2 Proof of Theorem 1

Assume the following:

1. For each $\mathbf{x} \in \mathcal{X}$, we have $\beta(\mathbf{x}, \cdot) \in C([0, \infty)) \cap L_{\text{loc}}^1([0, \infty))$, $\beta(\mathbf{x}, t) < 0$, $\forall (\mathbf{x}, t) \in \mathcal{X} \times [0, \infty)$ and $\lim_{t \to \infty} \beta(\mathbf{x}, t) = \beta^*(\mathbf{x}) < 0$

2. For each $\mathbf{x} \in \mathcal{X}$, $u(\mathbf{x}, \cdot) \in C([0, \infty))$, and $\lim_{t \to \infty} u(\mathbf{x}, t) = u^*(\mathbf{x}) \in \mathbb{R}$

3. $\|u(\mathbf{x}, t)\|_{L^\infty(\mathcal{X} \times [0, \infty))} = M < \infty$ and $\|\beta(\mathbf{x}, t)\|_{L^\infty(\mathcal{X} \times [0, \infty))} = \lambda < \infty$

To prove Theorem 1 we will require the following Proposition.

**Proposition C1.** *For each fixed $\mathbf{x} \in \mathcal{X}$ and each $r \geq 0$, the following holds*

$$\int_{t-r}^t \beta(\mathbf{x}, u)\,du \to r\beta^*(\mathbf{x}), \quad t \to \infty. \tag{C14}$$

*Proof.* Since $\beta(\mathbf{x}, t) \to \beta^*(\mathbf{x})$ pointwise in $\mathbf{x} \in \mathcal{X}$, for any $\delta > 0$, $\exists t_0 > 0$ such that

$$|\beta(\mathbf{x}, t) - \beta^*(\mathbf{x})| < \delta, \quad \forall t \geq t_0. \tag{C15}$$

For a fixed $r > 0$, let $\delta = \varepsilon/r$ and define $t_* = t_0 + r$. Then for all $t \geq t_*$, we have $t - r \geq t_0$, and so for all $u \in [t - r, t]$, with $t \geq t_*$

$$|\beta(\mathbf{x}, u) - \beta^*(\mathbf{x})| < \frac{\varepsilon}{r}. \tag{C16}$$

Therefore, for all $t \geq t_*$

$$\left| \int_{t-r}^{t} \beta(\mathbf{x}, u)\, du - r\beta^*(\mathbf{x}) \right| = \left| \int_{t-r}^{t} (\beta(\mathbf{x}, u) - \beta^*(\mathbf{x}))\, du \right|$$
$$\leq \int_{t-r}^{t} |\beta(\mathbf{x}, u) - \beta^*(\mathbf{x})|\, du \tag{C17}$$
$$< \varepsilon$$

From this we conclude that for each $\mathbf{x} \in \mathcal{X}$ and each $r \geq 0$

$$\int_{t-r}^{t} \beta(\mathbf{x}, u)\, du \to r\beta^*(\mathbf{x}). \tag{C18}$$

$\square$

*Proof of Theorem 1.* Since we are concerned with pointwise convergence we need only employ the standard topology on $\mathbb{R}$. Under the assumptions above, the variation-of-constants formula for a fixed $\mathbf{x} \in \mathcal{X}$ yields the solution

$$y(\mathbf{x}, t) = e^{\int_0^t \beta(\mathbf{x}, s)\, ds} y_0(\mathbf{x}) + \int_0^t e^{\int_s^t \beta(\mathbf{x}, u)\, du} u(\mathbf{x}, s)\, ds. \tag{C19}$$

Since $\beta(\mathbf{x}, \cdot) \in C([0, \infty)) \subset L^1_{\text{loc}}([0, \infty))$, the integrals are well-defined. Moreover, since $\beta(\mathbf{x}, t) < 0$ and $\beta(\mathbf{x}, t) \to \beta^*(\mathbf{x}) < 0$, we have

$$\int_0^t \beta(\mathbf{x}, s)\, ds \to -\infty, \tag{C20}$$

so that

$$\lim_{t \to \infty} e^{\int_0^t \beta(\mathbf{x}, s)\, ds} y_0(\mathbf{x}) = 0. \tag{C21}$$

Let us now define the inhomogeneous integral

$$I(\mathbf{x}, t) = \int_0^t e^{\int_s^t \beta(\mathbf{x}, u)\, du} u(\mathbf{x}, s)\, ds. \tag{C22}$$

Using the change of variable $r = t - s$, we obtain

$$I(\mathbf{x}, t) = \int_0^t e^{\int_{t-r}^t \beta(\mathbf{x}, u)\, du} u(\mathbf{x}, t - r)\, dr. \tag{C23}$$

If we set

$$f_t(r) = \begin{cases} e^{\int_{t-r}^t \beta(\mathbf{x}, u)\, du} u(\mathbf{x}, t - r) & 0 \leq r \leq t \\ 0 & r > t \end{cases}, \tag{C24}$$

then $(f_t(r))_{t \in [0, T)}$ defines a net and

$$I(\mathbf{x}, t) = \int_0^\infty f_t(r)\, dr. \tag{C25}$$

Now define the following limiting function

$$f(r) = e^{r\beta^*(\mathbf{x})} u^*(\mathbf{x}). \tag{C26}$$

For each fixed $r \geq 0$, $u(\mathbf{x}, t - r) \to u^*(\mathbf{x})$. Moreover, from Proposition C1, for each $\mathbf{x} \in \mathcal{X}$ and each $r \geq 0$

$$\int_{t-r}^t \beta(\mathbf{x}, u)\, du \to r\beta^*(\mathbf{x}), \tag{C27}$$

and therefore $f_t(r) \to f(r)$ pointwise in $r$.

Since the Dominated Convergence Theorem applies only to sequences, we now extract a sequence $(f_{t_n})_{n \in \mathbb{N}}$ such that $t_n \to \infty$ and consider the corresponding sequence of functions $(f_{t_n})$. This sequence satisfies

1. $f_{t_n}(r) \to f(r)$ for each $r \geq 0$ by the assumption on $u$ and Proposition C1

2. $|f_{t_n}(r)| \leq Me^{r\lambda} = g(r)$, in which $g \in L^1([0, \infty))$ (since $\lambda < 0$) is a dominating function

Therefore, by the Dominated Convergence Theorem, we have

$$\lim_{n\to\infty} \int_0^\infty f_{t_n}(r)\, dr = \int_0^\infty f(r)\, dr = u^*(\mathbf{x}) \int_0^\infty e^{r\beta^*(\mathbf{x})} dr = -\frac{u^*(\mathbf{x})}{\beta^*(\mathbf{x})}. \tag{C28}$$

Since the limit of the integral is independent of the particular sequence $(t_n)$, and because $\mathbb{R}$ is first-countable (i.e., sequential), it follows that the net $\left(\int_0^\infty f_t(r)\, dr\right)_{t\in[0,\infty)}$ converges to the same value

$$\lim_{t\to\infty} I(\mathbf{x}, t) = -\frac{u^*(\mathbf{x})}{\beta^*(\mathbf{x})}. \tag{C29}$$

This yields the final result

$$\lim_{t\to\infty} y(\mathbf{x}, t) = -\frac{u^*(\mathbf{x})}{\beta^*(\mathbf{x})}. \tag{C30}$$

$\square$

## C.3 Proof of Theorem 2

We again consider the linear FiDE problem (C1), now with $\mathcal{X} \subset \mathbb{R}^d$ a compact set (closed and bounded). We make the following assumptions:

1. $y_0(\mathbf{x}) \in C(\mathcal{X})$;
2. $\beta(\mathbf{x}, t) \in C(\mathcal{X} \times [0, \infty))$, with

$$\lim_{t\to\infty} \|\beta(\cdot, t) - \beta^*(\cdot)\|_{L^\infty(\mathcal{X})} = 0, \quad \beta^*(\mathbf{x}) \in C(\mathcal{X}) \tag{C31}$$

3. $u(\mathbf{x}, t) \in C(\mathcal{X} \times [0, \infty))$, with

$$\lim_{t\to\infty} \|u(\cdot, t) - u^*(\cdot)\|_{L^\infty(\mathcal{X})} = 0, \quad u^*(\mathbf{x}) \in C(\mathcal{X}) \tag{C32}$$

4. There exists $\lambda' > 0$ such that $\beta^*(\mathbf{x}) \leq -\lambda'$ for all $\mathbf{x} \in \mathcal{X}$.

We will require the following two Propositions in order to prove Theorem 2.

**Proposition C2.** *There exist finite $\lambda > 0$ and $M > 0$ such that*

$$\beta(\mathbf{x}, t) \leq -\lambda, \ \forall\, (\mathbf{x}, t) \in \mathcal{X} \times [0, \infty) \quad \text{and} \quad \|u(\mathbf{x}, t)\|_{L^\infty(\mathcal{X} \times [0, \infty)])} \leq M. \tag{C33}$$

*Proof.* By compactness of $\mathcal{X}$, $\beta^* \in C(\mathcal{X})$, and by assumption (4), we have

$$\beta^*(\mathbf{x}) \leq -\lambda' < 0 \quad \forall \mathbf{x} \in \mathcal{X}. \tag{C34}$$

Furthermore, since $\beta(\cdot, t) \to \beta^*(\cdot)$ uniformly in $L^\infty(\mathcal{X})$, there exists $T > 0$ such that for all $t \geq T$

$$\|\beta(\cdot, t) - \beta^*(\cdot)\|_{L^\infty(\mathcal{X})} < \lambda'/2. \tag{C35}$$

Hence, for all $\mathbf{x} \in \mathcal{X}$, $t \geq T$

$$\beta(\mathbf{x}, t) \leq \beta^*(\mathbf{x}) + \lambda'/2 \leq -\lambda' + \lambda'/2 = -\lambda'/2. \tag{C36}$$

On the compact set $\mathcal{X} \times [0, T]$, $\beta$ is continuous, hence bounded above. Let

$$\lambda = \min\left\{\lambda'/2, \ -\sup_{\mathcal{X} \times [0, T]} \beta(\mathbf{x}, t)\right\} > 0. \tag{C37}$$

noting that $\beta(\mathbf{x}, t) < 0, \forall (\mathbf{x}, t) \in \mathcal{X} \times [0, \infty)$, otherwise we can use $\lambda = \min\left\{\lambda'/2, \ \inf_{\mathcal{X} \times [0, T]}(-\beta(\mathbf{x}, t))\right\}$. Then for all $\mathbf{x}, t$, $\beta(\mathbf{x}, t) \leq -\lambda$.

Similarly, uniform convergence of $u(\cdot, t) \to u^*(\cdot) \in C(\mathcal{X})$ implies there exists $M > 0$ such that

$$|u(\mathbf{x}, t)| \leq M \quad \forall\, (\mathbf{x}, t) \in \mathcal{X} \times [0, \infty). \tag{C38}$$

$\square$

**Proposition C3.** *For each fixed $s \geq 0$, the following*

$$\int_s^t \beta(\mathbf{x}, u)\, du = \beta^*(\mathbf{x})(t - s) + o(t) \tag{C39}$$

*holds uniformly for $\mathbf{x} \in \mathcal{X}$ as $t \to \infty$.*

*Proof.* By the uniform convergence of $\beta$, for every $\epsilon > 0$, there exists $U_*(\epsilon) > 0$ such that for all $u \geq U_*$

$$\|\beta(\mathbf{x}, u) - \beta^*(\mathbf{x})\|_{L^\infty(\mathcal{X})} < \epsilon. \tag{C40}$$

Let $t \geq U_*$, then for any fixed $s \geq 0$

$$\int_s^t \beta(\mathbf{x}, u)\, du = \int_s^{U_*} \beta(\mathbf{x}, u)\, du + \int_{U_*}^t \beta(\mathbf{x}, u)\, du. \tag{C41}$$

The first integral is over a finite interval, and is thus uniformly bounded in $\mathbf{x}$. For the second term, observe that $\beta(\mathbf{x}, u) = \beta^*(\mathbf{x}) + \delta(\mathbf{x}, u)$, where $|\delta(\mathbf{x}, u)| < \epsilon$ for all $u \geq U_*$. Thus

$$\int_{U_*}^t \beta(\mathbf{x}, u)\, du = \beta^*(\mathbf{x})(t - U_*) + \int_{U_*}^t \delta(\mathbf{x}, u)\, du, \tag{C42}$$

with

$$\left| \int_{U_*}^t \delta(\mathbf{x}, u)\, du \right| \leq \epsilon(t - U_*). \tag{C43}$$

Therefore, since $s$ and $U_*$ are fixed

$$\int_s^t \beta(\mathbf{x}, u)\, du = \beta^*(\mathbf{x})(t - s) + \beta^*(\mathbf{x})(s - U_*) + \int_s^{U_*} \beta(\mathbf{x}, u)\, du + O(\epsilon t). \tag{C44}$$

Thus

$$\frac{1}{t} \left| \int_s^t \beta(\mathbf{x}, u)\, du - \beta^*(\mathbf{x})(t - s) \right| \leq \frac{c}{t} + \epsilon + \epsilon \frac{U_*}{t}, \tag{C45}$$

in which $c > 0$ is a constant independent of $t$ and $\mathbf{x}$. As $t \to \infty$, the right-hand side tends to $\epsilon$, and since $\epsilon > 0$ was arbitrary, (C39) follows, uniformly in $\mathbf{x}$ for each fixed $s \geq 0$. $\qquad\square$

*Proof of Theorem 2.* By the variation of constants formula, the solution is given by

$$y(\mathbf{x}, t) = e^{\int_0^t \beta(\mathbf{x}, s)\, ds} y(\mathbf{x}, 0) + \int_0^t e^{\int_s^t \beta(\mathbf{x}, u)\, du} u(\mathbf{x}, s)\, ds. \tag{C46}$$

By assumption (4), and Proposition C2, we have $\beta(\mathbf{x}, t) \leq -\lambda < 0$ for all $\mathbf{x}, t$, and so

$$\int_0^t \beta(\mathbf{x}, s)\, ds \leq -\lambda t \quad \Rightarrow \quad e^{\int_0^t \beta(\mathbf{x}, s)\, ds} \leq e^{-\lambda t}. \tag{C47}$$

Furthermore, $y_0(\mathbf{x}) \in C(\mathcal{X})$ with compact $\mathcal{X}$, so that $\|y_0(\mathbf{x})\|_{L^\infty(\mathcal{X})} = K < \infty$. Thus,

$$\lim_{t \to \infty} \sup_{\mathbf{x} \in \mathcal{X}} \left| e^{\int_0^t \beta(\mathbf{x}, s)\, ds} y_0(\mathbf{x}) \right| \leq \lim_{t \to \infty} K e^{-\lambda t} = 0, \tag{C48}$$

and therefore the first term in (C46) vanishes uniformly in $\mathbf{x}$.

Now let

$$I(\mathbf{x}, t) = \int_0^t f_t(\mathbf{x}, s)\, ds, \quad f_t(\mathbf{x}, s) = e^{\int_s^t \beta(\mathbf{x}, u)\, du} u(\mathbf{x}, s). \tag{C49}$$

From Proposition C3

$$\int_s^t \beta(\mathbf{x}, u)\, du = \beta^*(\mathbf{x})(t - s) + o(t), \tag{C50}$$

uniformly in $\mathbf{x}$ for each fixed $s \geq 0$, and since $\beta^*(\mathbf{x}) < 0$ we obtain

$$e^{\int_s^t \beta(\mathbf{x}, u)\, du} \sim e^{\beta^*(\mathbf{x})(t - s)} \to 0, \tag{C51}$$

also uniformly in $\mathbf{x}$ for each fixed $s \geq 0$. When analysing the behavior of $f_t(\mathbf{x}, s)$ for all $s \in [0, t]$, $s$ and $t$ may diverge as $t \to \infty$. In order to handle the behavior across the entire range $s \in [0, t]$, we use change of variable $r = t - s$, so that

$$I(\mathbf{x}, t) = \int_0^t e^{\int_s^t \beta(\mathbf{x}, u)\, du} u(\mathbf{x}, s)\, ds = \int_0^t e^{\int_{t-r}^t \beta(\mathbf{x}, u)\, du} u(\mathbf{x}, t - r)\, dr. \tag{C52}$$

Now, for each fixed $r \geq 0$, as $t \to \infty$, $s = t - r \to \infty$. Thus, both

$$\int_{t-r}^t \beta(\mathbf{x}, u)\, du \to \beta^*(\mathbf{x})r \quad \text{and} \quad u(\mathbf{x}, t - r) \to u^*(\mathbf{x}) \tag{C53}$$

hold uniformly in $\mathbf{x}$. Therefore

$$e^{\int_{t-r}^{t} \beta(\mathbf{x},u)\, du} u(\mathbf{x}, t - r) \to u^*(\mathbf{x}) e^{\beta^*(\mathbf{x})r} \tag{C54}$$

uniformly in $\mathbf{x}$, for each fixed $r \geq 0$. Moreover, since $\beta(\mathbf{x}, t) \leq -\lambda < 0$

$$e^{\int_{t-r}^{t} \beta(\mathbf{x},u)\, du} \leq e^{-\lambda r}, \tag{C55}$$

and therefore

$$|e^{\int_{t-r}^{t} \beta(\mathbf{x},u)\, du} u(\mathbf{x}, t - r)| \leq M e^{-\lambda r}. \tag{C56}$$

$(f_t(\mathbf{x}, r))_{t \geq 0}$ forms a net rather than a sequence, so we pick any sequence $(t_n)_{n \geq 1} \subset [0, \infty)$ such that $t_n \to \infty$. For each fixed $r \geq 0$, $f_{t_n}(\mathbf{x}, r) \to u^*(\mathbf{x}) e^{\beta^*(\mathbf{x})r}$ uniformly in $\mathbf{x}$, and $|f_{t_n}(\mathbf{x}, r)| \leq M e^{-\lambda r}$ with $M e^{-\lambda r} \in L^1([0, \infty))$. Thus, by the Dominated Convergence Theorem, we have

$$\lim_{n \to \infty} \int_0^\infty f_{t_n}(\mathbf{x}, r)\, dr = \int_0^\infty u^*(\mathbf{x}) e^{\beta^*(\mathbf{x})r}\, dr, \quad \text{uniformly in } \mathbf{x}. \tag{C57}$$

Since the limit is independent of the chosen sequence, it follows that

$$\lim_{t \to \infty} I(\mathbf{x}, t) = -\frac{u^*(\mathbf{x})}{\beta^*(\mathbf{x})}, \quad \text{uniformly in } \mathbf{x}. \tag{C58}$$

We finally conclude that

$$\lim_{t \to \infty} \left\| y(\mathbf{x}, t) + \frac{u^*(\mathbf{x})}{\beta^*(\mathbf{x})} \right\|_{L^\infty(\mathcal{X})} = 0. \tag{C59}$$

$\square$

# D  Joint Covariance Function For Linear FiDE

Let $(\Omega, \mathcal{F}, \mathbb{P})$ be a probability space supporting two independent stochastic processes $y_0 : \mathcal{X} \times \Omega \to \mathbb{R}$ and $u : \mathcal{X} \times [0, \infty) \times \Omega \to \mathbb{R}$. We place Gaussian process priors over both

$$y_0(\mathbf{x}) \sim \mathcal{GP}(0, k_0(\mathbf{x}, \mathbf{x}')), \quad u(\mathbf{x}, t) \sim \mathcal{GP}(0, k_u(\mathbf{x}, t, \mathbf{x}', t')). \tag{D1}$$

The solution to the linear fidelity-indexed dynamical system is given by

$$y(\mathbf{x}, t) = e^{\int_0^t \beta(\mathbf{x},z)\, dz} y(\mathbf{x}, 0) + \int_0^t e^{\int_s^t \beta(\mathbf{x},z)\, dz} u(\mathbf{x}, s)\, ds, \tag{D2}$$

in which $\beta(\mathbf{x}, t)$ is a deterministic function satisfying $\beta(\mathbf{x}, t) < 0$ and regular enough to ensure well-defined integrals. Thus, $y(\mathbf{x}, t)$ is a zero-mean GP (as an affine transformations of GPs), and its expectation is taken over the joint measure induced by both GP priors. The covariance function of $y(\mathbf{x}, t)$ is therefore given by $k(\mathbf{x}, t, \mathbf{x}', t') = \mathbb{E}_{\omega \sim \mathbb{P}}[y(\mathbf{x}, t; \omega) y(\mathbf{x}', t'; \omega)]$ and expanding the product in the argument yields

$$y(\mathbf{x}, t) y(\mathbf{x}', t') = e^{\int_0^t \beta(\mathbf{x},z)\, dz} e^{\int_0^{t'} \beta(\mathbf{x}',z)\, dz} y_0(\mathbf{x}) y_0(\mathbf{x}')$$
$$+ \int_0^t \int_0^{t'} e^{\int_s^t \beta(\mathbf{x},z)\, dz} e^{\int_{s'}^{t'} \beta(\mathbf{x}',z)\, dz} u(\mathbf{x}, s) u(\mathbf{x}', s')\, ds\, ds', \tag{D3}$$

noting that the cross terms disappear due to independence. Taking expectations and using linearity, the zero-mean assumptions and Fubini's Theorem (which holds for GPs under mild assumptions), we obtain

$$k(\mathbf{x}, t, \mathbf{x}', t') = \mathbb{E}[y(\mathbf{x}, t) y(\mathbf{x}', t')]$$
$$= e^{\int_0^t \beta(\mathbf{x},z)\, dz} e^{\int_0^{t'} \beta(\mathbf{x}',z)\, dz} k_0(\mathbf{x}, \mathbf{x}') + \int_0^t \int_0^{t'} e^{\int_s^t \beta(\mathbf{x},z)\, dz} e^{\int_{s'}^{t'} \beta(\mathbf{x}',z)\, dz} k_u(\mathbf{x}, s, \mathbf{x}', s')\, ds\, ds', \tag{D4}$$

using $\mathbb{E}[y_0(\mathbf{x}) y_0(\mathbf{x}')] = k_0(\mathbf{x}, \mathbf{x}')$ and $\mathbb{E}[u(\mathbf{x}, s) u(\mathbf{x}', s')] = k_u(\mathbf{x}, s, \mathbf{x}', s')$.

# E  Closed-from Kernels

**Constant $\beta(\mathbf{x}, t) = \beta$ and Periodic Kernel.**
Consider the periodic exponential sine squared (ESS) kernel, defined as

$$k_t^u(t, t') = \sigma^2 e^{-\frac{2 \sin^2(\pi|t-t'|/p)}{\ell^2}} \tag{E1}$$

in which $\sigma^2$ is the signal variance, $\ell$ is the length scale, and $p$ is the period. Assuming a constant $\beta(\mathbf{x}, t) = \beta$ we obtain

$$k(\mathbf{x}, t, \mathbf{x}', t') = e^{\beta(t-t_0)} e^{\beta(t'-t_0)} k_0(\mathbf{x}, \mathbf{x}') + \sigma^2 k_\mathbf{x}^u(\mathbf{x}, \mathbf{x}') \int_{t_0}^t \int_{t_0}^{t'} e^{\beta(t-s)} e^{\beta(t'-s')} e^{-\frac{2\sin^2(\pi|s-s'|/p)}{\ell^2}} ds ds'$$

$$= e^{\beta(t-t_0)} e^{\beta(t'-t_0)} k_0(\mathbf{x}, \mathbf{x}')$$

$$+ \sigma^2 k_\mathbf{x}^u(\mathbf{x}, \mathbf{x}') \left( \frac{p}{2\pi} \sum_{n=-\infty}^{\infty} \frac{1}{\beta - \frac{2\pi in}{p}} \left[ e^{\beta(t-t_0) - \frac{2\pi in}{p}(t-t_0)} - e^{\beta(t'-t_0) - \frac{2\pi in}{p}(t'-t_0)} \right] \right.$$

$$\left. \times \exp\left(-\frac{2}{\ell^2}\right) B_n\left(\frac{2}{\ell^2}\right) \right),$$

(E2)

in which $B_n(\cdot)$ is the modified Bessel function of the first kind of order $n$.

**Constant $\beta(\mathbf{x}, t) = \beta$ and Matérn Kernel.**
The Matérn kernel with parameter $\nu$ is defined as

$$k_t^u(t, t') = \sigma^2 \frac{2^{1-\nu}}{\Gamma(\nu)} \left( \sqrt{2\nu} \frac{|t-t'|}{\ell} \right)^\nu K_\nu\left( \sqrt{2\nu} \frac{|t-t'|}{\ell} \right),$$

(E3)

in which $\sigma^2$ is the signal variance, $\ell$ is the characteristic length scale, $\Gamma(\cdot)$ is the gamma function, and $K_\nu(\cdot)$ is the modified Bessel function of the second kind of order $\nu$. For $\nu = \frac{3}{2}$, the covariance function is

$$k(\mathbf{x}, t, \mathbf{x}', t') = e^{\beta(t-t_0)} e^{\beta(t'-t_0)} k^0(\mathbf{x}, \mathbf{x}')$$

$$+ \sigma^2 k_\mathbf{x}^u(\mathbf{x}, \mathbf{x}') \int_{t_0}^t \int_{t_0}^{t'} e^{\beta(t-s)} e^{\beta(t'-s')} \left( 1 + \frac{\sqrt{3}|s-s'|}{\ell} \right) \exp\left( -\frac{\sqrt{3}|s-s'|}{\ell} \right) ds ds'.$$

(E4)

The double integral in the second term can be evaluated analytically, yielding

$$k(\mathbf{x}, t, \mathbf{x}', t') = e^{\beta(t-t_0)} e^{\beta(t'-t_0)} k_0(\mathbf{x}, \mathbf{x}')$$

$$+ \sigma^2 k_\mathbf{x}^u(\mathbf{x}, \mathbf{x}') \int_{t_0}^t \int_{t_0}^{t'} e^{\beta(t-s)} e^{\beta(t'-s')} \left( 1 + \frac{\sqrt{3}|s-s'|}{\ell} \right) e^{-\frac{\sqrt{3}|s-s'|}{\ell}} ds ds'$$

$$= e^{\beta(t-t_0)} e^{\beta(t'-t_0)} k_0(\mathbf{x}, \mathbf{x}')$$

$$+ \sigma^2 k_\mathbf{x}^u(\mathbf{x}, \mathbf{x}') \left( \frac{1}{\beta^2 - \frac{3}{\ell^2}} \left[ e^{\beta(t-t_0)} + e^{\beta(t'-t_0)} - e^{-\frac{\sqrt{3}}{\ell}|t-t'|} \left( e^{\beta(t-t_0) - \frac{\sqrt{3}}{\ell}(t'-t_0)} + e^{\beta(t'-t_0) - \frac{\sqrt{3}}{\ell}(t-t_0)} \right) \right] \right.$$

$$+ \frac{\sqrt{3}}{\ell} \left[ \frac{1}{\beta^2 - \frac{3}{\ell^2}} \left( e^{\beta(t-t_0)} + e^{\beta(t'-t_0)} \right) - \frac{1}{\beta + \frac{\sqrt{3}}{\ell}} \left( e^{\beta(t-t_0) - \frac{\sqrt{3}}{\ell}(t-t_0)} + e^{\beta(t'-t_0) - \frac{\sqrt{3}}{\ell}(t'-t_0)} \right) \right]$$

$$- \frac{1}{\beta + \frac{\sqrt{3}}{\ell}} \left[ e^{\beta(t-t_0) - \frac{\sqrt{3}}{\ell}(t-t_0)} + e^{\beta(t'-t_0) - \frac{\sqrt{3}}{\ell}(t'-t_0)} \right]$$

$$\left. + \frac{1}{\beta - \frac{\sqrt{3}}{\ell}} \left[ e^{\beta(t-t_0) + \frac{\sqrt{3}}{\ell}(t-t_0)} + e^{\beta(t'-t_0) + \frac{\sqrt{3}}{\ell}(t'-t_0)} \right] \right).$$

(E5)

For $\nu = \frac{5}{2}$, we obtain

$$
k(\mathbf{x}, t, \mathbf{x}', t') = e^{\beta(t-t_0)} e^{\beta(t'-t_0)} k_0(\mathbf{x}, \mathbf{x}')
$$

$$
+ \sigma^2 k_{\mathbf{x}}^u(\mathbf{x}, \mathbf{x}') \int_{t_0}^t \int_{t_0}^{t'} e^{\beta(t-s)} e^{\beta(t'-s')} \left( 1 + \frac{\sqrt{5}|s - s'|}{\ell} + \frac{5(s - s')^2}{3\ell^2} \right) e^{-\frac{\sqrt{5}|s-s'|}{\ell}} ds ds'
$$

$$
= e^{\beta(t-t_0)} e^{\beta(t'-t_0)} k_0(\mathbf{x}, \mathbf{x}')
$$

$$
+ \sigma^2 k_{\mathbf{x}}^u(\mathbf{x}, \mathbf{x}') \left( \frac{1}{\beta^2 - \frac{5}{\ell^2}} \left[ e^{\beta(t-t_0)} + e^{\beta(t'-t_0)} - e^{-\frac{\sqrt{5}}{\ell}|t-t'|} \left( e^{\beta(t-t_0) - \frac{\sqrt{5}}{\ell}(t'-t_0)} + e^{\beta(t'-t_0) - \frac{\sqrt{5}}{\ell}(t-t_0)} \right) \right] \right.
$$

$$
+ \frac{\sqrt{5}}{\ell} \left[ \frac{1}{\beta^2 - \frac{5}{\ell^2}} \left( e^{\beta(t-t_0)} + e^{\beta(t'-t_0)} \right) - \frac{1}{\beta + \frac{\sqrt{5}}{\ell}} \left( e^{\beta(t-t_0) - \frac{\sqrt{5}}{\ell}(t-t_0)} + e^{\beta(t'-t_0) - \frac{\sqrt{5}}{\ell}(t'-t_0)} \right) \right]
$$

$$
+ \frac{5}{3\ell^2} \left[ \frac{1}{\beta^2 - \frac{5}{\ell^2}} \left( e^{\beta(t-t_0)} + e^{\beta(t'-t_0)} \right) - \frac{1}{\beta + \frac{\sqrt{5}}{\ell}} \left( e^{\beta(t-t_0) - \frac{\sqrt{5}}{\ell}(t-t_0)} + e^{\beta(t'-t_0) - \frac{\sqrt{5}}{\ell}(t'-t_0)} \right) \right]
$$

$$
- \frac{1}{\beta + \frac{\sqrt{5}}{\ell}} \left[ e^{\beta(t-t_0) - \frac{\sqrt{5}}{\ell}(t-t_0)} + e^{\beta(t'-t_0) - \frac{\sqrt{5}}{\ell}(t'-t_0)} \right]
$$

$$
+ \frac{1}{\beta - \frac{\sqrt{5}}{\ell}} \left[ e^{\beta(t-t_0) + \frac{\sqrt{5}}{\ell}(t-t_0)} + e^{\beta(t'-t_0) + \frac{\sqrt{5}}{\ell}(t'-t_0)} \right]
$$

$$
- \frac{5}{3\ell^2} \left[ \frac{1}{\beta^2 - \frac{5}{\ell^2}} \left( e^{\beta(t-t_0)} + e^{\beta(t'-t_0)} \right) - \frac{1}{(\beta + \frac{\sqrt{5}}{\ell})^2} \left( e^{\beta(t-t_0) - \frac{\sqrt{5}}{\ell}(t-t_0)} + e^{\beta(t'-t_0) - \frac{\sqrt{5}}{\ell}(t'-t_0)} \right) \right]
$$

$$
\left. + \frac{5}{3\ell^2} \left[ \frac{1}{(\beta - \frac{\sqrt{5}}{\ell})^2} \left( e^{\beta(t-t_0) + \frac{\sqrt{5}}{\ell}(t-t_0)} + e^{\beta(t'-t_0) + \frac{\sqrt{5}}{\ell}(t'-t_0)} \right) \right] \right).
$$

$$\tag{E6}$$

**Constant $\beta(\mathbf{x}, t) = \beta$ and Random Fourier Features.**
Since SE, periodic, and Matérn kernels are all stationary, we can extend the closed-form solution to any stationary kernel. Recall that any stationary kernel can be represented as a Fourier series, and the random Fourier features (RFF) kernel is defined as

$$
k_t^u(t, t') = \frac{\sigma^2}{m} \sum_{i=1}^m \cos(\omega_i(t - t')), \tag{E7}
$$

in which $\sigma^2$ is the signal variance, $m$ is the number of random features, and $\omega_i$ are randomly sampled frequencies from a distribution $p(\omega)$ (typically a Gaussian distribution with zero mean and variance $1/\ell^2$, where $\ell$ is the length scale of the squared exponential kernel being approximated). Assuming a constant $\beta(\mathbf{x}, t) = \beta$, the covariance function becomes

$$
k(\mathbf{x}, t, \mathbf{x}', t') = e^{\beta(t-t_0)} e^{\beta(t'-t_0)} k^0(\mathbf{x}, \mathbf{x}')
$$

$$
+ \frac{\sigma^2}{m} k_{\mathbf{x}}^u(\mathbf{x}, \mathbf{x}') \sum_{i=1}^m \int_{t_0}^t \int_{t_0}^{t'} e^{\beta(t-s)} e^{\beta(t'-s')} \cos(\omega_i(s - s')) ds ds'
$$

$$
= e^{\beta(t-t_0)} e^{\beta(t'-t_0)} k^0(\mathbf{x}, \mathbf{x}')
$$

$$
+ \frac{\sigma^2}{m} k_{\mathbf{x}}^u(\mathbf{x}, \mathbf{x}') \sum_{i=1}^m \left( \frac{1}{\beta^2 + \omega_i^2} \left[ e^{\beta(t-t_0)} \cos(\omega_i(t - t_0)) + e^{\beta(t'-t_0)} \cos(\omega_i(t' - t_0)) \right. \right.
$$

$$
- e^{\beta(t+t'-2t_0)} \cos(\omega_i(t - t')) \right]
$$

$$
\left. + \frac{\omega_i}{\beta^2 + \omega_i^2} \left[ e^{\beta(t-t_0)} \sin(\omega_i(t - t_0)) - e^{\beta(t'-t_0)} \sin(\omega_i(t' - t_0)) \right] \right).
$$

$$\tag{E8}$$

# F  Background on BOCA and Theorem 3

Here we describe the BOCA method of Kandasamy et al. Kandasamy et al. [3] (adapted to our problem). We set $\mathcal{X} = [0, 1]^d$ and $\mathcal{T} = [0, 1]$ and let $f(\mathbf{x}) = y(\mathbf{x}, 1)$ be the black-box function we aim to optimize, with different fidelities $t$ given by a function $y(\mathbf{x}, t) : \mathcal{X} \times \mathcal{T} \to \mathbb{R}$. Assume that

1. $y(\mathbf{x}, t) \sim \mathcal{GP}(0, \kappa)$ with a product kernel

$$\kappa\left(\mathbf{x}, t, \mathbf{x}', t'\right) = \kappa_0 \, k_t(|t - t'|) k_{\mathbf{x}}(\|\mathbf{x} - \mathbf{x}'\|), \tag{F1}$$

in which $\kappa_0 > 0$ and both $k_t$, $k_{\mathbf{x}}$ are valid kernels.

2. $\lambda(t)$ is the cost function for evaluating $y$ at fidelity $t$.

Let $\mathcal{A} \subset \mathcal{X}$, and let $\mathcal{A}_n = \{\mathbf{x}_1, \ldots, \mathbf{x}_n\} \subset \mathcal{A}$ be a finite-cardinality subset. Define noisy observations $y_{\mathcal{A}_n} = f_{\mathcal{A}_n} + \epsilon_{\mathcal{A}_n}$ with $\epsilon_n \sim \mathcal{N}(0, \eta^2)$. Here, $y_{\mathcal{A}_n} = (y_1, \ldots, y_n)^T$, $f_{\mathcal{A}_n} = (f(\mathbf{x_1}, \ldots, \mathbf{x_n})^\top$ and $\epsilon_{\mathcal{A}_n} = (\epsilon_1, \ldots, \epsilon_n)^\top$. For a continuous random vector $X \sim \mathcal{N}(\mu, \Sigma)$, the differential entropy is defined as

$$\mathbb{H}(X) = \frac{1}{2} \log\left[(2\pi e)^n \det(\Sigma)\right]. \tag{F2}$$

The mutual information or information gain $\mathbb{I}(y_{\mathcal{A}_n}; f_{\mathcal{A}_n})$ is a measure of the reduction in uncertainty in the values of $f_{\mathcal{A}_n}$ after acquiring observations $y_{\mathcal{A}_n}$. For Gaussian random vectors, information gain is given by the difference in differential entropy

$$\mathbb{I}(y_{\mathcal{A}_n}; f_{\mathcal{A}_n}) = \mathbb{H}(y_{\mathcal{A}_n}) - \mathbb{H}(y_{\mathcal{A}_n} \mid f_{\mathcal{A}_n}) = \frac{1}{2} \log \det\left(I + \eta^{-2} K_{\mathcal{A}_n}\right), \tag{F3}$$

in which $K_{\mathcal{A}_n} \in \mathbb{R}^{n \times n}$ is the kernel matrix with entries $(K_{\mathcal{A}_n})_{ij} = \kappa(\mathbf{x}_i, \mathbf{x}_j)$. The maximal information gain after $n$ evaluations on a subset $\mathcal{A} \subset \mathcal{X}$ is then defined as

$$\gamma_n(\mathcal{A}) = \max_{\mathcal{A}_n \subset \mathcal{A}, |\mathcal{A}_n| = n} \mathbb{I}(y_{\mathcal{A}_n}; f_{\mathcal{A}_n}). \tag{F4}$$

Let $\mathbf{x}^* = \arg\max_{\mathbf{x} \in \mathcal{X}} f(\mathbf{x})$ and $f^* = f(\mathbf{x}^*)$. The simple regret after capital $\Lambda$ is defined as

$$r(\Lambda) = \begin{cases} \min_{i \in \{1, \ldots, N\}: \, t_i = 1} f^* - y(\mathbf{x}_i, t), & \text{if any query was made at } t = 1, \\ +\infty, & \text{otherwise.} \end{cases} \tag{F5}$$

$\Lambda$ refers to the total budget or cumulative cost allowed for querying the objective function, and therefore bounds the cumulative cost of all function evaluations.

A measure of how informative fidelity $t$ is with regards to the maximum fidelity $t = 1$ is given by the following information-gap function

$$\xi(t) = \sqrt{1 - k_t(|t - 1|)^2}. \tag{F6}$$

A smaller $\xi(t)$ implies greater informativeness. If $\xi(t)$ is smooth, the gap decreases as $t \to 1$. Now define

$$\mathcal{X}_\rho = \{\mathbf{x} \in \mathcal{X} : f^* - f(\mathbf{x}) \le 2\rho\sqrt{\kappa_0}\xi(0)\} \tag{F7}$$

noting that $\xi(0)$ is the maximal fidelity gap. We note here that this expression appears to be different from that in Kandasamy et al. [3], in which the authors use $\xi(\sqrt{p})$ for $\mathcal{T} = [0, 1]^p$ throughout their paper. Clearly $\sqrt{p} \notin [0, 1]^p$ and this is a typing error (the correct expression is $\xi(\mathbf{0})$). Most queries at the highest fidelity $t = 1$ are made from the set $\mathcal{X}_\rho$. In relation to $\mathcal{X}_\rho$, let

$$\mathcal{X}_{\rho,n} = \left\{x \in \mathcal{X} : B_2\left(x, \frac{\sqrt{d}}{n^{\alpha/2d}}\right) \cap \mathcal{X}_\rho \ne \emptyset\right\}, \tag{F8}$$

in which $B_2(x, \varepsilon)$ denotes the Euclidean ($\ell^2$) ball of radius $\varepsilon$ centred at $x$, $n$ is the number of queries at any fidelity and $\alpha \in (0, 1)$. The set $\mathcal{X}_{\rho,n}$ includes all points in $\mathcal{X}$ that lie within a Euclidean distance $\frac{\sqrt{d}}{n^{\alpha/2d}}$ of $\mathcal{X}_\rho$, and can be interpreted as a polynomial-rate dilation of the latter. As $n \to \infty$, we have $\mathcal{X}_{\rho,n} \to \mathcal{X}_\rho$ at a rate of $n^{-\alpha/2d}$.

At iteration $n$, the BOCA algorithm selects a query point $(\mathbf{x}_n, t_n) \in \mathcal{X} \times [0, T)$ in two stages. First, it constructs an upper confidence bound (UCB) acquisition function $\varphi_n(\mathbf{x})$ over the domain $\mathcal{X}$, conditioned on the fidelity $t$

$$\varphi_n(\mathbf{x}) = \mu_{n-1}(\mathbf{x}, 1) + \sqrt{\beta_n} \, \sigma_{n-1}(\mathbf{x}, 1), \tag{F9}$$

in which $\mu_{n-1}(\mathbf{x}, 1)$ and $\sigma_{n-1}(\mathbf{x}, 1)$ are the mean and standard deviation of the posterior GP slice at $t = 1$ over $y(\mathbf{x}, t)$ (i.e., over $f(\mathbf{x}) = y(\mathbf{x}, 1)$) given all previous observations $\{y(\mathbf{x}_i, t_i)\}_{i=1}^{n-1}$, and $\beta_n$ is a $t$-varying confidence parameter. The latter is given by

$$\beta_n = 2 \log\left(\frac{\pi^2 n^2}{2\delta}\right) + 4d \log(n) + \max\left\{0, \, 2d \log\left(bd \log\left(\frac{6ad}{\delta}\right)\right)\right\}, \tag{F10}$$

in which $\delta \in (0,1)$ and $a, b > 0$ are constants depending on the kernel $k_{\mathbf{x}}$. Specifically, noting that $f(\mathbf{x}) \sim \mathcal{GP}(0, k_{\mathbf{x}}(\mathbf{x}, \mathbf{x}'))$ since $k_t(1,1) = 1$, it is assumed that there exist constants $a, b > 0$ such that

$$\mathbb{P}\left(\sup_{\mathbf{x} \in \mathcal{X}} \left| \frac{\partial}{\partial x_i} f(\mathbf{x}) \right| > J \right) \leq a e^{-(J/b)^2} \quad \text{for all } J > 0, \text{ and } i = 1, \ldots, d. \tag{F11}$$

Having chosen the next input $\mathbf{x}_n \in \arg\max_{\mathbf{x} \in \mathcal{X}} \varphi_n(\mathbf{x})$, the fidelity level $t_n \in [0,1)$ is chosen from the filtered set

$$\mathcal{T}_n(\mathbf{x}_n) = \left\{ t \in [0,1] : \lambda(t) < \lambda(1), \quad \sigma_{n-1}(\mathbf{x}_n, t) > \gamma(t), \quad \xi(t) > \beta_n^{-1/2} \xi(0) \right\}, \tag{F12}$$

where $\gamma(t) = \sqrt{\kappa_0}\, \xi(t) \left( \frac{\lambda(t)}{\lambda(1)} \right)^q$. $q$ depends on the kernel and for a SE kernel $q = (1 + d + 2)^{-1}$. If $\mathcal{T}_n(\mathbf{x}_n) \neq \emptyset$, the algorithm chooses the cheapest fidelity from this set, i.e.,

$$t_n = \arg\min_{t \in \mathcal{T}_n(\mathbf{x}_n)} \lambda(t), \tag{F13}$$

otherwise it sets $t_n = 1$. With the definitions above we can state the main result.

**Theorem F1** (Kandasamy et al. [3], Theorem 8). *Let $\mathcal{X} = [0,1]^d$, $\mathcal{T} = [0,1]$ and $y(\mathbf{x}, t) \sim \mathcal{GP}(0, \kappa)$ with $\kappa(\mathbf{x}, t, \mathbf{x}', t') = \kappa_0\, k_t(|t - t'|) k_{\mathbf{x}}(\|\mathbf{x} - \mathbf{x}'\|)$, in which $k_{\mathbf{x}}$ satisfies Assumption (F11). Choose $\delta \in (0,1)$ and execute BOCA with $\beta_n$ chosen as in F10). Then, for any $\alpha \in (0,1)$, there exist $\rho > 0$ and $\Lambda_0$ such that with probability at least $1 - \delta$*

$$r(\Lambda) \leq \sqrt{\frac{C_1 \beta_{2n_\Lambda} \gamma_{2n_\Lambda}(\mathcal{X}_{\rho,n})}{n_\Lambda}} + \sqrt{\frac{C_1 \beta_{2n_\Lambda} \gamma_{2n_\Lambda^\alpha}(\mathcal{X})}{n_\Lambda^{2-\alpha}}} + \frac{\pi^2}{6 n_\Lambda}, \quad \forall \Lambda \geq \Lambda_0, \tag{F14}$$

*in which $C_1$ is a universal constant and $n_\Lambda = \left\lfloor \frac{\Lambda}{\lambda(1)} \right\rfloor$, while $\rho > \max\{2, 1 + \sqrt{(1 + 2/\alpha)/(1 + d)}\}$.*

# G    Asymptotic behaviour of the LiFiDE kernel

We expand $\mathcal{I}(t, t')$ around $t' = t + \delta$ for small $\delta$ as

$$\mathcal{I}(t, t + \delta) = \mathcal{I}_0(t) + \mathcal{I}_1(t)\delta + \mathcal{O}(\delta^2), \tag{G1}$$

in which $\mathcal{I}_0(t)$ and $\mathcal{I}_1(t)$ are given by

$$\mathcal{I}_0(t) = \frac{e^{\beta^2 \ell^2 / 2} \sqrt{\pi} \ell}{\beta} \left[ \operatorname{erf}\left( \frac{t}{\ell} + \frac{\beta \ell}{\sqrt{2}} \right) - e^{-2\beta t} \left( \operatorname{erf}\left( \frac{t}{\ell} - \frac{\beta \ell}{\sqrt{2}} \right) + \operatorname{erf}\left( \frac{\beta \ell}{\sqrt{2}} \right) \right) \right], \tag{G2}$$

$$\mathcal{I}_1(t) = \frac{e^{\beta^2 \ell^2 / 2} \sqrt{\pi} \ell}{2\beta} \left[ \beta \operatorname{erf}\left( \frac{\beta \ell}{\sqrt{2}} \right) - \frac{2}{\sqrt{\pi} \ell} e^{-\left( \frac{\beta \ell}{\sqrt{2}} \right)^2} + \frac{e^{-\left( \frac{t}{\ell} + \frac{\beta \ell}{\sqrt{2}} \right)^2}}{\sqrt{\pi} \ell} - \frac{e^{-2\beta t} e^{-\left( \frac{t}{\ell} - \frac{\beta \ell}{\sqrt{2}} \right)^2}}{\sqrt{\pi} \ell} \right]. \tag{G3}$$

$\mathcal{I}_0(t)$ increases monotonically as $t \to \infty$. The crucial term in $\mathcal{I}_1(t)$ is

$$\frac{e^{-\left( \frac{t}{\ell} + \frac{\beta \ell}{\sqrt{2}} \right)^2}}{\sqrt{\pi} \ell} - \frac{e^{-2\beta t} e^{-\left( \frac{t}{\ell} - \frac{\beta \ell}{\sqrt{2}} \right)^2}}{\sqrt{\pi} \ell} = \frac{e^{-\frac{t^2}{\ell^2} - \frac{\beta^2 \ell^2}{2}}}{\sqrt{\pi} \ell} \left( e^{-\sqrt{2}\beta t} - e^{-\beta t(2 - \sqrt{2})} \right), \tag{G4}$$

which is strictly negative. For the squared exponential $k_{\mathrm{SE}}$, due to symmetry and stationarity

$$k_{\mathrm{SE}}(t, t + \delta) = \mathcal{I}_0 + \mathcal{I}_2 \delta^2 + \mathcal{O}(\delta^4), \tag{G5}$$

in which $\mathcal{I}_0$ and $\mathcal{I}_2$ are independent of $t$. Thus, the decay is $\mathcal{O}(\delta^2)$ compared to $\mathcal{O}(\delta)$ for the LiFiDE kernel. Therefore, correlations between values of $y(\mathbf{x}, t)$ and $y(\mathbf{x}, t')$ around $t' = t + \delta$ are more concentrated, and do not depend on $t$.

**Remark G1** (On Regret Bounds and Kernel Choice). *BOCA's theoretical regret bound depends on the fidelity-gap function $\xi(t)$, which in turn is determined by the fidelity kernel through*

$$\xi(t) = \sqrt{1 - k_t(|t - T|)^2}, \tag{G6}$$

*for a maximum fidelity $T$. For the SE kernel $k_t(|t - T|) = \exp\left( -\frac{(t - T)^2}{2\ell^2} \right)$ we have the approximation*

$$\xi(t) \approx \frac{|t - T|}{\ell} + \mathcal{O}(|t - T|^3), \tag{G7}$$

*around $t = T$, which decays linearly in the distance to the highest fidelity $T$. Thus, $\mathcal{X}_\rho$ is sharply localised and regret is relatively easy to control.*

*We can simplify the expression for the LiFiDE kernel as follows*

$$k(t, t') = c_1 e^{-\beta t} e^{-\beta t'} + c_2 \mathcal{I}(t, t'),  \tag{G8}$$

*in which $c_1$ and $c_2$ are the values of $k_0$ and $k_{\mathbf{x}}$ at a fixed $(\mathbf{x}, \mathbf{x}') \in \mathcal{X}^2$. Consider $t = T - \delta$ in the limit $\delta \to 0^+$. Expanding the kernel around $t' = t + \delta$ we obtain*

$$k(t, t + \delta) = k(t, t) + \mathcal{I}_1(t)\delta + \mathcal{O}(\delta^2),  \tag{G9}$$

*in which $\mathcal{I}_1(t)$ is defined in (G3). We now normalise the kernel so that it takes values in $[0, 1]$ by defining*

$$\tilde{k}(t, t + \delta) = \frac{k(t, t + \delta)}{\sqrt{k(t, t)k(t + \delta, t + \delta)}}  \tag{G10}$$

*Since $k(t, t + \delta) = k(t, t) + \mathcal{O}(\delta)$ and $k(t + \delta, t + \delta) = k(t, t) + \mathcal{O}(\delta)$, we have*

$$\tilde{k}(t, t + \delta)^2 = 1 + 2\frac{\mathcal{I}_1(t)}{k(t, t)}\delta + \mathcal{O}(\delta^2).  \tag{G11}$$

*The square fidelity gap becomes*

$$\xi(t)^2 = 1 - \tilde{k}(t, t + \delta)^2 = -2\frac{\mathcal{I}_1(t)}{k(t, t)}\delta + \mathcal{O}(\delta^2).  \tag{G12}$$

*Since $\mathcal{I}_1(t) < 0$, $\forall t$, we finally obtain*

$$\xi(t) \sim \sqrt{-2\frac{\mathcal{I}_1(t)}{k(t, t)}}\sqrt{\delta} = \sqrt{-2\frac{\mathcal{I}_1(t)}{k(t, t)}}\sqrt{|T - t|} \quad as \ |T - t| \to 0,  \tag{G13}$$

*which shows that near the highest fidelity level, the fidelity gap decays as $\sqrt{|T - t|}$, i.e., at a much slower rate than that for the SE kernel.*

# H Additional Experimental Results

Below we provide more detailed information on the sampling functions for all continuous MFBO experiments conducted in Section 5. Each marker represents a sampling point. Among the numerous seeds, we selected the first four for display. The results for the Branin, Currin, Park, nonlinear sin, Forrester, Bohachevsky, Borehole, Colville, and Himmelblau functions are shown in Fig. H1–Fig. H9, respectively.

We can see the advantage of CAMO in almost all cases in terms of the query cost and simple regret. The query cost is significantly reduced compared to the other methods, and the simple regret is also competitive. In particular, CAMO always exhibits a fast convergence rate in the early stage of the optimization process. Upon closer inspection, CAMO conducts most of its exploration in the low-fidelity region, reducing the query cost, which is consistent with the theoretical analysis. To explore the performance of different models combined with different acquisition functions, we provide additional results with different costs for the Forrester function in Fig. H10. The results are consistent with those in the main text. CAMO is the optimal choice, particularly when the cost function is exponential-like, which is more realistic in real-world applications such as finite element analysis and neural network architecture search (NAS).

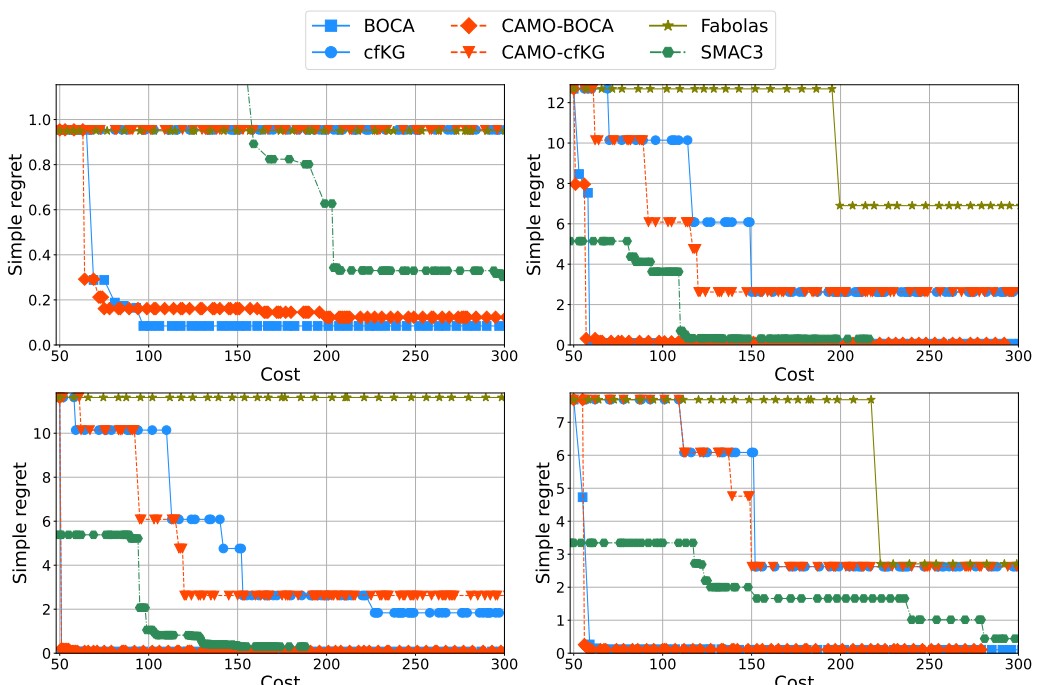

Figure H1: Branin function MFBO results from four random seeds.

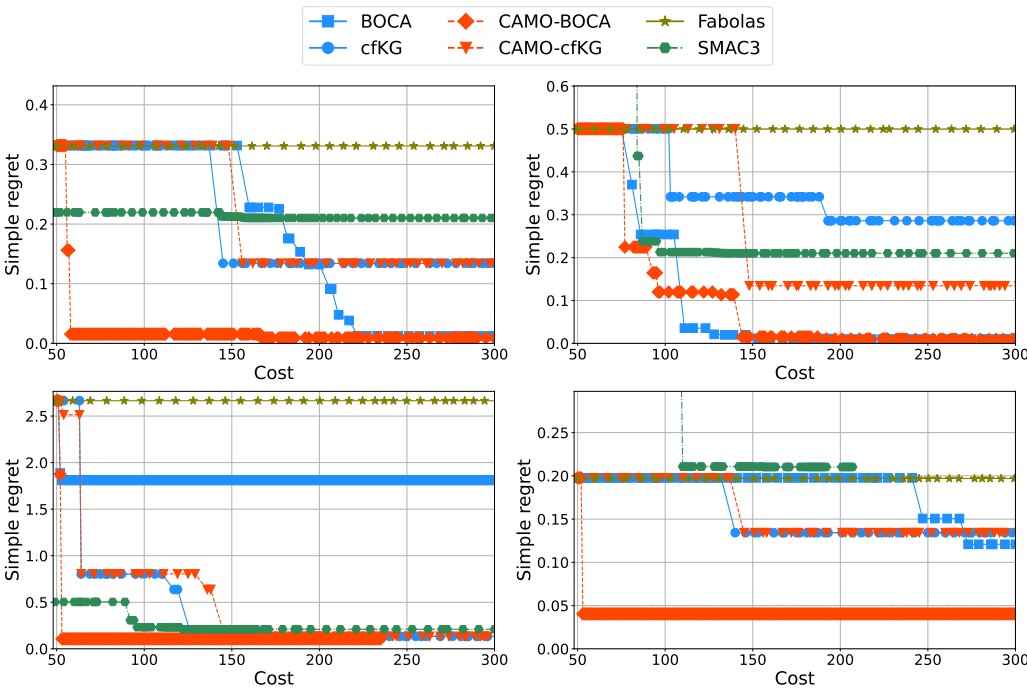

Figure H2: Currin function MFBO results from four random seeds.

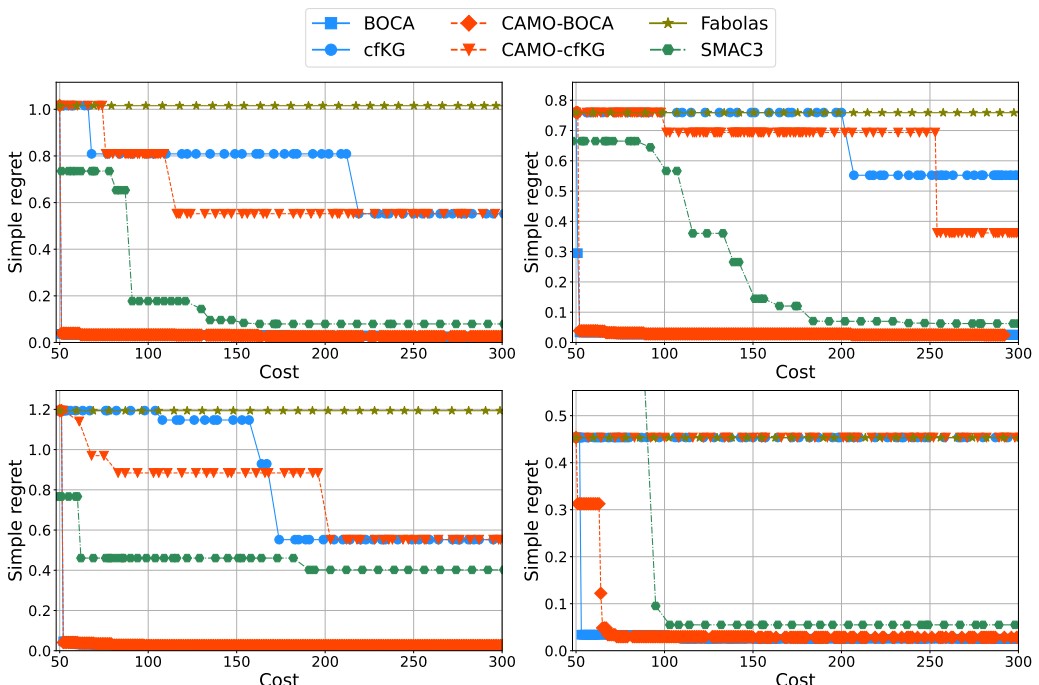

Figure H3: Park function MFBO results from four random seeds.

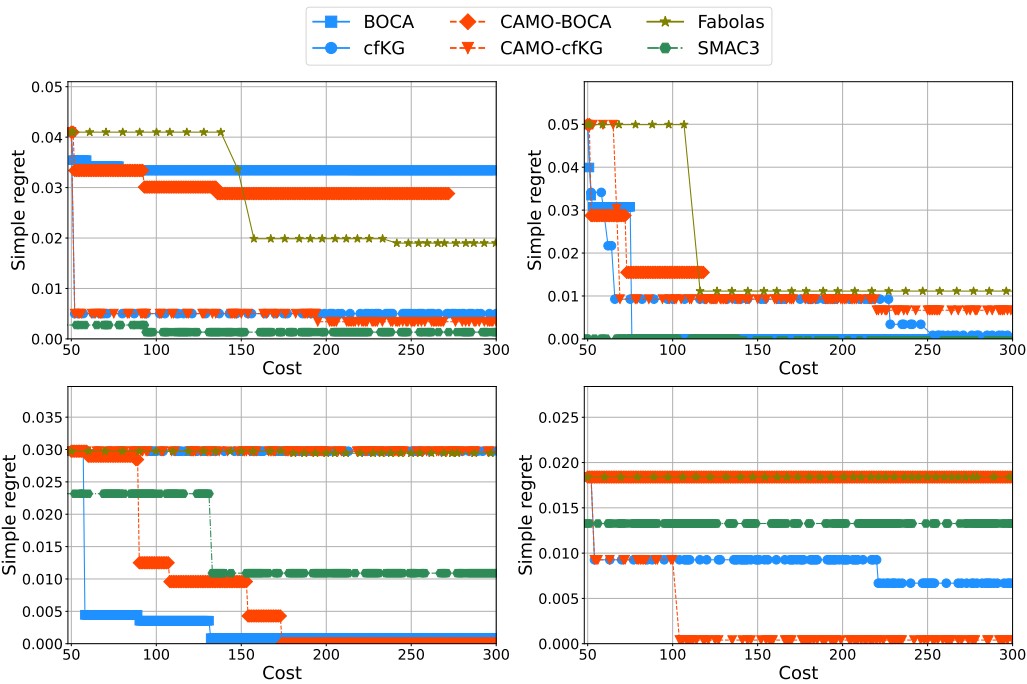

Figure H4: Nonlinear sin function MFBO results from four random seeds.

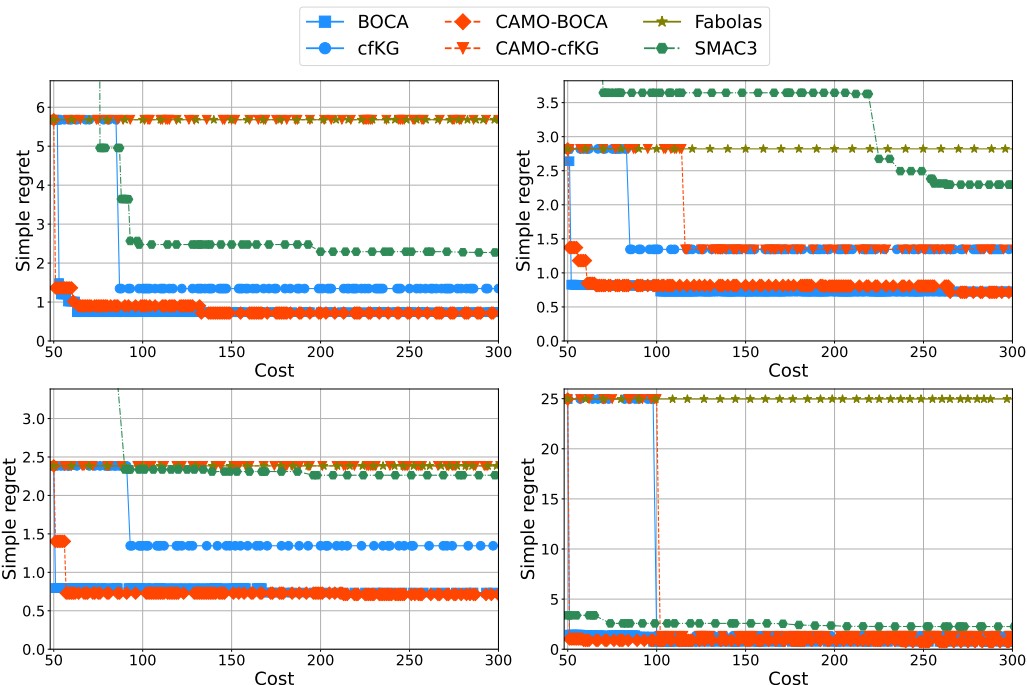

Figure H5: Forrester function MFBO results from four random seeds.

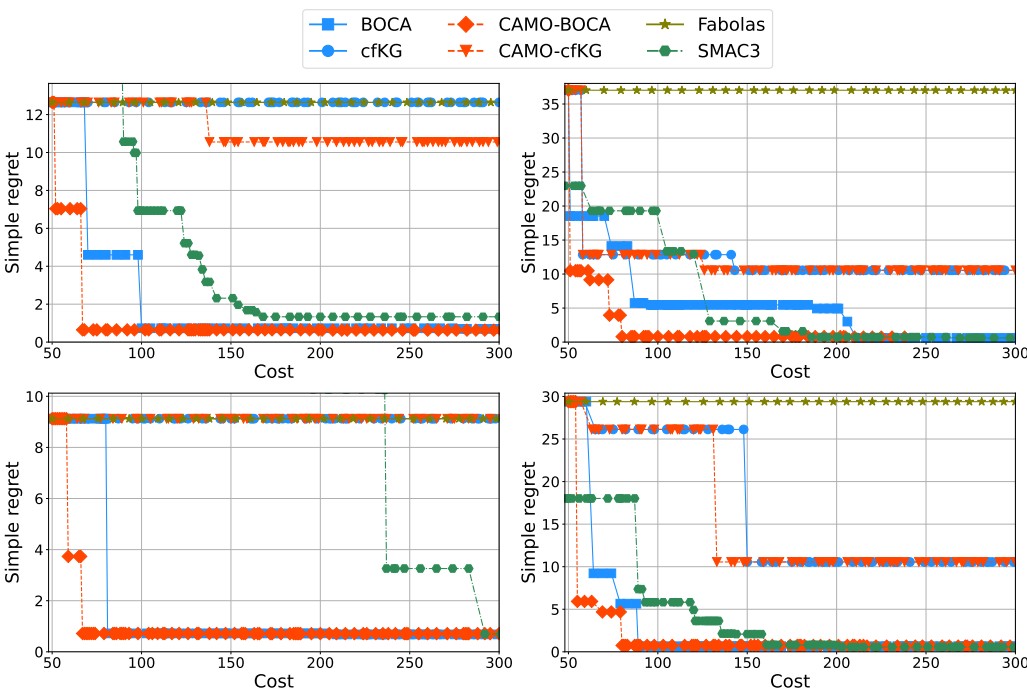

Figure H6: Bohachevsky function MFBO results from four random seeds.

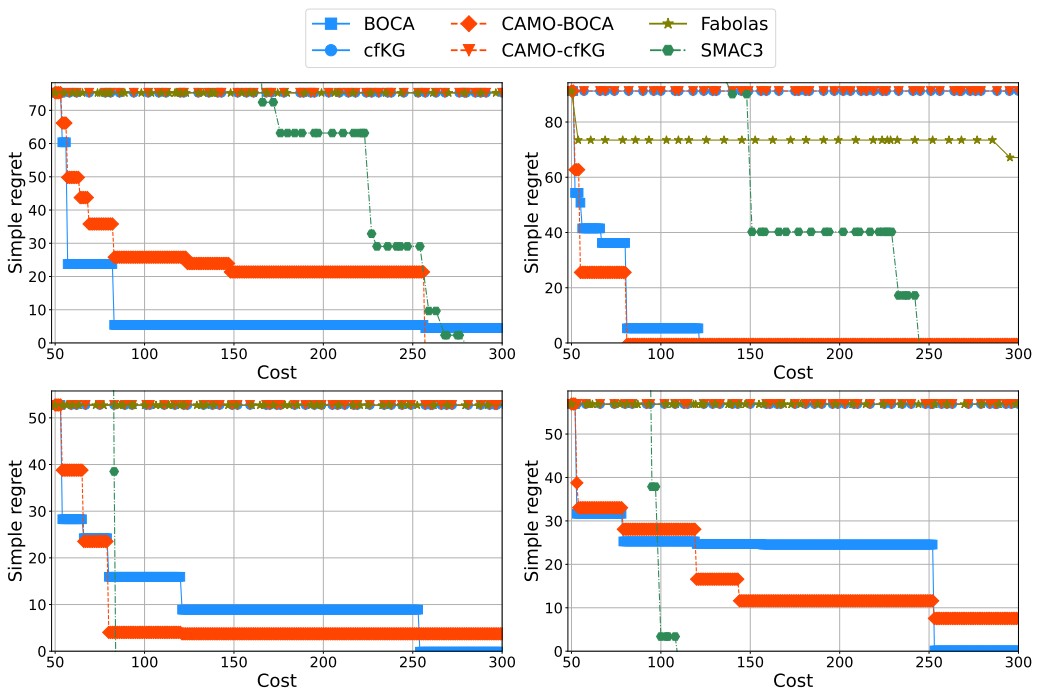

Figure H7: Borehole function MFBO results from four random seeds.

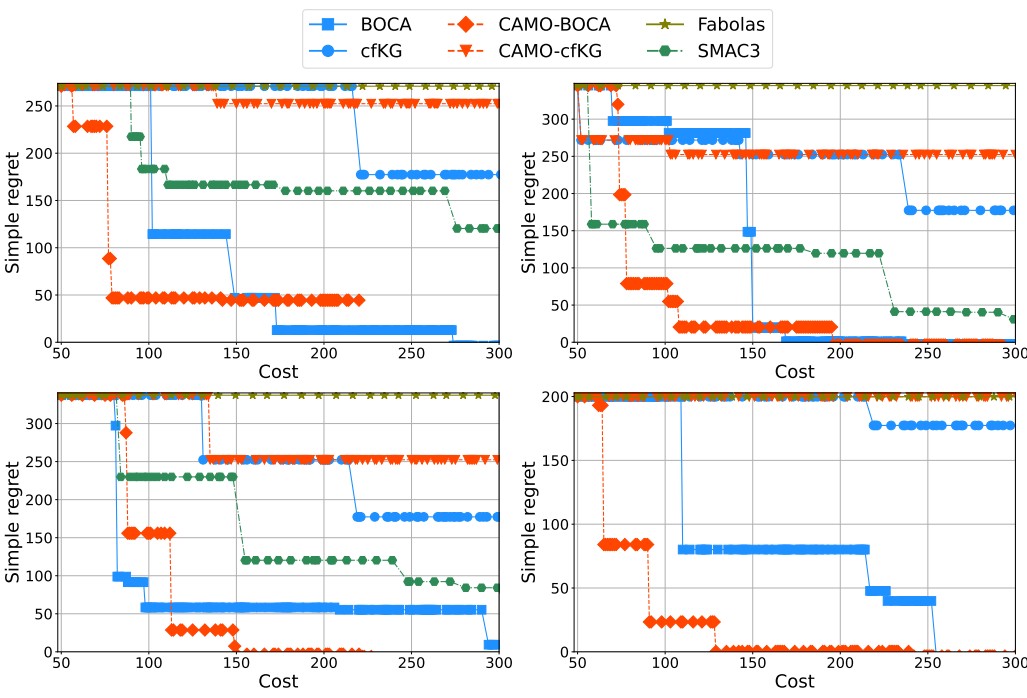

Figure H8: Colvile function MFBO results from four random seeds.

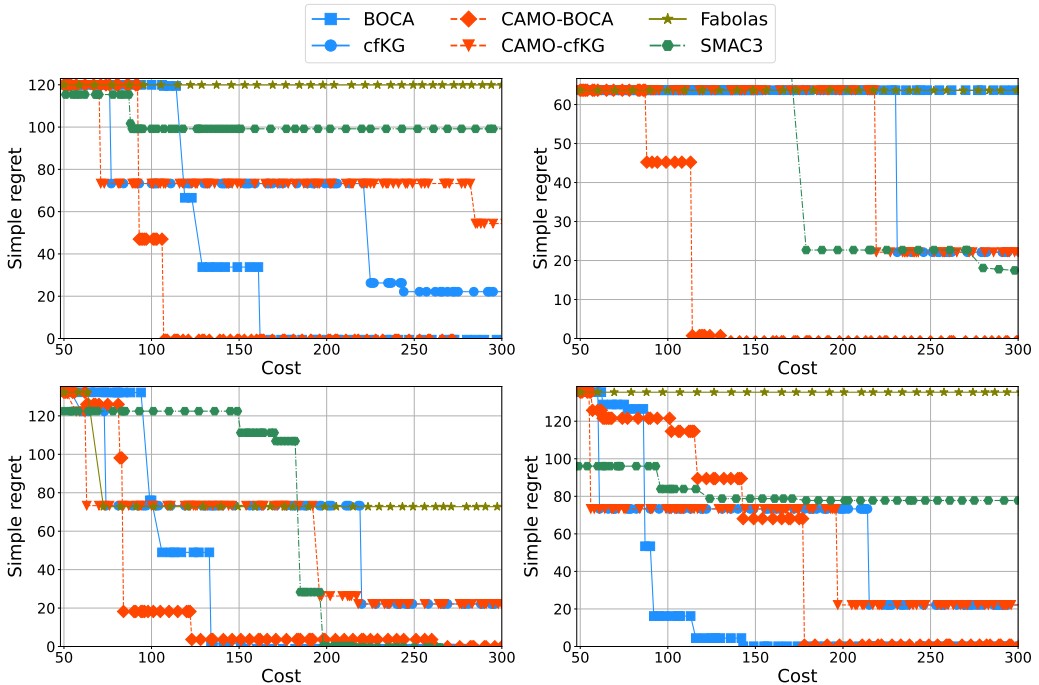

Figure H9: Himmelblau function MFBO results from four random seeds.

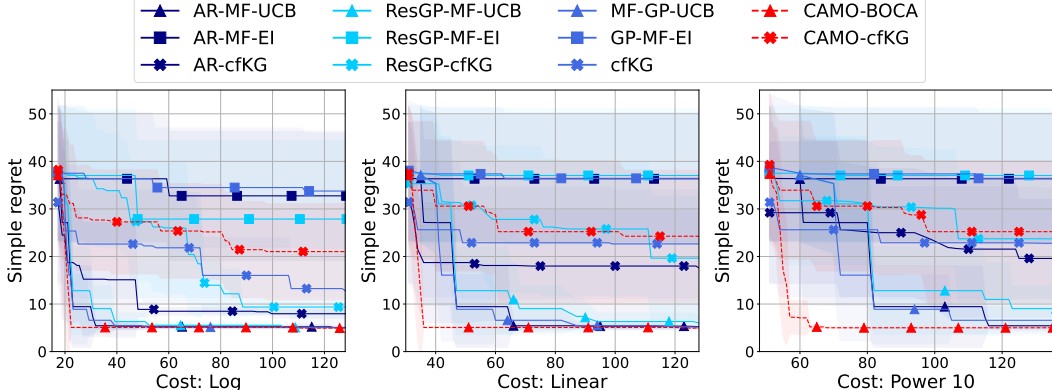

Figure H10: MFBO for the Forrester function with different cost functions.

# I  Synthetic Benchmarks

The definitions of the objective functions in the synthetic benchmark experiments are given below.

## Park Function

The input is two dimensional, $\mathbf{x} \in [0, 1]^2$ and there is one fidelity index $t \in [0, 1]$

$$f(\mathbf{x}, t) = \frac{(x_1 + 0.5t)^2 + (x_2 + 0.5t)^2}{2}. \tag{G14}$$

## Currin Function

The input is two dimensional, $\mathbf{x} \in [0, 1]^2$ and there is one fidelity index $t \in [0, 1]$

$$f(\mathbf{x}, t) = \left[1 - \exp\left(-\frac{1}{2x_2 t}\right)\right] \frac{2300x_1^3 + 1900x_1^2 + 2092x_1 + 60}{100x_1^3 + 500x_1^2 + 4x_1 + 20}. \tag{G15}$$

## Branin Function

The input is two dimensional, $\mathbf{x} \in [0, 1.5]^2$ and there is one fidelity index $t \in [0, 1]$

$$f(\mathbf{x}, t) = [x_2 - [b - 0.1 \times (1 - t)]x_1^2 + cx_1 - r]^2 + 10(1 - s)\cos(x_1) + 10, \tag{G16}$$

in which $b = \frac{5.1}{4\pi^2}$, $c = \frac{5}{\pi}$, $r = 6$ and $s = \frac{1}{8\pi}$.

## Nonlinear Sin Function

This is a two-level multi-fidelity function where input is one dimensional $x \in [0, 1.5]$. Low and high fidelity are given by

$$\begin{aligned} f_{\text{low}}(x) &= \sin(8\pi x), \\ f_{\text{high}}(x) &= (x - \sqrt{2})f_{\text{low}}(x)^2. \end{aligned} \tag{G17}$$

The low and high fidelity functions are shown in Figure I1.

## Forrester Function

Two-level multi-fidelity function with input $x \in [0, 1.5]$

$$\begin{aligned} f_{\text{low}}(x) &= 0.5f_{\text{high}}(x) + 10(x - 0.5) + 5, \\ f_{\text{high}}(x) &= (6x - 2)^2 \sin(12x - 4). \end{aligned} \tag{G18}$$

The low and high fidelity functions are shown in Figure I1.

## Bohachevsky Function

Two-level multi-fidelity function with input $\mathbf{x} \in [-5, 5]^2$

$$\begin{aligned} f_{\text{high}}(\mathbf{x}) &= x_1^2 + 2x_2^2 - 0.3\cos(3\pi x_1) - 0.4\cos(4\pi x_2) + 0.7, \\ f_{\text{low}}(\mathbf{x}) &= f_{\text{high}}(0.7x_1, x_2) + x_1 x_2 - 12. \end{aligned} \tag{G19}$$

## Borehole Function

Two-level multi-fidelity function with input variables $r_w \in [0.05, 0.15]$, $r \in [100, 50000]$, $T_u \in [63070, 115600]$, $H_u \in [990, 1110]$, $T_l \in [63.1, 116]$, $H_l \in [700, 820]$, $L \in [1120, 1680]$, $K_w \in [9855, 12045]$

$$f_{\text{high}}(\mathbf{x}) = f_b(\mathbf{x}, 2\pi, 1), \quad f_{\text{low}}(x) = f_b(\mathbf{x}, 5, 1.5)$$

$$f_b(\mathbf{x}, A, B) = \frac{AT_u(H_u - H_l)}{\log\left(\dfrac{r}{r_w}\right)\left(B + \dfrac{2LT_u}{\log\left(\dfrac{r}{r_w}\right) \cdot r_w^2 K_w} + \dfrac{T_u}{T_l}\right)}. \tag{G20}$$

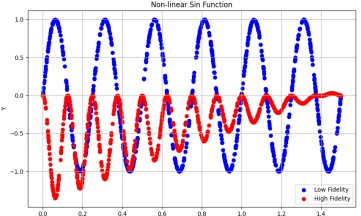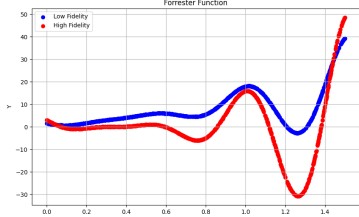

Figure I1: Low and high fidelity nonlinear sin function (left) and Forrester function (right).

## Colville Function

Two-level multi-fidelity function where input is four dimensional, $\mathbf{x} \in [-1, 1]^4$ and coefficient $A <= 0.68$, where low and high fidelity is given by:

$$
\begin{aligned}
f_{\text{high}}(\mathbf{x}) = {} & 100(x_1^2 - x_2)^2 + (x_1 - 1)^2 + (x_3 - 1)^2 + 90(x_3^2 - x_4) \\
& + 10.1((x_2 - 1)^2 + (x_4 - 1)^2) + 19.8(x_2 - 1)(x_4 - 1), \\
f_{\text{low}}(\mathbf{x}) = {} & f_{\text{high}}(A^2(x_1, x_2, x_3, x_4)) - (A + 0.5)(5x_1^2 + 4x_2^2 + 3x_3^2 + x_4^2).
\end{aligned} \tag{G21}
$$

## Himmelblau Function

Two-level multi-fidelity function with input $\mathbf{x} \in [-1, 1]^2$ and coefficient $A \leq 0.68$

$$
\begin{aligned}
f_{\text{high}}(\mathbf{x}) &= (x_1^2 + x_2 - 11)^2 + (x_2^2 + x_1 - 7)^2, \\
f_{\text{low}}(\mathbf{x}) &= f_{\text{high}}(0.5x_1, 0.8x_2) + x_2^3 - (x_1 + 1)^2.
\end{aligned} \tag{G22}
$$

# J   Details of Real-World Applications

## J.1   Mechanical Plate Vibration Design

The objective is to optimize the design of a 3-D simply supported, square, elastic plate with dimensions $10 \times 10 \times 1$[m], as illustrated in Fig. J1. The primary goal is to identify materials that maximise the fourth vibration mode frequency, thereby minimising the risk of resonance-induced damage caused by interactions with other components. The material properties under consideration include the Young's modulus (ranging from $1 \times 10^{11}$ to $5 \times 10^{11}$[Pa]), the Poisson ratio (between 0.2 and 0.6), and mass density (varying from $6 \times 10^3$ to $9 \times 10^3$[kgm$^{-3}$]). The plate is discretised using quadratic tetrahedral elements, as depicted in Fig. J1.

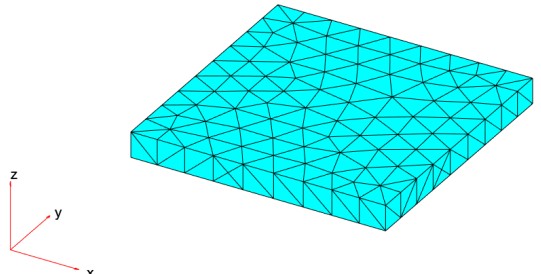

Figure J1: Quadratic tetrahedral element discretisation of the plate ($h_{\max} = 1.2$).

## J.2   Thermal Conductor Design

The second application focuses on optimising the design of a thermal conductor, as shown in Fig. J2(a). The heat source is located on the left side of the conductor, with the temperature initially at 0 and increasing to 100 degrees within 0.5 seconds. The heat transfer occurs through the conductor towards the right end. The conductor dimensions and material properties, including thermal conductivity and mass density (both equal to 1), are fixed. To facilitate installation, a hole must be bored into the center of the conductor. The top, bottom, and inner surfaces of the hole are thermally insulated, preventing heat transfer across these boundaries. The

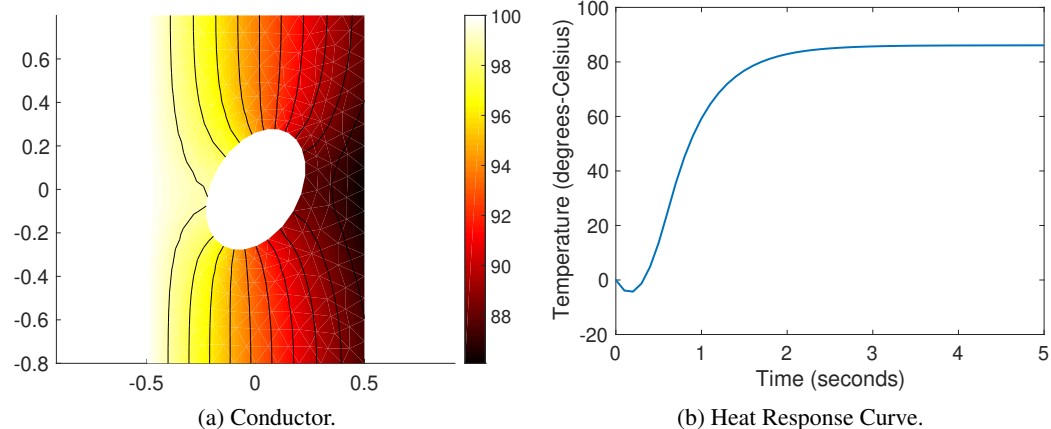

(a) Conductor.            (b) Heat Response Curve.

Figure J2: (a) Snapshot of the thermal conductor temperature solution; (b) the heat response curve on the right edge.

size and angle of the hole play a crucial role in determining the rate of heat transfer. In general, the hole is an ellipse, characterised by three parameters: the semi-minor and semi-major axes, and the orientation angle. The objective is to minimise the time required for the temperature at the right end to reach 70 degrees, thus maximising the heat conduction rate from left to right. To evaluate the time taken to reach the target temperature, the conductor is discretised using quadratic tetrahedral elements. The finite element method is then applied to solve the problem, yielding a response heat curve at the right edge, as illustrated in Fig. J2(b). By analysing this response curve, the time at which the temperature reaches 70 degrees can be determined.

