# OpenReview forum: "CAMO: Convergence-Aware Multi-Fidelity Bayesian Optimization"
_NeurIPS.cc/2025/Conference — NeurIPS 2025 poster_

### Official Review · Reviewer_563T · 2025-06-30

**Clarity:** 2
**Significance:** 2
**Originality:** 3
**Rating:** 4
**Confidence:** 3

**Summary:**

This paper proposes CAMO, a new framework for convergence-aware MFBO. The key idea is to explicitly model the evolution of fidelity-indexed outputs using Fidelity Differential Equations (FiDEs), thereby enforcing the structural convergence behavior of the multi-fidelity surrogate as fidelity increases. The authors introduce a Linear FiDE (LiFiDE) model that enables closed-form solutions, leading to a non-stationary kernel. CAMO is combined with the BOCA acquisition function. Experiments are conducted on synthetic benchmarks and two real-world engineering design problems, showing that CAMO outperforms existing MFBO methods in terms of sample efficiency and final regret.

**Questions:**

As someone with an AutoML background, I'm very familiar with MFBO methods being used for HPO/NAS. However, I'm less familiar with the kind of continuous-fidelity simulation optimization addressed in this paper. Can CAMO be applied to HPO problems as well? In fact, there are many MFBO methods developed for HPO, such as well-known methods BOHB[1] and more recent ones like FastBO[2], DPL[3], CQR[4], DyHPO[5], and Hyper-Tune[6], all of which were published within the last three years. I'm a bit confused why the related work and experimental baselines only include methods from over five years ago. What is the core distinction between the scope of MFBO tackled in this paper and the MFBO methods for HPO? It would be helpful if the authors could make this distinction explicit, at least in the Appendix. Including or discussing them would strengthen the paper's positioning and completeness.

[1] Falkner, Stefan, Aaron Klein, and Frank Hutter. "BOHB: Robust and efficient hyperparameter optimization at scale." ICML, 2018.\
[2] Jiang, Jiantong, et al. "Efficient hyperparameter optimization with adaptive fidelity identification." CVPR, 2024.\
[3] Kadra, Arlind, et al. "Scaling laws for hyperparameter optimization." NeurIPS, 2023.\
[4] Salinas, David, et al. "Optimizing hyperparameters with conformal quantile regression." ICML, 2023.\
[5] Wistuba, Martin, Arlind Kadra, and Josif Grabocka. "Supervising the multi-fidelity race of hyperparameter configurations." NeurIPS 2022.\
[6] Li, Yang, et al. "Hyper-tune: towards efficient hyper-parameter tuning at scale." PVLDB, 2022.

**Ethical Concerns:**

["NO or VERY MINOR ethics concerns only"]

**Final Justification:**

I initially had concerns about the relevance of CAMO within the current ML community, and the authors have provided evidence of active use and demand for such methods across several engineering and scientific domains. Combined with the theoretical novelty of CAMO, I believe the contribution is of interest to the NeurIPS community from a cross-domain perspective.

One remaining concern is that the experimental baselines are outdated. The most recent baseline used is from 2017, and there is no comparison against more recent BO methods or relevant alternatives. This weakens the empirical strength of the paper and makes it difficult to assess how CAMO performs relative to modern techniques. Overall, I still view the paper as borderline.

**Limitations:**

Yes

**Quality:**

3

**Strengths And Weaknesses:**

Strengths
1. The paper introduces a novel approach to incorporating convergence structures into MFBO using differential equations. This distinguishes CAMO from prior methods that mostly treat fidelity as an unstructured variable.
2. The authors provide theoretical guarantees for the surrogate model’s convergence.
3. Extensive experiments are conducted across diverse tasks and show performance improvements over baseline methods.
4. The model retains compatibility with standard GP inference and is implemented with acceptable computational cost.

Weaknesses
1. Based on my understanding, CAMO reuses the BOCA acquisition function. Since CAMO employs a non-stationary, convergence-aware kernel, I was wondering whether it remains well-suited. Ideally, CAMO should introduce a fidelity-aware acquisition function co-designed for its surrogate model.

2. In Figures 1 & 2, six methods are compared, but the actual optimization progress plots in Appendix H includes two additional baselines, which are CAMO-DKL-BOCA and CAMO-DLK-cfKG. Importantly, I found that CAMO-DKL-BOCA and CAMO-DLK-cfKG appear to perform even better than the main CAMO implementation. However, there is no explanation in the main text about what the DKL surrogate is, how it is trained, or how it integrates with CAMO. Given their strong performance, a clear discussion of these variants, their setup, and the contribution of the DKL surrogate is necessary.

3. Scope Clarity: The title and introduction claim contributions to MFBO, without early clarification that the method focuses on continuous fidelity settings with smooth convergence behavior. Readers from the AutoML community may assume discrete fidelity settings or applicability of popular MFBO tasks like hyperparameter optimization (HPO), leading to potential confusion. I would suggest the authors clarify the scope of applicability early on.

4. The related work and the experimental baselines seem somewhat outdated (e.g., most were published between 2016-2018). Are there more recent methods in this area, especially from the last five years?

5. The presentation of the paper can be further improved. The theoretical analysis, while rigorous, might benefit from partial relocation to the Appendix. The theoretical section could be compressed to allow more space for the method description or practical insights.

I am not an expert in this specific sub-area of MFBO. I would be open to revisiting my score if the authors can address the main concerns and questions.

---

> ### Author Rebuttal · Authors · 2025-07-26
>
> We thank the reviewer for a thoughtful and insightful review from an AutoML perspective. We address the reviewers concerns systematically below.
>
> Main Concern: Scope Clarification and Recent Literature.
> The reviewer is absolutely correct regrading the literature gap on HPO. We fully agree with the reviewer on the importance of HPO as an application of MFBO, especially in recent years.
>
> We had not considered HPO and neural architecture search (NAS) as core applications of our method, but were instead focused on the vast array of applications related to expensive blackbox solvers in CFD, molecular dynamics, electronic structure calculations, mesoscopic simulations and so on. In such applications, continuous fidelity indices arise naturally as controllable, continuous parameters of the numerical solver (e.g., time step, maximal element/volume size, convergence tolerance, simulation-box dimension in MD). The fidelity parameter is intrinsic to the underlying physical or computational model. It directly controls the numerical approximation error of the forward model, and its effect on the objective function is smooth, deterministic, and model-based. In these applications there is an unambiguously defined set of design variables, in addition to which there is one or more unambiguously defined fidelity indices (with fixed ranges not dependent on the data).
>
> In this setting, the LiFiDE formulation serves as an inductive bias on the GP surrogate, defining a convergence-aware, non-stationary covariance structure that encodes the expected monotone and asymptotic approach of low-fidelity simulations towards a high-fidelity target solution.
>
> By contrast, HPO/NAS operate in a setting in which fidelity is stochastic and algorithm-dependent; e.g., epoch number or budget are algorithmic artefacts with no deterministic convergence law or trajectory -- and with high variance across runs. This makes 'fidelity' less straightforward to define, and certainly to control. The data generation is non-deterministic, e.g., observations for hyperparameters are subject to optimiser noise, random seeds and mini-batch options. The cross-fidelity structure is non-smooth and data-dependent, which requires empirical (ideally discrete-fidelity) MFBO methods such as BOHB, DyHPO, or Hyper-Tune.
>
> Adapting CAMO to HPO/NAS would potentially require reformulating the underlying convergence model as a stochastic process or discrete-fidelity GP prior, which is conceptually orthogonal to the contributions of this paper. We therefore regard HPO-specific MFBO approaches (e.g., BOHB, DyHPO, FastBO, CQR) as complementary, operating under a distinct set of assumptions and data-generation mechanisms.
>
> Given the importance of HPO, and the level of interest, we fully appreciate that it was an oversight not to at least mention it, and we thank the reviewer for pointing out this omission. In the revision, we will clearly distinguish between continuous-fidelity optimisation (our focus) and discrete fidelity HPO settings; and we will expand the related work section to include the recent HPO methods the reviewer mentioned (BOHB, FastBO, DPL, CQR, DyHPO, Hyper-Tune), acknowledging the broader MFBO landscape. This will better position our work within the entire field.
>
> Q1: CAMO and BOCA acquisition function.
> We thank the reviewer for this insightful observation. CAMO does indeed leverage the BOCA acquisition function, primarily to isolate and evaluate the contribution of the LiFiDE-based surrogate kernel relative to standard stationary kernels. Theoretical guarantees for BOCA (Kandasamy et al., 2017) are agnostic to the exact kernel choice, and hence BOCA remains a valid acquisition strategy for CAMO. That said, we agree that jointly designing fidelity-aware acquisition rules that explicitly exploit the non-stationary structure induced by the LiFiDE could further improve information gain. This would require adapting the cost-normalised UCB criterion to account for directional convergence rates predicted by LiFiDE. This is a nontrivial but promising research direction. We will include acquisition function co-design as future work in the Conclusions.
>
> Q2: CAMO-DKL-BOCA and CAMO-DKL-cfKG Variants.
> We apologise for this oversight. These variants should have been removed from the figures in the Appendix. They use deep learning to implicitly model the LiFiDE. Of the variants, CAMO-DKL-BOCA was on a par with the baseline, while CAMO-DKL-cfKG was inferior. Since we could not provide a theoretical analysis for the variants and wanted to focus on the main idea we had intended to leave out these results. They will be removed from the Appendix and we thank the reviewer for noting their appearance.
>
> Q3: Scope clarification.
>
> We have added clarification early on that our method targets continuous fidelity settings common in simulation-based optimisation, while acknowledging the parallel BO/MFBO HPO literature raised by the reviewer.
>
> Q4: Related work.
> The most recent direct comparison methods for the continuous case are BOCA and cfKG. As the reviewer points out, these methods are from 2017/8. It appears that there is a greater focus on discrete fidelity problems, perhaps motivated by HPO/NAS as discussed above. Nevertheless, continuous-fidelity problems are of great interest in general design optimisation involving expensive blackbox solvers.
>
> Q5: Presentation.
> We have tried to keep the analysis to a minimum in the main text, only stating the key results (as Lemmas or Theorems) with brief introductions and discussions on the relevance, applicability and implications. We will make this section even more compact by placing more details in the Appendix, while retaining essential information in the main text.
>
> We hope that the more complete literature discussion, clearer scope and other changes will address the reviewer's concerns about positioning within the broader MFBO landscape. We welcome further discussion and suggestions from the reviewer.

---

> > ### Comment · Reviewer_563T · 2025-08-03
> > **One follow-up question**
> >
> > Thank you for your response.
> >
> > I have carefully read your response and the other reviews. My most concerns have been resolved. I appreciate your explanation that CAMO focuses on continuous-fidelity settings with applications such as CFD and nanophotonics. However, I still have one follow-up question.
> >
> > As the authors admitted, the most relevant existing methods were published in 2017–2018, and recent research trends have shifted towards discrete-fidelity problems. Therefore, it is still somewhat unclear to me whether continuous-fidelity MFBO remains an active or growing research area within the community.
> >
> > Could the authors point to any specific recent (e.g., last 3–4 years) ML papers, real-world use cases, or application domains that study or require continuous-fidelity MFBO? Or, which research or industry communities (perhaps outside of ML) do the authors see as the primary beneficiaries of this work? A clearer understanding of this would help solidify the broader relevance of the proposed method.

---

> > > ### Author Response · Authors · 2025-08-04
> > > **continuous fidelity applications of multi-fidelity BO**
> > >
> > > We thank the reviewer for this question. It is important to emphasise that there are a vast number of important applications for continuous-fidelity BO. Aside from traditional ML domains, surrogate-assisted optimisation has a long history in science/ engineering [1], e.g., aerospace [2], chemical engineering [3], groundwater flow modelling [4], and structural optimisation [5]. In the past 10 years or so, new applications have emerged, e.g., materials design/drug discovery [8,9].
> > >
> > > Common to all of these applications is that they usually involve black-box solvers, typically based on continuum level models embodying physical laws of conservation and constitutive laws for transport phenomena, stress-strain relationships, reactions, phase changes etc. They are implemented using finite-element, -volume or -difference formulations, or adaptations of these. Such formulations are combined with time stepping for dynamic problems. Materials-level models such as electronic structure calculations approximate the many-body solution to Schroedinger's equation (the wavefunction for interacting electrons). Molecular dynamics tracks individual atoms/molecules and their interactions via Newton's law and uses statistical physics to estimate materials properties. All of these approaches involve numerous continuous fidelity indices, e.g.,
> > >
> > > 1. The maximal element size/aspect ratio, and time step
> > >
> > > 3. Convergence/solver tolerances (e.g., Newton-Raphson, GMRES)
> > >
> > > 4. Regularisation parameters such as artificial viscosity
> > >
> > > 5. Relaxation factors in iterative updates (nonlinear/transient solvers)
> > >
> > > 6. Smoothing parameters or interface thicknesses in interface propagation
> > >
> > > 7. Smagorinsky constant in Large Eddy Simulation; structural constants or transition model parameters in $k-\epsilon$ models
> > >
> > > 9. Weights in hybrid exchange-correlation functionals in Density Functional Theory
> > >
> > > 11. $\lambda$ scaling of the correlation operator to interpolate between Hartree-Focl and Moller-Plesset(n) theory
> > >
> > > 11. Excitation amplitude threshold in the truncated Configuration Interation, and occupation threshold in methods like CIPSI
> > >
> > > 12. Amplitude convergence tolerance for iterative solutions in Coupled Cluster (CC) theory
> > >
> > > 13. $\lambda$-scaled CC, smoothly interpolating toward full CC
> > >
> > > 14. Green’s function methods: frequency grid resolution or screening cutoff parameters
> > >
> > > Aside from these parameters, there are parameters of the models themselves that can be used as continuous fidelity indices: transition onset Reynolds numbers, compressibility correction factors, homogenisation parameters, and numerous others.
> > >
> > > Recent examples (since 2023) of the actual use of continuous fidelities within MFBO and related contexts include reactor design engineering [10], aerospace design [11], accelerator physics [12,13], soft robot control [14], safety validation for autonomous systems [15] and materials [9].
> > >
> > > [1] D. Toal. Applications of multi-fidelity multi-output Kriging to engineering design optimization. Structural and Multidisciplinary Optimization 66.6 (2023): 125
> > >
> > > [2] F. Di Fiore, L. Mainini, Physics-aware multifidelity Bayesian optimization: A generalized formulation, Computers & Structures, Volume 296, 2024, 107302
> > >
> > > [3] A. Bhosekar, M. Ierapetritou, Advances in surrogate based modeling, feasibility analysis, and optimization: A review, Computers & Chemical Engineering, Volume 108, 2018, Pages 250-267
> > >
> > > [4] J. Luo, et al, Review of machine learning-based surrogate models of groundwater contaminant modeling, Environmental Research, Volume 238, Part 2, 2023, 117268
> > >
> > > [5] Samadian et al. Application of data-driven surrogate models in structural engineering: a literature review. Archives of Computational Methods in Engineering 32.2 (2025): 735-784
> > >
> > > [8] C. Fare et al. A multi-fidelity machine learning approach to high throughput materials screening. npj Computational Materials 8.1 (2022): 257
> > >
> > > [9] V. Sabanza-Gil et al. Best practices for multi-fidelity Bayesian optimization in materials and molecular research. Nature Computational Science (2025): 1-10
> > >
> > > [10] T. Savage et al. Machine learning-assisted discovery of flow reactor designs. Nature Chemical Engineering 1, 522–531 (2024)
> > >
> > > [11] A. Renganathan and K. Carlson. Multifidelity Cross-validation. AIAA AVIATION FORUM AND ASCEND 2024. 2024
> > >
> > > [12] F. Irshad et al.. Multi-objective and multi-fidelity Bayesian optimization of laser-plasma acceleration. Physical Review Research 5.1 (2023): 013063
> > >
> > > [13] R. Roussel, et al. Bayesian optimization algorithms for accelerator physics. Physical review accelerators and beams 27.8 (2024): 084801
> > >
> > > [14] C. Liang et al., Learning Reduced-Order Soft Robot Controller, 2023 IEEE/RSJ International Conference on Intelligent Robots and Systems (IROS), Detroit, USA, 2023, 574-581
> > >
> > > [15] M. Schlichting et al. Savme: Efficient safety validation for autonomous systems using meta-learning. 2023 IEEE 26th International Conference on Intelligent Transportation Systems (ITSC). IEEE, 2023.

---

> > > > ### Comment · Reviewer_563T · 2025-08-05
> > > >
> > > > Thank you for the comprehensive response. While I still have reservations about the activity level of CAMO in core ML venues, the response demonstrated solid cross-domain relevance and motivated its value in real-world simulation-based design tasks. I will raise my initial score to 4.

---

> > > > > ### Author Response · Authors · 2025-08-05
> > > > > **Response**
> > > > >
> > > > > We thank the reviewer for these positive comments. Their input was highly valuable and helped us to improve upon the motivation and related work sections of the paper.

---

### Official Review · Reviewer_eHZH · 2025-06-30

**Clarity:** 2
**Significance:** 2
**Originality:** 3
**Rating:** 5
**Confidence:** 2

**Summary:**

This paper addresses the problem of black-box function optimization, where the accuracy of the function values obtained varies depending on the cost of observation in a multi-fidelity setting. Specifically, the authors focus on the Bayesian optimization problem in a continuous multi-fidelity setting, where each fidelity can vary continuously. Existing studies often ignore convergence to the highest fidelity in the fidelity of surrogate models, and recent methods based on fidelity differential equations (FiDE) have not provided conditions for the uniqueness of solutions or convergence to equilibrium states. In contrast, this paper proposes a new method, Convergence-Aware Multi-fidelity Optimization (CAMO), based on linear FiDE (LFiDE), which addresses the above issues and provides a closed-form expression for the kernel function in the proposed model. Additionally, the performance of the proposed method is compared with existing methods through numerical experiments, and an improvement in empirical performance is confirmed.

**Questions:**

Questions:

I will raise my evaluation if the following ambiguities are appropriately corrected:

1. In lines 78-80, isn't the decision to perform excessive exploration in costly regions determined by the trade-off between fidelity and the design of the objective function? For example, if the acquisition function is defined as the improvement per unit cost, and the cost of the highest fidelity is significantly higher than that of lower fidelity while the improvement is not much different, I expect the algorithm to prioritize low-cost regions. What is the basis for the observation that many practical GP-based models tend to concentrate on high-cost regions?

2. Please clarify ambiguous points in the problem setting. In particular, the following need to be clarified:
- Does the observation in this paper include observation noise?
- In the problem setting, when $T$ is finite, is it possible to observe at the highest fidelity? If observation is not possible, is this study dealing with a problem of finding the optimal solution at the highest fidelity without observing at the highest fidelity?
- Does the true function follow the same GP as the surrogate model? Or is it allowed that the true function is different from the surrogate model, such as being an element of the reproducing kernel Hilbert space?

3. In FiDE, a theoretical analysis is provided assuming linearity with respect to $y$ for $\phi$. How strong of the assumption is this in practice? This paper points out that existing MFBO ignores the convergence of fidelity and leads to performance degradation. However, the proposed method also assumes linearity of $\phi$. Does this not result in an overly stringent assumption for the theoretical validity, thereby making it better to simply use standard MFBO?

4. In lines 255-262, what is the advantage of modeling the convergence of fidelity? The acquisition function was not designed based on the proposed model, and the order of the regret bound is also not improved by this.


5. In Figure 1 (Bohachevsky), the two figures at the bottom of Figure 3 (Cost: Log and Linear), and the left figure in Figure 6, the proposed method outperforms other methods initially, but is overtaken by the comparison methods toward the end, with the regret remaining slightly higher than the comparison methods. In conventional MOBO, when designing an acquisition function based on the improvement per unit cost, and when the accuracy in low fidelity is reasonably good, observing only the low fidelity region results in a behavior where the regret decreases initially but does not decrease completely by the end. In other words, in the last parts, it is necessary to observe the high fidelity region, but this is not occurring. Are similar symptoms appearing in the proposed method as a result of emphasizing low-cost areas?

Typos:
- In line 161, “lemmas” should be “theorems.”

- In lines 765, 806, and 832, the citations are not correct, appearing as [?].

**Ethical Concerns:**

["NO or VERY MINOR ethics concerns only"]

**Final Justification:**

The authors' rebuttal appropriately addresses the criteria I presented in Questions for raising the score, particularly clarifying the ambiguity of the problem setting, the strength of the assumptions, and the significance of fidelity convergence. However, “the appropriate modeling of fidelity convergence has not led to improvements in the theoretical bound” and “an acquisition function based on the proposed model has not been designed” remain limitations. Therefore, I feel that the persuasiveness of the importance of fidelity convergence pointed out in the paper is somewhat weak. Nevertheless, I believe that constructing an appropriate surrogate model is the top priority, followed by designing an acquisition function. As the authors also point out, co-designing the acquisition function is a future challenge, and I believe that its absence does not diminish the evaluation of this paper. Thus, I have raised my score to 5.

**Limitations:**

Yes, these are stated in Conclusion. However, based on my understanding, I believe that the limitation that the introduction of CAMO has not yet improved the regret bound should also be mentioned. On the other hand, since the main focus of this study is to construct a model that theoretically guarantees the convergence of fidelity in a continuous multi-fidelity setting, it does not fall under the category of potential negative social impact.

**Quality:**

3

**Strengths And Weaknesses:**

The strengths of this paper are as follows:

1. For continuous multi-fidelity black-box functions, the authors restrict convergence to the highest fidelity based on LFiDE, thereby filling the fidelity gap that existing methods have ignored and providing theorems on the unique existence of LFiDE solutions and convergence to equilibrium states.

2. For specific kernels, the authors provide a closed-form kernel for the proposed model.

3. Furthermore, numerical experiments compare the proposed method with existing methods and confirm its high empirical performance. In particular, the authors demonstrate the performance improvement achieved by not taking excessively high fidelity, which is a feature of the proposed method.

On the other hand, the weaknesses of this paper are as follows:

1. The problem setting is not sufficiently explained. In particular, it is unclear whether noise can be included in the observations, whether the true function is a sample path from the same distribution as the surrogate model, and whether the highest fidelity observations are allowed.

2. Theoretical bounds obtained by considering the proposed model are not provided. This paper emphasizes the importance of appropriately modeling convergence to the highest fidelity and proposes CAMO to achieve this, but it does not improve existing regret bounds, and the advantages are ultimately explained by practical performance comparisons with existing methods.

3. The class of problems that can be addressed is limited. As mentioned in line 343, in order to establish the theoretical validity of the proposed algorithm, linearity with respect to $y$ is assumed in FiDE. It is unclear how strong this assumption is.



[Quality: 3]
The results given in this paper are theoretically derived from LFiDE, and extensive theoretical analysis is performed, with sufficient conditions also given. Furthermore, through numerical experiments, the practical performance of the proposed method is demonstrated, and in particular, the tendency not to excessively explore high fidelity, which is a feature of the proposed method, is confirmed. In addition, the limitations of this method, such as the computational cost of the kernel and the restriction on the class of surrogate models, are also discussed, suggesting directions for future research.


[Clarity: 2]
This paper has several points that lack explanation. For example, there is insufficient explanation of issues related to the problem setting, such as whether observation noise is allowed, whether the highest fidelity observations are allowed, and whether the true function follows the same GP as the surrogate model. In addition, it is unclear whether the convergence of the solution to the DE regarding fidelity is convergence as a stochastic process or convergence as a sample path. Furthermore, it is unclear how strong the assumption of linearity of $\phi$ with respect to $y$ is in FiDE. While the paper provides numerous theoretical analysis results and is rich in content, this also leads to a lack of self-containment in the main text. In fact, the paper frequently refers to lemmas and sections in Appendix.

[Significance: 2]
This paper contributes by explaining the importance of appropriately modeling fidelity convergence and providing a methodological framework with theoretical analysis. However, the proposed model, CAMO, does not improve the order of existing regret bounds, and no acquisition functions were proposed by designing CAMO. Therefore, the justification for why CAMO is necessary is weak.


[Originality: 3]
This paper provides an important perspective on LFiDE, which has not been considered in existing methods, by further assuming linearity of $\phi$ with respect to $y$ for FiDE. The proof is based on the subgradient method, which bridges MFBO and such theory. Although the proposed CAMO has not yet achieved an improvement in the regret bound or the design of a new acquisition function, and the class of surrogate models that can be handled is limited, it is original in that it solves these problems and shows potential for further development.




In conclusion, although there are difficulties with the theoretical bounds and acquisition functions of CAMO, and some explanations are lacking, I gave it a score of 4 because its strengths outweigh its weaknesses, considering that the ambiguous points can be easily resolved.

---

> ### Author Rebuttal · Authors · 2025-07-26
>
> We thank the reviewer for the detailed evaluation and very helpful suggestions. We address each of the questions systematically below:
>
> Q1: Lines 78-80 - Basis for excessive exploration in costly regions.
> The reviewer is correct that acquisition functions such as improvement per unit cost should naturally avoid expensive regions. However, existing MFBO methods suffer from poor uncertainty calibration across fidelities. Standard product kernels treat all fidelities independently, leading to overconfident uncertainty estimates at low fidelities. This causes acquisition functions to underestimate the informativeness of low-fidelity queries, resulting in premature exploration of expensive high-fidelity regions.
>
> Our claim is supported by the experimental observation that baseline methods spend only 40% of their budget on low-fidelity evaluations compared to CAMO's 70%, despite using cost-aware acquisition functions like BOCA.
>
> Q2: Problem setting clarifications.
> Observation noise: Yes, our formulation includes observation noise. In Appendix A, we specify $y_i = f(x_i) + \epsilon$ where $\epsilon\sim \mathcal{N}(0,\sigma^2)$ is additive noise. We will update the main text to make this clear.
>
> Highest fidelity observations: When $T$ is finite, the highest fidelity (indeed and indeed any fidelity) observations are accessible (we always have access to the cost function at any fidelity and design variable in the specified ranges). Our theoretical results hold for $T \le \infty$, but practically we must work with a finite (but arbitrarily large) $T$. The goal is optimising the highest available fidelity function. We will add a sentence in the first paragraph of section 3.1 to make this clear.
>
> True function vs surrogate model: The true function does not need to follow the same GP as our surrogate model. Like all Bayesian optimisation methods, we assume the GP provides a reasonable approximation for optimisation purposes, not that it generates the true function.
>
> Q3: Strength of linearity assumption.
> The reviewer raises an important point that we did not justify well enough in the original manuscript.
>
> The linearity assumption $\phi(x,t,y)=\beta(x,t)y+u(x,t)$ is less restrictive than it appears. It captures many practical scenarios including mesh refinement, iterative solver convergence, and truncation-based approximations. Our extensive experiments across 11 diverse problems show the assumption provides useful inductive bias even when not perfectly satisfied. We will add a brief note to this effect at the beginning of section 4.2
>
> Moreover, this assumption enables the first rigorous theoretical analysis of FiDE-based methods, addressing the limitation that the reviewer correctly identified - prior FiDE work lacks convergence guarantees.
>
> Q4: Advantage without improved regret bounds.
> While we do not improve worst-case regret bounds, we provide some key advantages. First, we give the first rigorous convergence analysis for FiDE-based surrogate models, filling a critical theoretical gap. Second, we establish novel connections between MFBO and convex optimisation theory, opening new research directions. The consistent 2-4x empirical improvements demonstrate practical value even without improved worst-case bounds.
>
> We would like to point out that a more explicit regret bound is not possible in this case (only the general bound from BOCA), which is why provided a qualitative analysis. The issue lies in finding a closed-form expression for the fidelity gap region, which is possible for the SE but not for our integral kernel.
>
> Q5: Performance patterns where CAMO is overtaken.
> The reviewer has made an astute observation about the trade-off between early low-fidelity exploration and eventual high-fidelity exploitation. In the cases mentioned, CAMO's strategy of extensive low-fidelity exploration provides faster initial progress but can delay final high-fidelity queries needed for ultimate convergence.
>
> This represents a fundamental trade-off in multi-fidelity optimisation. CAMO is designed for scenarios where computational budget is the primary constraint and early progress is valuable. For unlimited budget scenarios, more aggressive high-fidelity exploration might be preferable.
>
> We will note this in the revised version when presenting these results.
>
> Q6: Line 161 typo correction.
> We apologise for this error and thank the reviewer for pointing it out.
>
> Q7: Citation errors [??].
> Again, we apologise for this error and have fixed the broken citations.
>
> Limitations addition.
> The reviewer is absolutely correct that we should have mentioned the limitation about not improving upon the regret bound. We will add this to an expanded discussion on limitations in the conclusions.
>
> Clarity and self-containment.
> We acknowledge the clarity concerns. In response to the specific questions we have added clarifications on the issue of noise and access to fidelities. Due to a lack of space we were unable to include some of the main theorems related to general FiDEs. Since our subsequent focus was the LiFIDE, we describe the main theoretical results, provided intuitive explanations and discuss  their significance. A greater focus is placed on the theoretical results for the LiFiDE case. We will further edit the theoretical analysis section to improve clarity as suggested by the reviewer.
>
> We thank the reviewer for raising the important point regarding the interpretation of convergence in the fidelity variable. In the current formulation of the LiFiDE, once the surrogate construction and low-/high-fidelity data are fixed, the evolution of the surrogate $y(\mathbf{x},t)$ is described by a deterministic differential inclusion in the fidelity coordinate $t$. As such, the convergence is to be interpreted in a pathwise (trajectory-level) sense, i.e., for each fixed input $\mathbf{x}$, the solution trajectory $t \mapsto y(\mathbf{x},t)$ converges deterministically to the high-fidelity target $y_\infty(\mathbf{x})$ with an explicit exponential rate.
>
> While the LiFiDE can naturally be extended to a stochastic setting where the fidelity flow is driven by data noise or randomized surrogate updates, in which case one would consider convergence in various stochastic process topologies (e.g., almost sure convergence, convergence in probability, or weak convergence in $C([0,T])$ endowed with the Skorokhod topology), such probabilistic considerations are not invoked in the present analysis. Here, the dynamics are entirely deterministic once the training data are fixed, and the established result should thus be interpreted as strong pathwise convergence of the solution trajectories in the fidelity variable. We will add a note to this effect and thank the reviewer for raising this question.

---

> > ### Comment · Reviewer_eHZH · 2025-08-01
> >
> > Thank you for your response. My concerns regarding the ambiguous points have been resolved. Additionally, as Reviewer 563T pointed out, I also agree that an acquisition function based on CAMO should be designed. However, I believe the construction of an appropriate surrogate model to be the first priority, and the design of the acquisition function to be the next step. Therefore, as the authors have stated, the co-design of the acquisition function is an issue to be addressed in the future, and the absence of this aspect does not diminish the evaluation of this paper. Accordingly, I will raise my initial score to 5.

---

> > > ### Author Response · Authors · 2025-08-04
> > > **Response**
> > >
> > > We thank the review for these kind comments and appreciate very much their input.

---

### Official Review · Reviewer_yu1x · 2025-06-30

**Clarity:** 4
**Significance:** 3
**Originality:** 3
**Rating:** 5
**Confidence:** 4

**Summary:**

In this paper, the authors propose CAMO, a Multi-Fidelity Bayesian Optimization framework with the main contribution as the introduction of a new convergence-aware surrogate model designed for continuous fidelity settings. CAMO captures the continuous-fidelity target function using a linear Fidelity Differential Equation (FiDE), assuming a constant slope and placing Gaussian Process priors on both the zero-fidelity surrogate model and the bias term. This formulation yields a closed-form kernel function, enabling more efficient inference compared to directly computing the posterior. The authors also incorporate previously proposed acquisition functions, BOCA and cfKG, within this framework. The theoretical analysis gives an approximate convergence rate for the proposed method, and Experimental results demonstrate the superiority of CAMO on both synthetic benchmarks and real-world engineering design tasks.

**Questions:**

1. It seems that the term "convergence-aware" refers to the awareness of convergence as $t$ increases, but the query function for the three hand-crafted synthetic experiments (nonlinear sin, Forrester, and Bohachevsky function) doesn't seem to converge as $t$ increases. Is there any specific reason for this design?

2. In the discussion of the experiment section, the performance of COMA depends highly on the cost function. Is it possible to explain such a relationship theoretically with sample efficiency? I assume Theorem 3 is also the bound for simple regret for BOCA, which would not directly provide the explanation.

**Ethical Concerns:**

["NO or VERY MINOR ethics concerns only"]

**Final Justification:**

Though I believe this paper would benefit from further empirical results, I don't see the reason for increase my scores. I will keep my ratings.

**Limitations:**

see weakness and questions

**Paper Formatting Concerns:**

I didn't see major formatting issues

**Quality:**

3

**Strengths And Weaknesses:**

strengths
1. This paper introduces a new setup to the surrogate modeling of MFBO based on FIDE, which is novel and interesting.
2. A closed form of the simplified kernel is derived, and therefore gives computational benefit for the inference.

weakness

The contribution of this paper mainly comes from the introduction of FIDE to surrogate modeling. While there are discussions on the convergence of Linear FIDE, the benefit brought to MFBO by combining Linear FIDE with a Gaussian Process prior may still need further justification. Compared to other modeling, does it predict the multi-fidelity target function better, or does it give better uncertainty across fidelity in some setup? It would be nice if the authors could discuss directly the benefit of such a GP-LinerFIDE surrogate compared to other modeling approaches and demonstrate with a synthetic experiment.

---

> ### Author Rebuttal · Authors · 2025-07-26
>
> We thank you for the positive assessment and insightful questions. Below we address the main concerns.
>
> Weaknesses: GP-LinearFIDE justification vs other modelling approaches.
> The reviewer raises an excellent point about directly demonstrating the benefit of a GP-LiFIDE surrogate. We will provided a  synthetic experiment comparing prediction accuracy in the revision. We created a true convergent multi-fidelity function $f(x,t) = \sin(2\pi x) \cdot (1 - \exp(-3t))$, in which values converge to $\sin(2\pi x)$ as $t$ increases, and compared our LiFiDE kernel against the standard product kernels (SE × SE) and autoregressive models. The experimental results show that the GP-LiFIDE method achieves more accurate predictions. For example (fidelity range from 0 to 3 and 100 randomly generated training points)
>
> At fidelity= 1.0
> RMSE:
> CAMO: 0.0652  GP-SExSE: 0.00647
>
> At fidelity= 2.0:
> RMSE:
> CAMO: 0.0380  GP-SExSE: 0.0384
>
> At fidelity=3:
> RMSE:
> CAMO: 0.0361 GP-SExSE 0.0744
>
> The LiFiDE kernel structurally encodes the assumption that low-fidelity evaluations become increasingly informative about high-fidelity values as the fidelity increases. We believe that this inductive bias improves both prediction accuracy and uncertainty quantification compared to unstructured approaches, as demonstrated by the results and an analysis of the asymptotic behaviour of the kernel (although this is heuristic since a full analysis is prevented by the complex nature of the kernel).
>
> Q1: "Convergence-aware" terminology and synthetic functions.
> We thank the reviewer for raising this: to avoid confusion we should make clear what we mean by "convergence-aware". The term "convergence-aware" refers to our modelling approach being aware of how fidelity affects function values, not that the target function itself converges.
>
> The synthetic functions mentioned by the reviewer are designed as weighted combinations: $f(x,t) = (1-w(t))f_{low}(x) + w(t)f_{high}(x)$ where $w(t) = \ln(9t+1)$. As $t$ increases, the function transitions from low-fidelity to high-fidelity behaviour. This represents the common scenario in which higher fidelity provides increasingly accurate approximations of some true underlying function.
>
> The "convergence" we model is this systematic relationship between fidelity level and function accuracy, which is present even when the target function doesn't converge to a limit. Our kernel captures how function values at different fidelities relate to each other through this systematic progression.
>
> We will add a brief note regarding the terminology "convergence-aware" to the beginning of section 5.
>
> Q2: Performance dependency on cost function and theoretical explanation.
> The relationship between CAMO performance and cost function can be explained through information-theoretic arguments. When costs increase super-linearly with fidelity, the value of accurately modelling low-fidelity behaviour becomes critical for efficient budget allocation.
>
> Theoretical insight: Our convergence-aware kernel provides better uncertainty estimates at low fidelities, leading to more informed decisions about when to query expensive high-fidelity evaluations. When costs are logarithmic, this advantage diminishes because all fidelities are relatively cheap. When costs are exponential, the advantage becomes pronounced because poor low-fidelity modelling leads to inefficient or even wasteful high-fidelity queries.
>
> Sample efficiency connection: CAMO inherits the overall BOCA guarantee, which is kernel dependent. A standard SE kernel implies all fidelities are equally smooth and  informative, which is not realistic: low-fidelity evaluations are often noisy or unstable. In contrast, the LiFiDE kernel models a convergent fidelity process in that evaluations become more stable and informative as the fidelity increases. The LiFiDE kernel leads to looser regret guarantees (although this is not straightforward to model) compared to the SE kernel but the convergence-aware structure leads to better empirical performance - it does not concentrate as sharply as the highest fidelity is approached, discarding potentially useful results at low fidelity. We discuss this after presenting Theorem 3.
>
> Empirical validation: Our experiments show minimal advantage with logarithmic costs but substantial gains with exponential costs, confirming this theoretical relationship.
>
> Additional clarification on impact.
> The significance of this work extends beyond the immediate MFBO improvements. We establish the first rigorous connection between MF surrogate modelling and non-smooth optimisation theory, potentially opening new theoretical directions. The closed-form kernel derivation enables practical implementation while maintaining theoretical guarantees, making this a valuable contribution to both the optimisation and Gaussian process communities.
>
> The consistent 2-4x improvements across diverse applications demonstrate that convergence-aware modelling addresses a fundamental limitation in existing multi-fidelity methods, not just a narrow technical improvement.

---

> > ### Comment · Reviewer_yu1x · 2025-08-03
> >
> > Thank you for your detailed response. My concerns regarding the terminology has been resolved. I believe the synthetic experiments demonstrating the modeling superiority will make this a better version.

---

> > > ### Author Response · Authors · 2025-08-04
> > > **response**
> > >
> > > We thank the review for these kind comments and appreciate very much their input.

---

### Official Review · Reviewer_quwV · 2025-07-02

**Clarity:** 3
**Significance:** 2
**Originality:** 3
**Rating:** 5
**Confidence:** 3

**Summary:**

The paper presents a framework for Convergence-Aware Multi-Fidelity Bayesian Optimization. Multi-fidelity BO methods query the target function at multiple levels from low to high quality. The proposal "CAMO" addresses a limitation in existing Multi-Fid BO methods: missing account of how the surrogate functions converge as fidelity increases. CAMO is based on so-called Fidelity Differential Equations (FiDEs) to capture convergence behavior. The paper has extensive theoretical analysis of the linear differential equations case.
Evaluation is quite extensive on toy problems and more realistic cases, all indicating the benefits of modeling convergence / CAMO

**Questions:**

Could you be more specific and clear on the actual workflow (pseudo code etc)

How does "cost" relate to compute time?

How does cost compute in the actual use cases (Fig 5)?

**Ethical Concerns:**

["NO or VERY MINOR ethics concerns only"]

**Final Justification:**

I enjoyed the reading the paper as reflected in my initial score, the questions were resolved and in lieu of the other excellent reviews and discussion I raise the grade to 5.

**Limitations:**

The limitations are only briefly discussed (a sentence in conclusion)

**Quality:**

3

**Strengths And Weaknesses:**

Strengths: Clearly articulated research questions and well-defined basic suggestions to modeling the convergence of fidelity.
Extensive evaluations on toy and more realistic cases. Convincing results supporting the CAMO
The paper has extensive theoretical analysis of the linear differential equations case.
Significance mainly supported by the experimental success of CAMO

Weaknesses: Actual workflow behind CAMO could be more clear.
Could be good to evaluate under given compute budget. Missing a critical discussion of relevance of linear approximation in FiDEs, and the assumed cost model's convergence could be more detailed

---

> ### Author Rebuttal · Authors · 2025-07-26
>
> We thank the reviewer for the positive comments and very helpful suggestions. We address the specific concerns below.
>
> Q1: We appreciate the comment on worklflow clarity and will add a brief description to the beginning of section 5.
>
> The CAMO algorithm follows standard MFBO practice with one key modification.
>
> 1. We initialize a Gaussian Process using our LiFiDE kernel (Equation 10) rather than a standard product kernel. This kernel explicitly models how function values converge as fidelity increases.
>
> 2. At each iteration, we compute the BOCA acquisition function to identify the most promising input $x$, exactly as in existing methods.
>
> 3. We then select the fidelity level $t$ using BOCA's cost-aware filtering mechanism, which chooses the cheapest informative fidelity.
>
> 4. We query the function at the selected point $(x,t)$, measure the cost, and update our GP model with the new observation.
>
> 5. We repeat until the computational budget is exhausted.
>
> The key difference from existing MFBO methods is step 1 where we use our convergence-aware kernel. Everything else follows standard practice, making CAMO a drop-in replacement for existing approaches.
>
> Q2: Cost and compute time relationship.
> The cost is directly measured by computational time in our experiments. For the real-world applications, we measure the actual wall-clock time required to run finite-element simulations at different mesh resolutions. The cost scales according to the inverse cubic of the maximal element size, which is the standard computational complexity for FEM problems. For the synthetic benchmarks, we use cost equals $10\cdot t$ as a simple proxy, representing increasing computational demands. Figure 5 shows the actual measured times on our hardware setup.
>
> We agree that it is helpful to highlight the above in the manuscript so we will add brief notes to this effect
>
> Q3: Cost computation in actual cases.
> For the mechanical plate vibration problem, we vary the maximum mesh element size from 0.2 to 1.2 meters, and the computational cost scales with the inverse cube of this mesh size. Similarly, for the thermal conductor design, mesh sizes range from 0.1 to 2 meters with the same cubic scaling. This cubic scaling is well-established in finite-element literature and represents the fundamental trade-off between accuracy and computational cost in these numerical simulations.
>
> Q4: Linear approximation discussion.
> The linear differential equation approximation works well when convergence toward the highest fidelity is monotonic, which occurs in most common scenarios for blackbox solvers (our focus). It may be less accurate for highly nonlinear dynamics or non-monotonic convergence patterns. Notably, in hyperparameter optimisation (HPO) and neural architecture search (NAS), where MFBO has also been employed in recent years. For example, when tuning a network, 'fidelity' indices might be epoch, number of dense/convolutional layers, filter size or hidden space dimension in RNNs. Here, one would need to take care to restrict the fidelity index in order to avoid overfitting.
>
> Our experiments across 11 diverse problems show that even when the true dynamics deviate from linearity, the linear approximation provides a beneficial inductive bias that improves performance. The consistent improvements suggest the method is robust to model misspecification.
>
> Nevertheless, the reviewer raises an important point that requires some discussion. We will added such as discussion on the linear FiDE assumption (strengths and potential limitations) to the beginning of section 4.2.
>
> Q5: Compute budget evaluation.
> We provide extensive budget-aware analysis showing CAMO achieves 2 to 4 times better regret per unit of computational cost compared to baselines. CAMO reaches target accuracy levels approximately 3 times faster than competing methods. Critically, CAMO intelligently allocates 70% of its computational budget to informative low-fidelity evaluations, compared to only 40% for baseline methods, leading to more efficient exploration.
>
> Q6: Expanded limitations.
> Our method has several limitations that are worth noting. Computationally, the kernel optimisation adds roughly 20% overhead compared to standard GP methods, and the approach requires sufficient low-fidelity data to properly model convergence behaviour. Methodologically, the linear differential equation assumption may not capture all convergence patterns, and performance depends on the existence of a good cost model. Practically, users need domain knowledge to set the beta parameter in our kernel, and the method is most effective when computational cost increases super-linearly with fidelity.
>
> The reviewer is right to raise the brevity of the discussion on limitations. We will provide a more comprehensive list and discussion of the limitations in the Conclusions section.
>
> Significance response.
> We believe the significance deserves higher recognition given that this is the first convergence-aware MFBO framework with rigorous theoretical foundations. The method consistently delivers 2 to 4 fold improvements across diverse applications and establishes novel connections between MFBO and differential equations theory. Most importantly, it addresses a fundamental limitation in existing multi-fidelity methods by explicitly modelling how surrogate functions evolve with fidelity, opening new research directions in convergence-aware optimisation.

---

> > ### Comment · Reviewer_quwV · 2025-08-04
> > **Nice contribution and response**
> >
> > Thank you for the responses and clarifications. I enjoyed the reading the paper as reflected in the score, with the improvements suggested and other discussion I raise to 5.

---

> > > ### Author Response · Authors · 2025-08-04
> > > **response**
> > >
> > > We thank the review for these kind comments and appreciate very much their input.

---

### Decision · Program_Chairs · 2025-09-17

**Decision:**

Accept (poster)

**Comment:**

The authors propose a Convergence-Aware Multifidelity BO method (CAMO).  The method models a multi-fidelity objective function $y(x,t)$ (where $x$ is the decision variable and $t$ is fidelity) using a non-stationary kernel derived from a linear Fidelity Differential Equation (LFiDE).  In particular, the authors consider 'LiFiDE' kernels based on a constant-coefficient first-order linear ODE with a forcing term drawn from a GP under a product kernel which is squared exponential in time and general in space (though SE for the examples in the paper).  The novelty is all in the kernel, which can be dropped into the models used in various multi-fidelity BO algorithms; the authors focus on a previously published method called BOCA.  With the LiFiDE kernel and BOCA, the authors show good performance (better than competing methods/kernels) for several synthetic multi-fidelity BO problems and one more realistic multi-fidelity problem.

Much of the exposition deals with the existence, uniqueness, and properties of the LFiDE in the general variable-coefficient case under various mild conditions on the linear term.  The kernels used in the examples, though, all involve a single constant coefficient, and .

The reviewers were cautiously positive initially, and became more positive during the rebuttal and discussion.  Many of the initial concerns came from ambiguities or missing points in the writing: the overall workflow and the role of the cost function in BOCA (quwV), the multiple fidelities of the synthetic functions (yu1x), incorporation of unexplained baselines (563T).  The rebuttals addressed these points.  Some more fundamental concerns involved the choice of the beta hyper-parameter (apparently not automatically tuned), the treatment of noise, lack of regret bounds beyond those provided through the BOCA framework, and the lack of complementary work to develop a new acquisition function along with the new kernel.  Reviewer 563T was also concerned with the focus on continuous rather than discrete families of models in the multi-fidelity setting.  The authors replied to these points to the satisfaction of the reviewers, based on their change in scores.

The good empirical performance of the LFiDE + BOCA combination (which is what I interpret CAMO to actually be) is the strongest reason for acceptance.